# A Unified Latent Space disentangled VAE Framework with Robust Disentanglement Effectiveness Evaluation

## Abstract

Evaluating and interpreting latent representations, such as variational autoencoders (VAEs), remains a significant challenge for diverse data types, especially when ground-truth generative factors are unknown. To address this, we unify several state-of-the-art disentangled VAE approaches for latent space disentanglement into one framework – bfVAE. To assess the effectiveness of a disentangled VAE model and enhance latent space interpretability, we propose Feature Variance Heterogeneity via Latent Traversal (FVH-LT) and Dirty Block Sparse Regression in Latent Space (DBSR-LS). To ensure robust interpretability of learned latent space, we develop a greedy alignment strategy (GAS) that mitigates label switching and aligns latent dimensions across runs to set the foundation of result aggregation. We also introduce a convenient scalar latent space separation index (LSSI) based on the GAS-aligned outputs of FVH-LT and DBSR-LS to summarize the overall latent structural separation without knowledge of the ground-truth generative factors. We compare bfVAE to five VAE models and validate the effectiveness FVH-LT, DBSR-LS, and LSSI in on seven tabular and image datasets. Under our examined experimental settings, bfVAE provides a more flexible disentanglement framework achieves more favorable overall trade-off between disentanglement and reconstruction than the benchmark VAE models; FVH-LT and DBSR-LS reliably uncover semantically meaningful and domain-relevant latent structures and generally yield consistent results; and LSSI makes an effective quantitative summary of latent structural separation.

## 1 Introduction

### 1.1 Background and Motivation

Understanding and interpreting latent representations in deep generative models (DGMs) remains a fundamental challenge in the development of efficient and trustworthy AI systems, primarily due to their black-box nature. Disentangled representation learning offers a promising solution by aiming to uncover independent and interpretable generative factors underlying complex data, providing a structured and interpretable latent space (Wang et al., 2024; Shen et al., 2022; Higgins et al., 2018; Paige et al., 2017; Bengio et al., 2013). Studies in different domains further underscore the practical benefits of disentangled representation learning, including medical imaging, single-cell transcriptomics, autonomous driving, and large language models (Liu et al., 2022; Baek et al., 2025; Qin et al., 2023; Pourkeshavarz et al., 2024), highlighting their positive impacts not only for interpretability but also for enhancing robustness in real-world AI systems.

We focus on variational autoencoders (VAEs) (Kingma & Welling, 2013), a widely popular DGM that utilizes latent spaces, in this work. In particular, disentangled VAEs aim to learn structured latent spaces in which distinct latent dimensions (LDs) capture independent and semantically meaningful factors of variation in the data, while retaining reconstruction quality. Disentangled VAEs have been employed in different domains and continue to drive methodological advances, such as in high-resolution image reconstruction (Uppal et al., 2025) and data compression and controlled data augmentation in Global Navigation Satellite System interference classification (Heublein et al., 2025).

Most existing work on disentangled VAE focuses on image data, with relatively limited attention to other data modality, tabular data included. Our empirical studies suggest current disentangled VAE formulations,

although effective for image data, may be ineffective for tabular data. Additionally, current disentanglement evaluation metrics are applicable only to synthetic data where ground-truth generative factors are known to enable quantitative evaluation (Higgins et al., 2017; Kim & Mnih, 2018). Other approaches rely largely on qualitative assessment by visualizing reconstructed images to assign semantic meaning to each LD – an approach not applicable to other data types that lack natural visual representations. In summary, both lines of work have key limitations in their disentanglement evaluation in practice (Locatello et al., 2018).

In summary, there remains a lack of a general framework for disentangled VAEs and disentanglement evaluation techniques and metrics for different data modalities especially when ground-truth generative factors can not be precisely defined – the typical case in real-world applications. Furthermore, the inherent instability of deep neural networks training (e.g., random initialization, stochastic optimization, and local minima, etc) exacerbates the evaluation challenges. Some previous work may ignore the instability and report results from a single run, making results hard to reproduce. Finally, latent spaces are subject to label-switching, where indexing of latent factors changes across independent runs, further complicating the interpretability and reproducibility of the learned representations (Rodríguez & Walker, 2014; Stephens, 2000).

## 1.2 Our Contributions and Findings

We propose *bfVAE* that unifies several state-of-the-art disentangled VAE models in a coherent framework. To address the challenge of interpreting latent representations when ground-truth generative factors are unknown, we introduce *Feature Variance Heterogeneity via Latent Traversal (FVH-LT)* and *Dirty Block Sparse Regression in Latent Space (DBSR-LS)*. Building on the outputs of FVH-LT and DSBR, we further define the *Latent Space Separation Index (LSSI)*, a scalar non-causal surrogate summary of structural separation among latent dimensions (LDs) when the underlying generative factors and their correspondence to latent dimensions are unavailable. To address the instability and uncertainty during VAE training and ensure robustness of FVH-LT and DBSR-LS, we recommend training VAEs multiple times and aggregating the resulting interpretations. Since LDs are subject to label-switching across independent runs, we develop a *greedy alignment strategy (GAS)* to align LDs across models before result aggregation. For each proposed method, we provide the underlying intuition and, where appropriate, the corresponding mathematical rationale.

We run extensive experiments on synthetic and real data of various types and sizes to evaluate the united bfVAE framework and the proposed interpretability tools FVH-LT, DBSR-LS, GAS and LSSI. Across all datasets, bfVAE provides overall more favorable trade-off between disentanglement and reconstruction compared to the benchmark VAE frameworks in the studied experimental settings due to the flexibility in its formulation. LSSI agrees well with existing supervised disentanglement benchmarks, supporting its effectiveness as a metric for latent structural separation. Both FVH-LT and DBSR-LS consistently and reliably uncover disentangled and semantically meaningful LDs. To our knowledge, FVH-LT and DBSR-LS are the first techniques, coupled with the GAS procedure, that evaluate the latent space disentanglement effectiveness in VAEs for multiple data types without precise knowledge of ground-truth generative factors.

## 1.3 Related Work

Building on the classical VAE formulation, various extensions have been proposed to promote disentanglement of latent space for enhanced interpretability, while balancing reconstruction fidelity. $\beta$-VAE (Higgins et al., 2017) introduces a hyperparameter $\beta$ that scales the Kullback–Leibler (KL) divergence in the classical VAE objective with $\beta = 1$ – which we refer to as vanilla VAE in this work. Larger $\beta$ may lead to a higher degree of disentanglement. Disentangled $\beta$-VAE (Burgess et al., 2018) introduces an additional capacity parameter $C$ to $\beta$-VAE that controls the information transmitted from the input space to the latent space. Factor-VAE (Kim & Mnih, 2018) adds a total correlation penalty to the vanilla VAE, reducing dependencies among the LDs. $\beta$-TCVAE (Chen et al., 2018) arrives at a closely related objective by decomposing the KL term in the ELBO into index-code mutual information, total correlation, and dimension-wise KL, and promotes disentanglement by imposing a stronger penalty on the total correlation term. DIP-VAE (Kumar et al., 2017) penalizes the off-diagonal elements in the covariance matrix of the expected posterior means over the marginal distribution of data (DIP-VAE-I) or the posterior covariance matrix over its marginal variational distribution (DIP-VAE-II). Li et al. (2025) generalizes the total correlation penalty to a partial correlation penalty. Disentanglement is also explored in conditional VAE (cVAE) models (Sohn et al.,

2015; Harvey et al., 2021; Zhang et al., 2020) and integrated structural causal models with a bidirectional generative process to achieve causal disentanglement (Shen et al., 2022).

Prior work has proposed disentanglement metrics that require ground-truth generative factors. Higgins et al. (2017) measure disentanglement by fixing one factor, varying the others, computing average latent differences, and training a classifier to identify the fixed factor. The factor-VAE metric (Kim & Mnih, 2018) similarly uses latent differences, selecting the LD with the smallest empirical variance and applying majority voting. SAP (Kumar et al., 2017) measures the gap between the top two LDs predicting each factor, while Chen et al. (2018) use mutual information between top LDs and factors. DCI (Eastwood & Williams, 2018) trains regressors to estimate each LD's importance for predicting generative factors.

## 2 A Unified Disentangled VAE framework

In this section, we first introduce bfVAE as a unified framework for disentangled VAE, and then provide an information bottleneck perspective on the formulation.

### 2.1 bfVAE

bfVAE extends the existing $\beta$-VAE and factor-VAE models (thus the name bfVAE). Denote the input by $\mathbf{x} = \{\mathbf{x}_i\}_{i=1}^n$, where $\mathbf{x}_i$ denotes the $i$-th observed sample. Consider a standard VAE setting with prior $p(\mathbf{z}_i) = \mathcal{N}_K(0, I) \, \forall i$ for the latent variables $\mathbf{Z}$ of dimension $K$. Denote the variational posterior of $\mathbf{Z}$ by $q(\mathbf{z}_i|\mathbf{x})$. The bfVAE loss is defined as

$$\mathcal{L}(\boldsymbol{\phi}, \boldsymbol{\omega}) = \frac{1}{n} \sum_{i=1}^n \left\{ \underbrace{-\mathbb{E}_{q(\mathbf{z}_i|\mathbf{x}_i, \boldsymbol{\phi})} \log p(\mathbf{x}_i \mid \mathbf{z}_i, \boldsymbol{\omega})}_{\text{T1: reconstruction error}} + \underbrace{\beta \cdot |\text{KL}\left(q(\mathbf{z}_i \mid \mathbf{x}_i, \boldsymbol{\phi})\|p(\mathbf{z}_i)\right) - C|}_{\text{T2: capacity constraint}} \right\}$$
$$+ \quad \gamma \cdot \underbrace{\text{KL}\left(q(\mathbf{z})\|\bar{q}(\mathbf{z})\right)}_{\text{T3: disentanglement regularization}}, \tag{1}$$

where $\boldsymbol{\omega}$ and $\boldsymbol{\phi}$ are unknown model parameters; and $\beta \geq 0, \gamma \geq 0$, and $C \geq 0$ are hyperparameters.

Different terms in Eq. 1 serve different purposes. T1 ensures reconstruction fidelity by maximizing the expected log-likelihood. T2 penalizes the distance between the prior-posterior KL divergence and target $C$: if $\text{KL} = C$, T2 disappears. $C$ is referred to as the target "capacity in that it constrains the latent space's capacity – larger $C$ allows the posterior to extract more information from the input data and to drift further from the prior. During training, $C$ starts at 0 and gradually increases to reach the pre-specified value for a controlled capacity expansion. T3 encourages statistical independence among the LDs, where $q(\mathbf{z}) := n^{-1} \sum_i q(\mathbf{z}|\mathbf{x}_i)$ and $\bar{q}(\mathbf{z}) := \prod_{j=1}^K q(z_j)$. Some existing VAE models are special cases of Eq. 1 by setting $\beta, \gamma$, and $C$ at specific values: when $\gamma = 0, C = 0$, bfVAE reduces to $\beta$-VAE and when $\beta = 1, C = 0$, it reduces to factor-VAE; when $\gamma = 0, C = 0, \beta = 1$, it becomes the vanilla VAE.

### 2.2 An Information Bottleneck Perspective on bfVAE

We can interpret the bfVAE formulation in Eq. 1 via the variational information bottleneck framework (Alemi et al., 2016; Burgess et al., 2018). First, the bfVAE formulation in Eq. 1 is not a classical information bottleneck objective in the form of $\max \left\{ I(\mathbf{Z}; \mathbf{Y}) - \beta I(\mathbf{X}; \mathbf{Z}) \right\}$[1] (Tishby et al., 2000), but inspired by it. Each of the three terms in Eq. 1 plays a different but complementary role in shaping the latent space in bfVAE.

Specifically, T1 in Eq. 1 encourages $\mathbf{z}$ to retain information needed to reconstruct the input $\mathbf{x}$. If optimized alone, this term would favor highly informative latent representations, resembling a standard autoencoder, with little pressure to compress or organize the latent space. T2 constraints the mutual information between $\mathbf{x}$ and $\mathbf{z}$ and encourages the aggregated posterior $q(\mathbf{z})$ to match the prior, which can be easily seen by decomposing the KL divergence $\mathbb{E}_{p(\mathbf{x})}\left[\text{KL}(q_{\boldsymbol{\phi}}(\mathbf{z} \mid \mathbf{x})\|p(\mathbf{z}))\right] = I_q(\mathbf{x}; \mathbf{z}) + \text{KL}(q(\mathbf{z})\|p(\mathbf{z}))$. $C$ and $\beta$ in T2 have distinct roles. $C$ specifies the *target bottleneck capacity* or the desired amount of information that $\mathbf{z}$ is allowed to carry: smaller $C$ imposes a narrow bottleneck and forces aggressive compression from $\mathbf{x}$ to $\mathbf{z}$, whereas

---

[1]$\mathbf{X}$ is the input, $\mathbf{Y}$ is the target, and $\mathbf{Z}$ is a representation of $\mathbf{X}$ that is maximally informative about $\mathbf{Y}$ while being as compressed as possible with respect to $\mathbf{X}$. $\beta$ acts as the Lagrange multiplier that regulates information of $\mathbf{X}$ preserved in $\mathbf{Z}$.

larger $C$ permits more information to pass through. $\beta$, on the other hand, controls *how strongly this target is enforced*. In $\beta$-VAE, the KL term is simply controlled by $\beta$. By introducing $C$, the model can control the bottleneck more directly in the sense that it is not merely encouraged to reduce KL, but rather to allocate approximately $C$ units of latent capacity, leading to more direct control over the reconstruction-compression tradeoff. T3 penalizes redundancy and dependence among $\mathbf{z}$, encouraging the retained information to be organized into approximately independent and factorized components. Different from T2, which primarily controls how much information is transmitted, T3 determines how the transmitted information is structured after passing through the bottleneck. In summary, the bfVAE formulation extends the standard variational bottleneck from controlling only the quantity of latent information to also shaping its structure across dimensions. The resulting latent space is expected to be not only informative and compressed, but also more interpretable and disentangled.

The interpretation above considers the latent representation as a whole. At a finer level, the effect of Eq. 1 can be expressed unevenly across individual LDs. Specifically, under the joint action of T1, T2, and T3, some LDs actively encodes nontrivial information in $\mathbf{x}$, whereas others contribute little. This motivates a dimension-wise characterization of the latent space, formalized in Def. 1. This distinction provides a more refined view of how the bottleneck allocates information across LDs.

**Definition 1 (informative and noninformative latent dimensions)** *A latent dimension $j$ in a VAE is informative if the expected KL divergence between its posterior and prior distributions exceeds an information threshold $\epsilon$:*

$$\mathbb{E}_{p(\mathbf{x})}\big[\mathrm{KL}(q_{\boldsymbol{\phi}}(z_j|\mathbf{x},\boldsymbol{\phi})\|p(z_j))\big] > \epsilon, \; where \; \epsilon \gg 0. \tag{2}$$

*A latent dimension $j$ is non-informative (or collapsed) if its posterior conveys negligible information about the input $\mathbf{x}$:*

$$\mathbb{E}_{p(\mathbf{x})}\big[\mathrm{KL}(q_{\boldsymbol{\phi}}(z_j|\mathbf{x},\boldsymbol{\phi})\|p(z_j))\big] < \delta, \; for \; a \; vanishingly \; small \; \delta. \tag{3}$$

In the case of $p(\mathbf{z}) = \mathcal{N}(0,1)$ and $q(z_j|\mathbf{x}) = \mathcal{N}(\mu_j(\mathbf{x}), \sigma_j^2(\mathbf{x}))$ – common choices in VAEs, an informative LD $j$ means $q(z_j|\mathbf{x})$ has successfully "escaped the information bottleneck, achieved through either a non-zero $\mu_j(\mathbf{x})$ or $\sigma_j^2(\mathbf{x})$ away from 1. When $q(z_j|\mathbf{x},\boldsymbol{\phi}) \approx p(z_j)$, the $j$-th LD effectively behaves as noise. We note that while an informative LD encodes meaningful information in $\mathbf{x}$, it does not automatically offer semantic interpretation on what information it encodes (thus the need for techniques like FVH-LT and DBSR-LS that are introduced in Sections 3.1 and 3.2).

We hypothesize that the relative prevalence of informative and non-informative LDs is governed jointly by $C$ and $\beta$ of T2, but also relates to the input type and characteristics of $\mathbf{x}$. Larger $C$ generally leads more informative LDs, particularly when accompanied by large $\beta$, which strongly enforces the target. For fixed large $C$, a small $\beta$ may yield an intermediate regime in which the separation between informative and non-informative LDs is less distinct. In contrast, smaller $C$ imposes a tighter bottleneck and leads to more non-informative LDs; similarly, this effect becomes stronger as $\beta$ increases. Previous work on $\beta$-VAE for image data often adopts relatively large $\beta$; since $C = 0$ in standard $\beta$-VAE, this can be viewed as placing strong pressure toward a low-capacity information bottleneck. This forces the model to encode only the aspects of $\mathbf{x}$ that most improve reconstruction, which may work well for images as high-dimensional images often contain substantial redundancy across input features. In contrast, for relatively low-dimensional $\mathbf{x}$ or when semantics are more dispersed across $\mathbf{x}$, as is often the case for tabular data, there is typically less redundancy and less compressible information. Applying the same bottleneck setting may discard substantial signal, reduce the number of "true informative LDs, and degrade reconstructed or synthetic data. Instead, a larger $C$ with an appropriately tuned $\beta$ may allow the latent space to retain more nontrivial information from $\mathbf{x}$. Our experiments in Sec. 4 provide empirical evidence in support of this claim.

## 3 Latent Space Interpretability Techniques

While bfVAE promotes informative and orthogonal LDs through capacity and dependence regularization, it itself does not explicitly attach semantic meaning to each LD or identify feature–latent associations, motivating us to propose FVH-LT and DBSR-LS as dedicated latent space interpretability techniques to quantify, visualize, and interpret the disentangled LDs To enable stable aggregation of FVH-LT and DBSR-LS results across multiple evaluation runs, we introduce the GAS procedure to resolve LD misalignment and

set the foundation for result aggregation. Finally, we proposed the LSSI to quantify the degree of structural separation captured by the latent–feature association matrix output from FVH-LT and DBSR-LS.

## 3.1 FVH-LT

FVH-LT quantifies the associations between input features and LDs through LT. As illustrated in Fig. 1, the trained VAE encoder first maps each input instance to its latent representation. FVH-LT then varies one LD at a time while holding the remaining LDs fixed, and measures the resulting changes in the outputs reconstructed by the decoder. The resulting LD-feature association matrix $\mathbf{S}$ summarizes how strongly each feature responds to variations in each LD and can be conveniently visualized as a heatmap, providing an interpretable understanding of individual LD. FVH-LT does not require access to ground-truth generative factors or external supervision, making it applicable to real-world data where such information is unavailable.

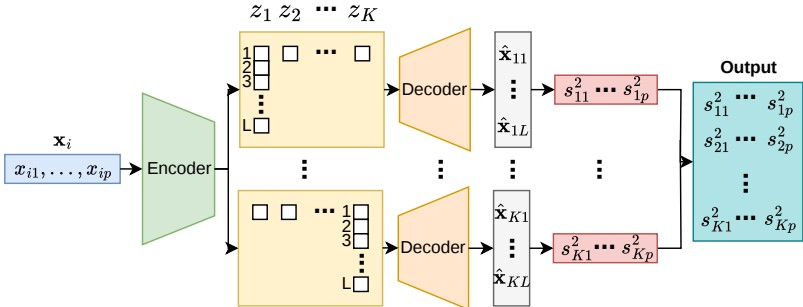

Figure 1: The FVH-LT Workflow. $\mathbf{x}_i$ contains input features for the $i$-th observation; $z_k$ is the $k$-th LD and $\hat{\mathbf{x}}_{ikl}$ for $k=1,\ldots,K$ and $l=1,\ldots,L$ is reconstructed output for $\mathbf{x}_i$ at $l$-th fixed value from the LT of $z_k$; $s_{kj}$ is the sample variance of the restructured data for the $j$-th feature from the LT on LD $k$.

Alg. 1 presents the pseudo-code for FVH-LT. The LT ranges can be defined in several ways: 1) a fixed range across all $n$ input samples and $K$ LDs; 2) a data- and dimension-dependent range, such as $[\mu_{ik} - c\sigma_{ik}, \mu_{ik} + c\sigma_{ik}]$, where $\hat{\mu}_{ik}$ and $\hat{\sigma}_{ik}^2$ are the learned posterior mean and variance of the $k$-th LD for observation $i$; 3) a combination of the previous two: range centered at $\mu_{ik}$ with a global interval half-width. In our experiments, we find the latter two strategies are more effective in identifying informative LDs as the fixed LT ranges in the first may fail to probe high-density regions of the posterior when $\mu_{ik}$ deviates from 0, reducing sensitivity of the LT. As for the number of LT steps $L$, a large $L$ creates a fine grid for LT to better detect pattern changes across the LD; a small $L$ would lead to a coarse grid, potentially obscuring meaningful signals, but at a lower computational cost. Because the true number of informative LDs $K_0$ is unknown, one needs to supply an estimate $K$. The good news is that FVH-LT is robust to the choice of $K$ as long as $K > K_0$.

FVH-LT can be interpreted as a decoder-mediated latent-to-feature response analysis. For notation simplicity, we drop the index $r$ in the following analysis. Denote the learned decoder by $\hat{g} : \mathbb{R}^K \to \mathbb{R}^p$. Applying a first-order Taylor expansion and the variance operator to $g$ at a fixed point $z_{ik}^*$ along LD $k$ for sample $\mathbf{x}_i$, we have $\hat{g}_j(z_{ik}^* + \delta_{ikl}) \approx \hat{g}_j(z_{ik}^*) + \delta_{ikl}\left[J_{\hat{g}}(\mathbf{z}_i^*)\right]_{jk}$, where $\mathbf{z}_i^* = (\mathbf{z}_{i,-k}, z_{ik}^*)$, $J_{\hat{g}}(\mathbf{z}_i^*) \in \mathbb{R}^{p \times K}$ is the decoder Jacobian, and $\delta_{ikl}$ can be regarded as the adversarial perturbation in LD $k$ from LT, and

$$s_{ijk}^2 = \hat{\mathrm{V}}\left[\hat{g}_j(z_{ik}^* + \delta_{ikl})\right] \approx \left[J_{\hat{g}}(\mathbf{z}_i^*)\right]_{jk}^2 \hat{\mathrm{V}}(\delta_{ikl}).$$

If the LT points are equally spaced over the LT range $(-a_{ik}, a_{ik})$, which is what we use in the experiments, then $\widehat{\mathrm{Var}}(\delta_{ikl}) = (2a_{ik})^2 L(L+1)/(12(L-1)^2) = a_{ik}^2 L(L+1)/(3(L-1)^2)$. In other words, FVH-LT measures the feature-level variation induced by controlled finite-grid perturbations of each LD. When the LT-grid variance is fixed across LDs, the FVH-LT entry is proportional to the squared local sensitivity of reconstructed feature $j$ to perturbations in LD $k$.

## 3.2 DBSR-LS

DBSR promotes shared-sparsity pattern and task-specific sparsity pattern in regression coefficients in a multi-task supervised learning setting (Jalali et al., 2010). We make a novel use of DBSR towards latent

---

**Algorithm 1** The FVH-LT procedure

---
1: **Input**: training data $\mathbf{x}_{n \times p}$, prior $\mathbf{z} = (z_1 \ldots, z_K) \sim f(\mathbf{z})$, LT range $[-a_{ik}, a_{ik}]$ for $k = 1, \ldots, K$ and $i = 1, \ldots, n$, LT steps $L$, run number $R$, hyper-parameter initialization
2: **Output**: Variance matrix $S$ of reconstructed $\hat{\mathbf{x}}$ given LT in $\mathbf{z}$; posterior-prior KL divergence
3: Pre-processing: standardize $\mathbf{x}$ as needed
4: **for** $r = 1$ to $R$ **do**
5:     Train $\text{VAE}_r$ on $\mathbf{x}$ with the loss function in Eq. 1
6:     **for** $i = 1$ to $n$ **do**
7:         **for** $k = 1$ to $K$ **do**
8:             Calculate $d_{i,rk} = D_{\text{KL}}(\hat{q}_k(z_{ik}|\mathbf{x}_i)||p(z_{ik}))$
9:             Generate $\mathbf{z}_{ik} = \{z_{ikl}\}_{k=1,\ldots,L}$ over its LT range $[-a_{ik}, a_{ik}]$, fixing $\mathbf{z}_{i,-k}$ at randomly sampled values from $\hat{q}_{-k}(\mathbf{z}_{i,-k}|\mathbf{x}_i)$
10:             Feed $\{z_{ikl}, \mathbf{z}_{i,-k}\}$ to the decoder of $\text{VAE}_r$ to reconstruct $\{\hat{x}_{irjl,k}\}_{j=1,\ldots,p}$ for $l = 1$ to $L$
11:             Calculate sample variance $s^2_{irj,k}$ of $(\hat{x}_{irj1}, \ldots, \hat{x}_{irjL})$ for $j = 1$ to $p$
12:         **end for**
13:     **end for**
14:     Calculate $\bar{s}^2_{rjk} = n^{-1} \sum_{i=1}^n s^2_{irj,k}$
15: **end for**
16: Apply GAS (Alg. 3) to align $\bar{s}^2_{rj,k}$ across $r = 1, \ldots, R$ runs
17: Compute $s^2_{jk} = \frac{1}{R} \sum_{r=1}^R \bar{s}^2_{rj,k}$ for $j = 1, \ldots, p$ and $d_{ik} = \frac{1}{R} \sum_{r=1}^R d_{i,rk}$ for $k = 1, \ldots, K$; $i = 1, \ldots, n$
18: **Return** variance matrix $S = [s^2_{jk}]_{j=1,\ldots,p;k=1,\ldots,K}$ and KL divergence $\mathbf{d} = \{d_{ik}\}_{i=1,\ldots,n;k=1,\ldots,K}$

---

space separation and interpretability by treating each LD as the outcome of a different regression task and leveraging the shared and task-specific sparsity to identify feature–latent associations. This formulation decomposes the associations into components unique to individual LDs and those shared across multiple LDs, providing a structured view of how information is distributed across LDs. In particular, DBSR-LS estimates two regression coefficient matrices $D$ and $B$ via minimizing the following loss

$$\mathcal{L}(D, B) = (2n)^{-1} \sum_{k=1}^K \left\| \boldsymbol{\mu}_Z^{(k)} - \left( D_{[k,]} + B_{[k,]} \mathbf{x}^{(k)} \right) \right\|_2^2 + \lambda_D \|D\|_{1,1} + \lambda_B \|B\|_{1,\infty}. \tag{4}$$

$\boldsymbol{\mu}_{Z,n \times 1}^{(k)}$ contains the posterior means of the $k$-th latent variables of the $n$ samples and $\mathbf{x}_{n \times p}^{(k)}$ is the corresponding design matrix comprising the observed features. $D$ and $B$ are both of dimension $K \times p$ with $D_{[k,]}$ and $B_{[k,]}$ denoting their $k$-th columns. While both $\|D\|_{1,1} = \sum_{j=1}^p \sum_{k=1}^K |D_{kj}|$ and $\|B\|_{1,\infty} = \sum_{j=1}^p \max_k |B_{kj}|$ promotes sparsity, the former captures the latent-dimension-specific sparsity whereas the latter represents shared sparse structures across the LDs. As a result, nonzero entries in $D$ indicate features associated distinct LDs. Eq. 4 is formulated using the posterior means $\boldsymbol{\mu}_Z$ as the response variable. An alternative is to use random samples from the posterior distribution though we expect $\boldsymbol{\mu}_Z$ yields more stable estimates.

Alg. 2 lists the pseudo-code for DBSR-LS. For hyperparameters shared with Alg. 1, their specifications are similar. For those unique to Alg. 2 (i.e., $\lambda_B \geq 0, \lambda_D \geq 0$), they can be tuned to balance sparsity in the regression coefficients with prediction accuracy – large values may lead to overly sparse $\hat{D}$ while small values may limit the disentanglement capacity of DBSR-LS.

DBSR-LS is proposed as an interpretable linear surrogate for summarizing first-order LD-feature associations and is not intended to replace the encoder or provide a complete predictive model of $q_\phi(z|x)$ (the encoder itself would be sufficient for that reason and there would be no need to train a linear surrogate). The dirty-model structure is useful because it separates shared associations from LD-specific associations. If different LDs are genuinely separated, their sparse feature-association profiles should differ; if multiple LDs rely on the same features, the DBSR-LS matrix will reveal this overlap. DBSR-LS estimates LD-feature associations directly from learned latent space and is not decoder-mediated like FVH-LT; it may thus serve as a useful complementary approach when decoder-based LT is unstable.

---

**Algorithm 2** The DBSR-LS procedure

---

1: **Input**: training data $\mathbf{x}_{n \times p}$, prior $\mathbf{z} = (z_1 \ldots, z_r) \sim f(\mathbf{z})$, initialized hyper-parameters ($\lambda_B$, $\lambda_D$, those in VAE), run number $R$

2: **Output**: latent-dimension-specific aestimated coefficient matrix $\hat{D}$ and its absolute value $|\hat{D}|$

3: Pre-processing: standardize $\mathbf{x}$ as needed

4: **for** $r = 1$ to $R$ **do**

5:     Train $VAE_r$ on $\mathbf{x}$ with the loss in Eq. 1 and extract posterior means $\boldsymbol{\mu}_{n \times K, r}$ of $q(\mathbf{z}|\mathbf{x})$

6:     Compute $d_{r,ik} = D_{\mathrm{KL}}(q_k(z_{ik}|\mathbf{x})||p(z_{ik}))$ for $k = 1, \ldots, K$ and $i = 1, \ldots, n$

7:     Run DBSR-LS with $\mathbf{x}$ as the design matrix and $\boldsymbol{\mu}_r[, k]$ as $Y^{(k)}$ to estimate $D_r$ and $B_r$

8: **end for**

9: Apply GAS (Alg. 3) to align $D_r$ and $|D_r|$, respectively, across $r = 1, \ldots, R$

10: Calculate the averages $|\hat{D}| = R^{-1} \sum_{r=1}^{R} |D_r|$.

---

### 3.3 GAS for Latent Space Alignment

Aggregating FVH-LT output $S$ and the DBSR-LS output $|\hat{D}|$ in Algs. 1 and 2 across multiple runs can stabilize learned LD-feature associations results. However, such aggregation is complicated by label switching in the latent space.[2] Consequently, latent representations are not directly comparable on a dimension-by-dimension basis cross runs, and naïve aggregation would yield misleading results.

To address this issue, we introduce GAS to align LDs in $S$ and $|\hat{D}|$ before their aggregation. In a nutshell, GAS clusters LDs in each run based on their informativeness defined in Def. 1, selects the run with the maximum number of informative LDs as the reference, and uses it as an anchor for alignment. The matched LDs are then re-indexed to establish consistent LD correspondence across runs. Alg. 3 lists the pseudo-code, which has been integrated in Alg. 1 (Line 16) and Alg. 2 (Line 9), and Fig. 2 illustrates the workflow.

---

**Algorithm 3** The GAS

---

1: **Input**: KL divergence $\mathbf{d} = \{d_{rk}\}_{r=1,\ldots,R;\ k=1,\ldots,K}$, matrix $\{A_r\}_{r=1,\ldots,R}$ to be aligned (variance matrix $S$ from FVH-LT, coefficient matrix $|D|$ from DBSR-LS), correlation threshold $\rho$

2: **Output**: aligned $A$

3: In each run $r = 1$ to $R$, cluster the LDs into two groups based on $\{d_{rk}\}_{k=1,\ldots,K}$; define the cluster with the higher average KL as the informative set $\mathcal{I}_r$.

4: Define $r^* \overset{\triangle}{=} \arg\max_{r=1,\ldots,R} |\mathcal{I}_r|$

5: **for** $r \neq r^*$ **do**

6:     Define $\mathbf{C}_{|\mathcal{I}_{r^*}| \times |\mathcal{I}_r|}$, where $C_{ij} = \mathrm{corr}(v_{r^*i}, v_{rj})$ for $i \in \mathcal{I}_{r^*}, j \in \mathcal{I}_r$

7:     Initialize position mapping $\mathcal{F}_r = \emptyset$

8:     **while** unmatched indices remain in both $\mathcal{I}_{r^*}$ and $\mathcal{I}_r$ & $\max(\mathbf{C}) > \rho$ **do**

9:         $(k, j) \overset{\triangle}{=} \arg\max_{i',j'} C_{i'j'}$

10:         $\mathcal{F}_r \leftarrow \{\mathcal{F}_r, j \to i\}$; $\mathbf{C} \leftarrow \mathbf{C}_{-i,-j}$

11:     **end while**

12:     Randomly match remaining positions in $(A_{r^*}, A_r)$

13:     Re-align by LDs: $A_r \leftarrow \mathcal{F}_r(A_r)$

14: **end for**

---

The GAS procedure is designed as a greedy and computationally efficient (see Sec. 3.4) alignment strategy for repeated VAE runs. Its purpose is not to solve a generic assignment problem over all LDs, but to align informative LDs of the same LD-feature association structure to set the foundations to aggregate disentanglement results. Standard post-hoc matching methods, such as Hungarian matching (Kuhn, 1955; Munkres, 1957), could be considered as alternative solvers for the alignment subproblem within GAS, but they can not replace GAS and several critical steps from Alg. 3 remain essential for properly solving the

---

[2]Specifically, LDs obtained across different runs are not necessarily aligned even when $K$ is fixed. For example, the first LD in one run may correspond to the third LD in another run.

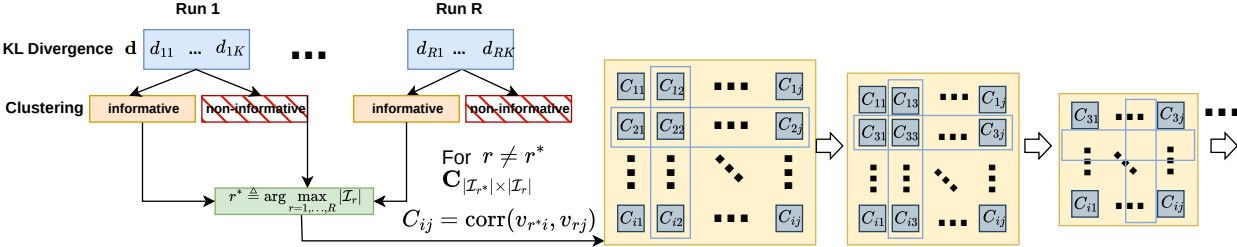

Figure 2: The GAS procedure. $d_{rk}$ denotes the prior-posterior KL divergence of $k$-th LD; $C_{ij}$ represents the correlation between the $i$-th LD in reference run $r^*$ and the $j$-th LD in run $r$ in the assocation matrix output from FVH-LT (variance matrix $S$) or DBSR-LS (regression coefficient matrix $|D|$).

alignment problem in this setting [3]. More broadly, comparing GAS with alternative alignment strategies, including Hungarian matching, optimal transport, and unbalanced matching methods (Villani, 2009; Peyré & Cuturi, 2019; Chizat et al., 2018; Liero et al., 2018), is an interesting direction for future work.[4]

### 3.4 Computational Cost and Scalability of FVH-LT, DBSR-LS, and GAS

Let $n$ denote the number of samples, $p$ the number of input features, $K$ the number of latent dimensions (LDs), $L$ the number of LT steps, $R$ the number of repeated training runs, and $T$ the number of solver iterations used by DBSR-LS. FVH-LT evaluates each trained model by traversing each LD over $L$ grid points for each sample and then computing feature-wise variances of the decoded outputs. Ignoring the cost of VAE training itself, the FVH-LT evaluation cost is $\mathcal{O}(RnKL(c_{\text{dec}} + p)\}$, where $c_{\text{dec}}$ denotes the cost of one decoder forward pass. If $K$ and the decoder size are treated as fixed, this reduces to $\mathcal{O}(RLn)$.

For DBSR-LS, the main cost comes from solving the sparse multi-task regression problem (the original DBSR work (Jalali et al., 2010) does not provide a computational complexity analysis). For matrix-free proximal-gradient implementation of DBSR-LS, it requires $\mathcal{O}(Knp)$ operations per iteration to compute the least-squares gradients. The proximal update costs $\mathcal{O}(Kp)$ for the element-wise $\ell_{1,1}$ penalty and $\mathcal{O}(pK \log K)$ or $\mathcal{O}(pK)$ for the $\ell_{1,\infty}$ penalty, depending on whether a sorting-based or linear-time projection is used. The total cost is thus $\mathcal{O}\left(T(Knp + pK \log K)\right)$. For a Gram-matrix implementation, it first forms $X_k^\top X_k$ and $X_k^\top y_k$ for each task with cost $\mathcal{O}(Knp^2)$. Each iteration costs $\mathcal{O}(Kp^2 + Knp + pK \log K)$, where $\mathcal{O}(Kp^2)$ comes from multiplying by the Gram matrices and $\mathcal{O}(Knp)$ term may arise from objective evaluation or backtracking line search. The total complexity is $\mathcal{O}\left(Knp^2 + T(Kp^2 + Knp + +pK \log K)\right)$, which is consistent with follow-up analyses of dirty-model implementations (Rose, 2014). This explains why DBSR-LS is more expensive than FVH-LT when $p, K$ or $T$ is large.

In contrast, the additional cost of GAS is comparatively small. GAS is applied after FVH-LT or DBSR-LS has already produced a $K \times p$ LD-feature association matrix per run. Let $k_r = |\mathcal{I}_r| \le K$ denote the number of informative LDs in run $r$, then computing the correlation matrix between the reference run $r^*$ and run $r$ costs $\mathcal{O}(k_{r^*} k_r p)$; and the greedy matching step costs $\mathcal{O}(\max k_r^2 \log(\max k_r))$ if the pairwise correlations are sorted once. So the total GAS cost across $R$ runs is approximately $\mathcal{O}(\sum_{r \ne r^*} pk_{r^*}k_r) + \mathcal{O}(R(\max k_r^2)\log(\max k_r))$, Since $K_r \le K$ is typically very small compared with $n$ and $p$, this GAS cost is usually much smaller than the cost of VAE training, FVH-LT, and DBSR-LS.

For practical applications, there are multiple way to reduce computational time if needed. Since repeated runs are independent, they can be parallelized across CPUs/GPUs; within FVH-LT, LT can be also batched and parallelized over samples, LDs, and LD grid points; GAS is easily vectorized because it consists primarily of matrix correlations and greedy matching over a small matrix. For high-dimensional data of large $p$, we recommend FVH-LT as the default procedure and using DBSR-LS selectively, for example on preselected features or domain-defined feature groups.

---

[3]Hungarian matching provides a globally optimal one-to-one assignment once the cost matrix and the two sets of objects are fixed. A naïve application to all LDs would force matches even among noninformative LDs. A more meaningful application would still require the followings steps used in GAS: LD clustering, reference-run selection, handling unequal numbers of informative LDs across runs, and thresholding of weak matches.

[4]Such a comparison would require a dedicated benchmark of post-hoc alignment algorithms for stochastic VAE latent spaces, which is beyond the scope of the present work that focuses on disentangled VAEs and ground-truth-free interpretability.

### 3.5 Latent Space Separation Index (LSSI)

We introduce LSSI for two reasons. First, as discussed in Sec. 1, existing disentanglement metrics require access to ground-truth generative factors, which is often unavailable in real-world applications. In contrast, LSSI is computed without ground-truth factor information, which also implies that LSSI should be interpreted as a non-causal proxy for disentanglement-oriented separation metrics [5]. Second, while FVH-LT and DBSR-LS produce a quantitative LD-feature association matrix $\in \mathbb{R}^{K \times p}$ that is most effectively visualized as heatmaps. LSSI complements this visual interpretation by proving a convenient single-number summary of the latent structural separation, which is especially useful when $p$ is large.

We motivate the mathematical formulation of LSSI from an axiomatic perspective. A well-separated latent representation should have the following properties: 1) different informative LDs should have minimally overlapping feature-association profiles; 2) LDs with similar feature-association profiles should be discounted; 3) The level of informativeness of LDs should be considered; 4) non-informative and weakly informative LDs should not be treated as meaningful factors and be discounted; and 5) the score should be invariant to the overall scale of the association matrix $A$.

**Definition 2 (LSSI)** *Let $A \in \mathbb{R}^{K \times p}$ denote the output matrix from FVH-LT or DBSR-LS, $A_{[k,]} \in \mathbb{R}^p$ denote the k-th row of A, and $\ell_1$-norm $\|A_{[k,]}\|_1 = \sum_{i=1}^{p} |A_{k,i}|$. Let $\tau$ be the p-th largest value among all entries in $|A|$ and $K_{\max}$ be the maximum number informative LD from R runs.*

- *When true generative factors are unknown (the most likely practical setting): Define $\tilde{A}_{ki}^{(\tau)} := |A_{ki}| \cdot \mathbf{1}\{|A_{ki}| \geq \tau\}$, $\rho_k := \|\tilde{A}_{[k,]}^{(\tau)}\|_1$, $\pi_k := \rho_k \left(\sum_{\ell=1}^{K} \rho_\ell\right)^{-1}$, and $K_{\text{eff}} := \left(\sum_{k=1}^{K} \pi_k^2\right)^{-1}$ (the effective number of informative LD), let $\mathcal{P} = \{(k,j) : 1 \leq k < j \leq K\}$ denote the set of all unordered pairs of LDs, then*

$$\text{LSSI}(A) = c_1 c_2 \left( \frac{\sum_{(k,j) \in \mathcal{P}} \|\, |A_{[k,]}| - |A_{[j,]}| \,\|_1}{\sum_{(k,j) \in \mathcal{P}} \left( \|A_{[k,]}\|_1 + \|A_{[j,]}\|_1 \right)} \right) \in [0,1], \tag{5}$$

$$\text{where } c_1 = \frac{|\{i : \exists k \text{ such that } |A_{ki}| \geq \tau\}|}{p} \text{ and } c_2 := \min\left\{ \frac{K_{\text{eff}} - 1}{K_{\max} - 1}, 1 \right\}, \tag{6}$$

- *When ground-truth generative factors are known (the benchmark setting for comparison with existing disentanglement metrics that rely on ground-truth factors): let $\mathcal{P}_0 = \{(k,j) : 1 \leq k < j \leq K_0\}$ denote the set of all unordered pairs of LDs that correspond to the $K_0$ ground-truth factors, then*

$$\text{LSSI}_0(A) = c_1 c_2 \left( \frac{\sum_{(k,j) \in \mathcal{P}_0} \|\, |A_{[k,]}| - |A_{[j,]}| \,\|_1}{\sum_{(k,j) \in \mathcal{P}_0} \left( \|A_{[k,]}\|_1 + \|A_{[j,]}\|_1 \right)} \right) \in [0,1],$$

$$\text{where } c_1 \text{ is the same as in Eq. 6 and } c_2 = \frac{|\{k : \exists i \text{ such that } |A_{ki}| \geq \tau\} - 1|}{K_0 - 1}. \tag{7}$$

$K_{\text{eff}}$ in Def. 2 is known as the participation-ratio or inverse-Simpson effective number (Bell & Dean, 1970; Wegner, 1980; Hill, 1973; Jost, 2006). It equals $K_{\max}$ when the signal is evenly distributed over $K_{\max}$ LDs and approaches 1 when the signal is concentrated in a single LD. Thus, it provides a soft measure of how many LDs meaningfully contribute, while penalizing highly concentrated row-mass distributions.

The pairwise component of LSSI is $d(k,j) = (\|\, |A_{[k,]}| - |A_{[j,]}| \,\|_1)(\|A_{[k,]}\|_1 + \|A_{[j,]}\|_1)^{-1}$, which measures the normalized non-overlap between two the feature association profiles in rows $k$ and $j$. Let $w_{kj} := \|A_{[k,]}\|_1 + \|A_{[j,]}\|_1$, then third term in LSSI is $\sum_{k<j} w_{kj} d(k,j)(\sum_{k<j} w_{kj})^{-1}$, which is a mass-weighted average and measures how distinct the LD-feature association profiles are across LDs. The factors $c_1$ and $c_2$ serve as correction terms for this raw separation score. In particular, $c_1$ measures feature coverage, ensuring that a high LSSI value is assigned only when the learned LDs explain a substantial portion of the input features.

---

[5] In the strongest sense, disentanglement is often understood as recovering independent or causally meaningful factors of variation (Bengio et al., 2013). Establishing such a causal interpretation generally requires known ground-truth factors, interventions, or additional domain assumptions. Existing disentangled VAE objectives, $\beta$-VAE and factor-VAE included, encourage separation of LDs through KL-based or total-correlation regularizers which target statistical properties (e.g., independence) of the learned representation, rather than causal recovery of the true generative factors. The corresponding $\beta$-VAE and factor-VAE disentanglement metrics are post-hoc evaluations applied after model training.

$c_2$ measures LD utilization and penalizes solutions in which most signal is concentrated in only one or a few LDs. Together, $c_1$ and $c_2$ ensure that LSSI reflects not only separation among LD-feature profiles, but also sufficient feature coverage and non-degenerate use of the latent space.

In the case of $\|A_{[k,]}\|_1 = 0$ and $\|A_{[j,]}\|_1 = 0$ for a pair of $(k, j)$ or when there exists a row $i$ such that $A_{k,i} \geq A_{j,i}$ for all $i$ and $j \neq k$, then $\mathrm{LSSI}(A) = 0$ and $\mathrm{LSSI}_0(A) = 0$ (illustrated in Fig. 3). Second,

Figure 3: Illustration of two degenerate cases resulting in LSSI = 0. Left: All entries of matrix $A$ are 0, indicating the complete absence of informative LDs. Right: A single LD dominates all other LDs across all input features in $A$, resulting in no disentanglement.

LSSI is conveniently bounded in $[0, 1]$, which facilitates interpretation, model comparison, and hyperparameter tuning. A value of 1 corresponds to a perfectly separated latent space, where $A$ has a block-separable structure: each informative LD is associated with a distinct, non-overlapping subset of input features while non-informative LDs have negligible feature associations. A value of 0 corresponds to a degenerate latent space with no meaningful separation among LD-feature association profiles, such as when a single dominant LD is associated with most features or multiple LDs encode highly overlapping information. Intermediate values quantify the continuum between these endpoints: larger values indicate stronger structural separation, whereas smaller values indicate more overlapping or entangled feature encoding or one or a very few LDs are highly dominant among all informative LDs.

## 4 Experiments

We benchmark the proposed bfVAE in its disentanglement efficiency against the vanilla VAE and four existing disentangled VAE models, and assess the robustness of FVH-LT, DBSR-LS, and LSSI in quantifying the disentanglement effect. We use image and tabular data (Tab. 1) with a particular emphasis on tabular data. Image data is included as it is the common benchmark data type for VAE disentanglement assessment. We benchmarks LSSI against two representative existing disentanglement metrics in Higgins et al. (2017) and Kim & Mnih (2018).[6] The two synthetic datasets vary in $p$ and $n$. For the five real datasets, RNA-seq (Fiorini, 2016) contains high-dimensional cancer gene-expression profiles across multiple tumor types; white wine (Cortez et al., 2009) includes features describing white wine characteristics with a label on the quality score; FIFA 2018 (Mathan, 2018) contains various soccer match statistics; and MNIST (Deng, 2012) and CelebA (Liu et al., 2015) are widely used benchmark image data.

Table 1: Datasets used in the experiments

| Dataset | Training $n$; test $n$ | No. of Features $p$ | Known Ground Truth? |
|---|---|---|---|
| FA15 (tabular)[†] | 800/200 | 15 | Yes ($K_0 = 4$ factors) |
| FA100 (tabular)[†] | 40,000/10,000 | 100 | Yes ($K_0 = 6$ factors) |
| White wine (tabular)[#] | 3,918/980 | 12 | No |
| FIFA 2018 (tabular)[#] | 102/26 (51/13 matches) | 16 | No |
| RNA-seq (tabular)[#] | 640/161 | 20,264 | No |
| MNIST (image)[#] | 50,000/10,000 | 784 ($28 \times 28$) | No |
| CelebA (image)[#] | 35,000/5,000 | $12,288$ ($64 \times 64 \times 3$) | No |

[†] We simulated the FA15 and FA100 from factor analysis (FA) models.

[#] Publicly available at Kaggle, PyTorch, or UC Irvine Machine Learning Repository.

In each experiment, the dataset is split into a training set and a test set: the training set is used to train VAE and perform FVH-LT and DBSR-LS and the test set evaluates model fitting and convergence of VAE

---

[6]Other disentanglement metrics based on complementary criteria exist (Sec 1.3), and they also require known ground-truth factors. Because these alternatives introduce additional implementation choices and are not universally regarded as superior across settings, we consider the metrics in Higgins et al. (2017) and Kim & Mnih (2018) sufficient as representative benchmarks.

training. $R$ was set at 10 in all experiments except for CelebA, where $R = 5$ due to computational constraints. More details on model configurations and algorithmic hyperparameters are provided in App. B; We provide a subset of the experimental results in the main text and list the comprehensive results in App. G.

The main findings are as follows. 1) Across synthetic tabular, real tabular, and image datasets, bfVAE provides the most favorable overall trade-off between disentanglement and reconstruction among the benchmark VAE frameworks in the studied experimental settings. 2) In the synthetic experiments with known ground-truth factors, the trends of LSSI across different models agrees well with existing supervised disentanglement benchmarks, supporting its effectiveness as a metric for latent structural separation without ground-truth factor knowledge. 3) Both FVH-LT and DBSR-LS can uncover semantically meaningful or domain-aligned latent structures and generally yield consistent conclusions. FVH-LT is computationally more efficient at large $p$ [7] and often gives clearer informative-LD signals (LDBS-LS can be faster at small $p$). 4) Overall, bfVAE with FVH-LT and LSSI offers a practical and interpretable framework for learning separated latent representations without sacrificing reconstruction quality.

### 4.1 Tabular data with knowledge on true generative factors

Fig. 4 the results in the disentanglement effect via bfVAE assessed by FVH-LT in FA15 and FA100. bfVAE effectively disentangles the latent structure, with the learned associations between informative LDs and input features closely aligned with the ground truth. It is worth noting the robustness of FVH-LT to over-specified $K$; that is, it does not mis-identify non-informative LDs as informative, resulting in a low false positive rate.

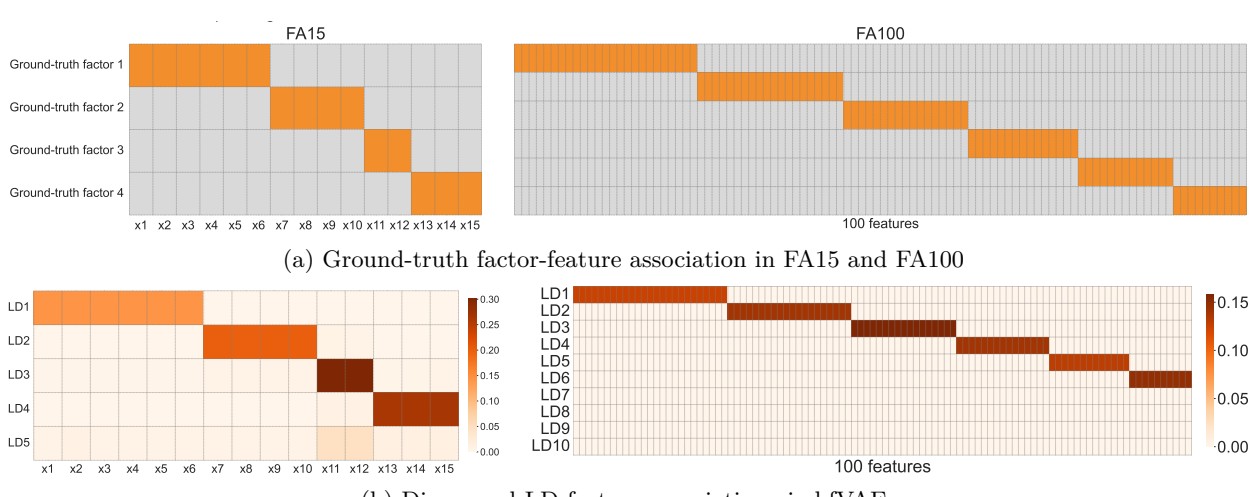

(a) Ground-truth factor-feature association in FA15 and FA100

(b) Discovered LD-feature association via bfVAE

Figure 4: Ground-truth factor-feature association and discovered LD-feature association via bfVAE assessed by FVH-LT in FA15 and FA100. Darker cells indicate stronger associations between LDs and input features. Values in the LD-feature FVH-LT matrix are not shown due to readability reasons. DBSR-LS results for bfVAE and FVH-LT and DBSR-LS in the VAE benchmarks are in Apps. D.1- D.4.

The LSSI, reconstruction $\ell_2$ error on the test sets, and the estimated number of generative factors are reported in Tab. 2 for bfVAE and the benchmark VAEs. A further break-down of the LSSI by $c_1, c_2$ and the raw LSSI score in Eq. 5 is provided in Appendix. E. Overall, bfVAE achieves the best trade-off: it ranks first or second in LSSI across datasets and methods, while delivering substantially lower reconstruction error than all competing models; and recovers the true number of generative factors. Among the benchmarks, Factor-VAE achieves competitive LSSI scores – often second-best – but at the cost of poor reconstruction fidelity and under-discover generative factors 50% of the time. Conversely, $\beta$-VAE reconstructs well but scores poorly on LSSI and over-identities generative factors 50% of the time. Vanilla VAE and the two DIP-VAE variants perform the worst on both disentanglement and reconstruction.

---

[7] In the CelebA dataset ($p = 12, 288$) and RNA-seq ($p = 20, 264$), the high computational costs for DBSR-LS actually exceed our resources. Due to space limitations and for the reasons listed above, we focus on the FVH-LT results in the main text; the DBSR-LS results can be found in Appendix.

Table 2: LSSI, reconstruction error, and uncovered informative LD counts in synthetic FA15 and FA100.

| Method | Data | Metric | bfVAE | factor-VAE | $\beta$-VAE | Vanilla VAE | DIP-VAE-I | DIP-VAE-II |
|---|---|---|---|---|---|---|---|---|
| FVH-LT | FA15 | $\uparrow$ LSSI | **0.891** | *0.807* | 0.366 | 0.544 | 0.633 | 0.498 |
| | | $(\to 0)\, K_{\max}-K_0$[†] | **0** | -1 | **0** | **0** | **0** | **0** |
| | FA100 | $\uparrow$ LSSI | 0.913 | **0.940** | 0.252 | 0.866 | 0.874 | *0.929* |
| | | $(\to 0)\, K_{\max}-K_0$[†] | **0** | **0** | 3 | **0** | **0** | **0** |
| DBSR-LS | FA15 | $\uparrow$ LSSI | *0.979* | **0.987** | 0.906 | 0.597 | 0.962 | 0.841 |
| | | $(\to 0)\, K_{\max}-K_0$[†] | **0** | -1 | **0** | 0 | **0** | -1 |
| | FA100 | $\uparrow$ LSSI | *0.636* | 0.602 | 0.217 | 0.591 | 0.592 | **0.660** |
| | | $(\to 0)\, K_{\max}-K_0$[†] | **0** | **0** | 3 | **0** | **0** | **0** |
| FA15 | | $\downarrow$ recon. $\ell_2$[‡] | *0.75 ± 0.29* | 2.13 ± 0.17 | **0.64 ± 0.03** | 2.18 ± 0.18 | 1.70 ± 0.06 | 2.36 ± 0.26 |
| FA100 | | $\downarrow$ recon. $\ell_2$[‡] | **0.39 ± 0.01** | 1.28 ± 0.03 | *0.56 ± 0.02* | 1.35 ± 0.08 | 1.49 ± 0.06 | 1.38 ± 0.06 |

[†]: The closer to 0, the better; true factor counts $K_0 = 4$ in FA15 and $K_0 = 6$ in FA100

[‡] Per-sample reconstruction $\ell_2$ error on the test set: mean ± standard deviation across 10 runs.

**Bold** denotes the best result (highest LSSI; lowest $\ell_2$ error) by dataset and method; ***italic bold*** denotes the second best.

Since FA15 and FA100 are synthetic data, we can benchmark our proposed LSSI and $LSSI_0$ per Eq. 7 against the existing disentanglement metrics in the literatureHiggins et al. (2017); Kim & Mnih (2018), both of which require known generative factors). The results are provided in Tab. 3. First, regardless of which disentanglement metric is used, bfVAE consistently achieves the highest scores, confirming it as the best-performing disentangled VAE framework. Second, we observe a consistent monotonic relationship between LSSI (Tab. 2), $LSSI_0$ and the two existing disentanglement metrics: higher LSSI and $LSSI_0$ correspond to higher scores on both benchmarks, validating LSSI as an effective disentanglement metric. Third, the similarity between LSSI and $LSSI_0$ suggests that $LSSI_0$ is robust to over-specification of $K$.1 and show smaller differences across methods.

Table 3: $LSSI_0$ and baseline disentanglement scores in synthetic FA15 and FA100

| Data | Metric | bfVAE | factor-VAE | $\beta$-VAE | Vanilla VAE | DIP-VAE-I | DIP-VAE-II |
|---|---|---|---|---|---|---|---|
| FA15 | $LSSI_0$ FVH-LT | **0.963** | 0.811 | 0.405 | 0.703 | 0.661 | 0.780 |
| | $LSSI_0$ DBSR-LS | **1.000** | 0.999 | 0.972 | 0.851 | 0.995 | 0.578 |
| | Higgins et al. (2017) [‡] | **1.00 ± 0.00** | 0.99 ± 0.01 | 0.90 ± 0.07 | 0.96 ± 0.06 | 0.84 ± 0.11 | 0.94 ± 0.07 |
| | Kim & Mnih (2018) [‡] | **0.99 ± 0.03** | 0.90 ± 0.06 | 0.89 ± 0.09 | 0.84 ± 0.07 | 0.84 ± 0.13 | 0.81 ± 0.06 |
| FA100 | $LSSI_0$ FVH-LT | **0.990** | 0.986 | 0.417 | 0.860 | 0.932 | 0.962 |
| | $LSSI_0$ DBSR-LS | **0.766** | 0.743 | 0.351 | 0.702 | 0.681 | 0.738 |
| | Higgins et al. (2017) [‡] | **1.00 ± 0.00** | 1.00 ± 0.00 | 0.64 ± 0.08 | 0.99 ± 0.03 | 1.00 ± 0.00 | 1.00 ± 0.00 |
| | Kim & Mnih (2018) [‡] | **1.00 ± 0.00** | 1.00 ± 0.00 | 0.77 ± 0.13 | 0.99 ± 0.03 | 1.00 ± 0.00 | 1.00 ± 0.00 |

**Bold** indicates the VAE model with the best score within each dataset and metric.

[‡] Scores are reported as mean ± SD over 10 runs. LSSI and $LSSI_0$ are based on the averaged FVH-LT and DBSR-LS over 10 runs, per definition

## 4.2 Tabular data without knowledge on generative factors

Fig. 5 displays the FVH-LT results from bfVAE on the RNA-seq, white wine, the 2018 FIFA datasets. For RNA-seq, the top $3,000$ unique genes are selected fromthe full LD-feature association matrix of $15 \times 20,264$ for better visualization. Tab. 4 reports the LSSI, reconstruction $\ell_2$ error on the test sets, and the estimated number of generative factors via bfVAE and the benchmark VAEs. A further break-down of the LSSI by $c_1, c_2$ and the raw LSSI score in Eq. 5 is provided in Apps. E.3- E.5. The findings are similar to Tab. 2: bfVAE achieves the best trade-off between disentanglement and reconstruction. Specifically, bfVAE ranks mostly the best in LSSI across datasets and methods with the second best reconstruction error on white wine and FIFA 2018 whereas $\beta$-VAE achieves the best errors in both; for RNA-seq, the reconstruction errors are about the same across all VAEs models with $\sim 1\%$ differences. The second best LSSI score varies are shared by factor-VAE and DIP-VAE-I where bfVAE is the best; $\beta$-VAE achieves the best LSSI for white wine with FVH-LT whereas bfVAE is the second best.

Compared with Tab. 2, the LSSI scores in Tab. 4 are lower than those observed in synthetic settings, which is expected because real-world datasets are typically more complex, noisy, and heterogeneous, and their underlying generative factors may not exhibit a clean block-separable structure. For RNA-seq, the data is very

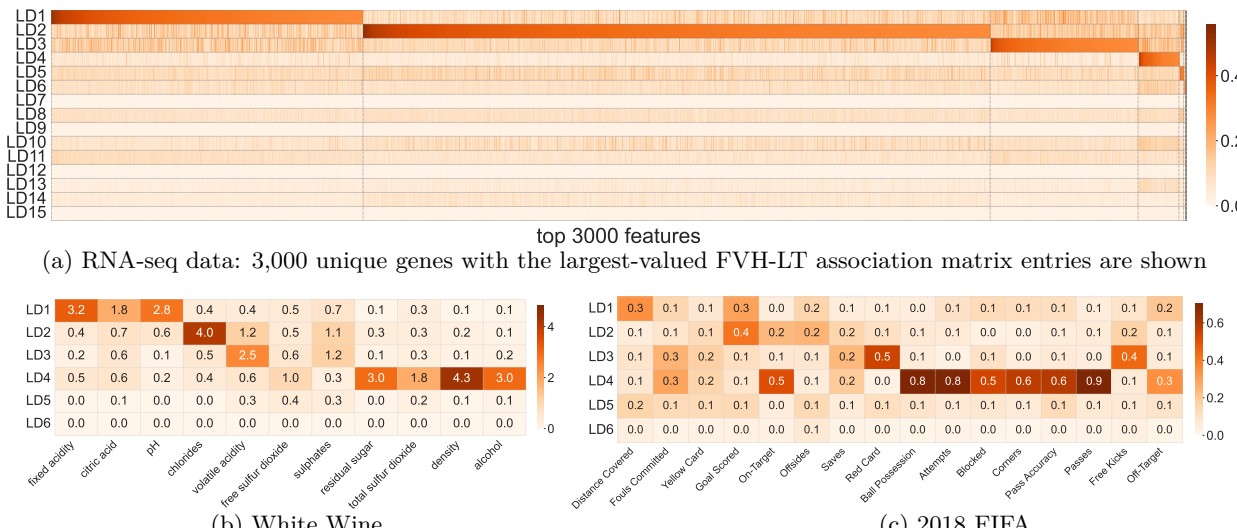

(a) RNA-seq data: 3,000 unique genes with the largest-valued FVH-LT association matrix entries are shown

(b) White Wine

(c) 2018 FIFA

Figure 5: Disentanglement effects of bfVAE assessed by FVH-LT in datasets without knowledge of ground-truth generative factors. Darker cells indicate stronger associations between LD and input features. DBSR-LS results for bfVAE and FVH-LT and DBSR-LS results for the benchmark VAE models are in Apps. D.5- D.9.

Table 4: LSSI, reconstruction error, and uncovered informative LD counts in real tabular data.

| Method | Data | Metric | **bfVAE** | factor-VAE | $\beta$-VAE | Vanilla VAE | DIP-VAE-I | DIP-VAE-II |
|--------|------|--------|-----------|------------|-------------|-------------|-----------|------------|
| FVH-LT | RNA-seq | ↑ LSSI | **0.081** | 0.057 | 0.056 | 0.052 | *0.072* | 0.058 |
| | | $K_{\max}$ | 11 | 14 | 12 | 12 | 13 | 14 |
| | White wine | ↑ LSSI | **0.390** | *0.308* | 0.175 | 0.220 | 0.090 | 0.280 |
| | | $K_{\max}$ | 5 | 4 | 5 | 5 | 4 | 4 |
| | FIFA 2018 | ↑ LSSI | *0.132* | 0.020 | **0.197** | 0.031 | 0.104 | 0 |
| | | $K_{\max}$ | 5 | 2 | 5 | 2 | 5 | 3 |
| DBSR-LS | White wine | ↑ LSSI | **0.736** | 0.478 | 0.466 | 0.356 | *0.719* | 0.396 |
| | | $K_{\max}$ | 5 | 4 | 5 | 5 | 4 | 4 |
| | FIFA 2018 | ↑ LSSI | **0.455** | 0.127 | 0.404 | 0.081 | *0.450* | 0.040 |
| | | $K_{\max}$ | 5 | 2 | 5 | 2 | 5 | 3 |
| | RNA-seq | | $105.51 \pm 0.26$ | $104.60 \pm 0.15$ | $104.91 \pm 0.22$ | $105.15 \pm 0.21$ | $104.11 \pm 0.17$ | $104.22 \pm 0.18$ |
| | white wine | ↓ recon. $\ell_2$[‡] | *2.03 ± 0.04* | $2.37 \pm 0.06$ | **1.52 ± 0.03** | $2.35 \pm 0.06$ | $2.08 \pm 0.02$ | $2.33 \pm 0.04$ |
| | FIFA 2018 | | *2.84 ± 0.14* | $3.49 \pm 0.03$ | **2.75 ± 0.08** | $3.53 \pm 0.04$ | $3.28 \pm 0.09$ | $3.53 \pm 0.04$ |

**Bold** denotes the highest LSSI score by dataset and method; ***bold italic*** denotes the second best.

[‡] Per-sample reconstruction $\ell_2$ error on the test set: mean $\pm$ standard deviation across 10 runs.

high-dimensional, genes are co-regulated in modules and pathway programs may overlap, and a single gene may participate in multiple biological processes. Therefore, the latent structure may involve overlapping gene sets rather than clean, disjoint feature blocks, providing a biological explanation for the low LSSI values. For FIFA 2018, FactorVAE, vanilla VAE, and DIP-VAE-I exhibit a dominant-LD pattern in FVH-LT and DBSR-LS, where most features associate with one LD while others remain largely non-informative (Apps. D.8–D.9). DIP-VAE-II shows an even more degenerate dominant-row pattern, yielding a zero-valued LSSI. Overall, bf-VAE generally achieves clearer latent-feature association patterns and stronger interpretability on real-world tabular datasets and especially useful for relative model comparison, benchmarking, and hyperparameter tuning is latent structural separation is the main goal. As LSSI is applied to a broader range of real-world datasets, empirical reference ranges can be developed to guide the interpretation of what constitutes strong or moderate latent-space separation in practice.

The outputs from FVH-LT and DBSR-LS can be combined with domain knowledge to yield the highest utility. For RNA-seq, prior pan-cancer studies show that gene-expression variation is strongly organized by tissue/cell-of-origin and by pathway-level programs (Hoadley et al., 2018; Peng et al., 2015; Sanchez-Vega et al., 2018; Thorsson et al., 2018). Thus the uncovered latent structure can be interpreted in light of the known biology of pan-cancer transcriptomes to make the best of the result. For white wine, LD1 is associated with fixed acidity, citric acid, and pH, reflecting the wine acidity profile; LD2 is associated with chlorides; LD3 with volatile acidity; LD4 is strongly associated with density, alcohol, residual sugar, and

total sulfur dioxide, which are characteristics related to wine flavor. LD5 and LD6 is non-informative. In the 2018 FIFA data, LD1 is strongly associated with Distance Covered, Goal Scored, and Off-Target – reflecting match intensity and attacking activity. LD2 encodes Goal Scored, with weaker associations to On-Target, Offsides, Saves, and Free Kicks, representing scoring-related attacking outcomes.LD3 is associated with Fouls Committed, Red Card, Free Kicks, and Ball Possession, reflecting game control. LD4 shows structure with strong associations to On-Target, Ball Possession, Attempts, Blocked, Corners, Pass Accuracy, and Passes, corresponding to attacking buildup and shot creation.LD5 and LD6 are noninformative.

We also applied FVH-LT in the bf-CVAE model on the white wine dataset given that it contains label $Y$ (wine quality). The implementation of bf-CVAE and full FVH-LT results are provided in App. H. Tab. 5 shows the last row of the LD-feature association matrix with LT on the conditioning variable $Y$. VC, C, D, and A exhibit the highest variance, indicating strong relations with wine quality and consistency with domain knowledge (see App. C). These results suggest a potential use case for conditional generation: a wine manufacture could fix $Y$ at a high-quality score, sample informative LDs that exhibits strong associations between these four features, and feed the samples to the decoder to generate candidate feature profiles for wines targeting that quality level.

Table 5: Variance of reconstructed features with LT on wine quality score via FVH-LT.

| fixed acidity | **volatile acidity** (VC) | citric acid | residual sugar | **chlorides** (C) | free SO$_2$ |
|---|---|---|---|---|---|
| 0.08 | **0.17** | 0.08 | 0.06 | **0.14** | 0.05 |

| total SO$_2$ | **density** (D) | pH | sulphates | **alcohol** (A) | **quality** |
|---|---|---|---|---|---|
| 0.09 | **0.25** | 0.07 | 0.05 | **0.54** | **2.9** |

### 4.3 Image Data CelebA and MNIST

The FVH-LT results in CelebA and MNIST are visualized in Figs. 6 and 7, respectively. CelebA contains three color channels (red, green, and blue). In the FVH-LT analysis, heatmaps were first computed separately by channel, then globally normalized and merged to produce a composite visualization in Fig. 6(a). The LT of a representative image in Figs. 6(b) and 7(b) illustrates how informative LDs correspond to meaningful pixels in the images. The results clearly demonstrate the specific factors each dimension represents, highlighting the disentangled structure captured in the latent space.

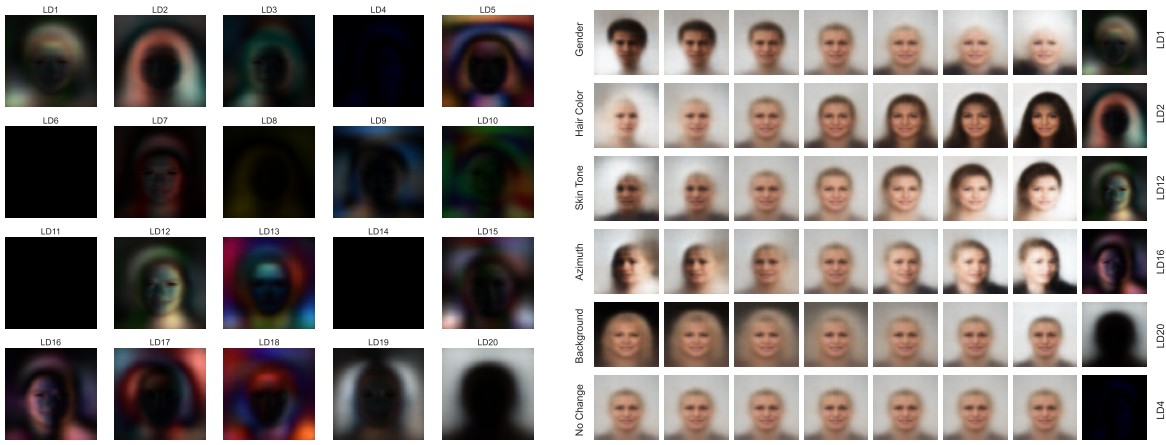

(a) FVH-LT results averaged over 1000 images and 5 runs. Brighter regions indicate higher variance, highlighting image areas that are sensitive to LT.

(b) Example images in Columns 1 to 7 reconstructed from LT on 6 LDs in a single run; the corresponding FVH-LT heatmaps are in Column 8.

Figure 6: Disentanglement effects of bfVAE on CelebA assessed by FVH-LT. FVH-LT shows which facial features correspond to what informative LD: LT of non-informative LD4 produces no visible change in the reconstruction and a completely dark heatmap in (a) and (b); in (b), LD1, 2, 12, 16, and 20 primary encode sex, hair style and color, skin tone and illumination, and azimuth (horizontal head orientation), background information, respectively. FVH-LT results for the benchmark VAE models are in Apps. D.11.

Tab. 6 reports the LSSI, reconstruction $l_2$ error and $K_{\max}$. bfVAE achieves the highest LSSI score on both datasets and the lowest reconstruction error on MNIST. Although its reconstruction error is not the smallest in CelebA, it remains comparable to the benchmark VAEs with reconstruction errors and is substantially

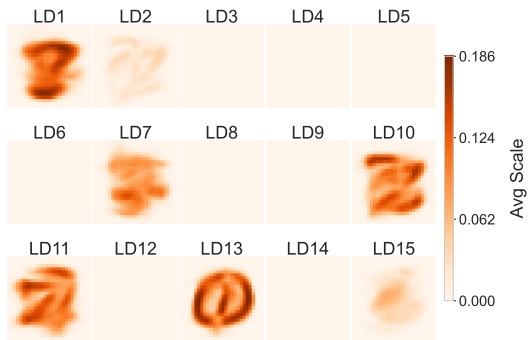
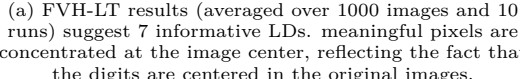
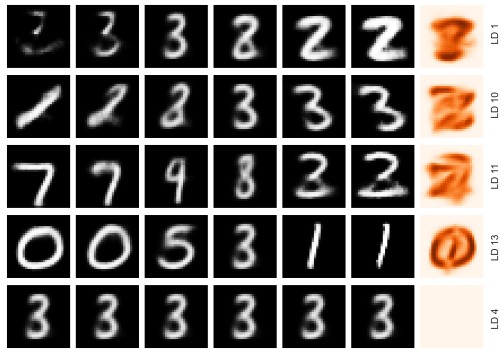

(a) FVH-LT results (averaged over 1000 images and 10 runs) suggest 7 informative LDs. meaningful pixels are concentrated at the image center, reflecting the fact that the digits are centered in the original images.

(b) Example LT results for an image with digit 3: Columns 1 to 6 are reconstructed from LT on 5 LDs in a single run; Column 7 shows the corresponding FVH-LT heatmaps.

Figure 7: Disentanglement effects of bfVAE in MNIST assessed by FVH-LT. (a) and (b) show that LD1, LD10, LD11, and LD13 capture variation between digits 3 and 2, 1 and 3, 7 and 3, and 0 and 1 respectively. Key structural elements that differentiate the digits are evident such as the loops and center vertical line in LD13 and the curved regions near top-left and bottom-right in LD10. The 7 informative LDs produce obvious changes in reconstructed images under LT (first four rows in (b)) with interpretable semantic patterns. For example, LT of LD1 generates digits 2, 3, 8 and the morphing process is reflected in the last column of (b) that suggests a blend of 2,3,8. LD10 also captures directional variations in handwriting. Along the LT, the generated digit changes from a slanted 1 into a 3 which gradually changes its slant and curvature. FVH-LT results for the benchmark VAE models are in Apps. D.10.

lower than that of $\beta$-VAE. Across both datasets, bfVAE shows consistent performance and provides a favorable trade-off between latent-space separation and reconstruction accuracy. In contrast, the performance of the other methods appears more dataset-dependent.

Table 6: LSSI, reconstruction error, and uncovered informative LD counts in CelebA and MNIST.

| Data | Metric | **bfVAE** | factor-VAE | $\beta$-VAE | Vanilla VAE | DIP-VAE-I | DIP-VAE-II |
|---|---|---|---|---|---|---|---|
| CelebA | ↑ LSSI | **0.116** | *0.052* | 0 | 0.029 | 0.031 | 0.029 |
| | ↓ recon. $\ell_2$‡ | $11.07 \pm 0.23$ | $10.36 \pm 0.03$ | $19.58 \pm 0.10$ | $10.16 \pm 0.03$ | $10.21 \pm 0.03$ | $10.21 \pm 0.05$ |
| | $K_{max}$ | 14 | 8 | 1 | 10 | 10 | 10 |
| MNIST | ↑ LSSI | **0.184** | 0.089 | 0.053 | 0.078 | 0.088 | *0.095* |
| | ↓ recon. $\ell_2$‡ | $3.10 \pm 0.11$ | $3.46 \pm 0.02$ | $5.11 \pm 0.00$ | $3.28 \pm 0.01$ | $3.30 \pm 0.02$ | $3.33 \pm 0.02$ |
| | $K_{max}$ | 7 | 9 | 14 | 10 | 8 | 9 |

‡ Per-sample reconstruction $\ell_2$ error on the test set: mean $\pm$ standard deviation across runs.
**Bold** denotes the highest LSSI score within each dataset; ***bold italic*** denotes the second best.

### 4.4 Runtime Result for Model Training and Disentanglement Evaluation

Tab. 7 summarizes the approximate computational time per run for VAE training, FVH-LT, and DBSR-LS. Since the experiments were conducted on several platforms, the time should not be compared across datasets that used different platforms.

Table 7: Approximate single-run time for VAE model training, FVH-LT, and DBSR-LS.

| | FA15 | FIFA 2018 | White wine | CVAE White wine | FA100 | RNA-seq | MNIST | CelebA |
|---|---|---|---|---|---|---|---|---|
| $n$‡ | 800 | 102 | 3,918 | 3,918 | 40,000 | 640 | 50,000† | 35,000 |
| Epochs | 150 | 350 | 150 | 75 | 150 | 100 | 35 | 50 |
| model training time (min) | 3 (M2) | 1 (M2) | 10 (M2) | 20 (M2) | 4 (G4) | 1 (G4) | 2 (A100) | 4 (A100) |
| FVH-LT† time (min) | 2 (M2) | <1 (M2) | 15 (M2) | 16 (M2) | 17 (G4) | 10 (G4) | 14 (A100) | 60 (A100) |
| DBSR-LS time (min) | 1 (M2) | <1 (M2) | 1 (M2) | – | 2 (G4) | – | 40 (A100) | – |

‡ $n$ for model training, FVH-LT, and DBSR-LS are the same in all datasets except for MNIST and CelebA, which is $n$ listed is for model training and $n = 1000$ for FVH-LT and DBSR-LS.
† The number of LT steps was $L = 500$ RNA-seq and $L = 1,000$ for the rest.
M2: a local machine with an Apple M2 Max CPU and 96 GB RAM; G4: a Google Colab G4 GPU; A100: an A100 GPU node with two NVIDIA A100 80GB PCIe GPUs and 64 Intel Xeon Platinum 8358 CPU cores at 2.60GHz.
Model training time does not include cross-validation or hyperparameter tuning. FVH-LT and DBSR-LS times are reported for one evaluation run. GAS in all datasets took < 1 minute per run.

### 4.5 Visualizing Posterior Distributions of Informative and Non-informative LDs

Algs. 1 and 2 calculate the posterior-prior KL divergence for each LD as an intermediate product, on which informative vs non-informative LDs in Def. 1 are based. To illustrate how the posterior deviate from the prior $\mathcal{N}(\mathbf{0}, I)$, we plot the posterior distributions from a single run of bfVAE with $\gamma = 0$ and $K = 15$ on a random subset of 1,000 MNIST training images in Fig. 8. The results clearly suggest there are seven informative LDs, the posteriors of which exhibit low variances ($< 0.1$) and many also have posterior means shifted from 0, leading to a prior-posterior KL divergence of $\sim 8$ on average. The rest eight LDs are non-informative with an KL divergence of $\sim 3 \times 10^{-3}$ on average. Similar patterns are consistently observed in other runs.

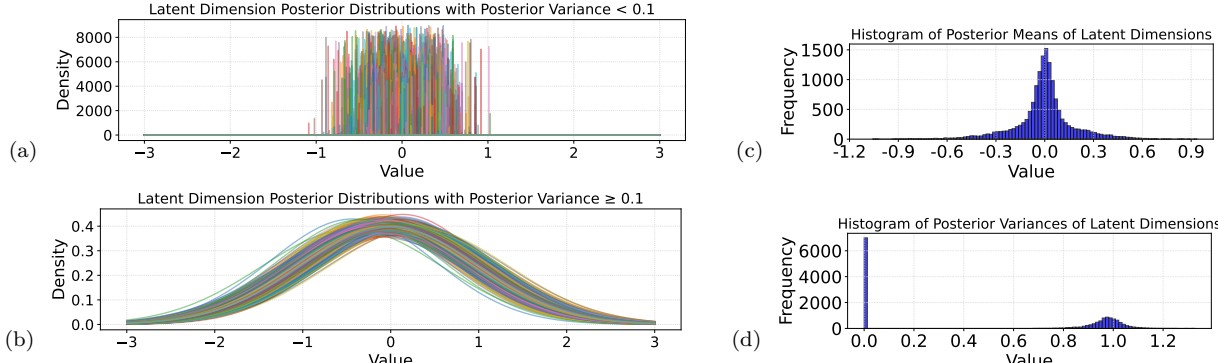

Figure 8: An example of posterior statistics across LDs on MNIST (15 LDs $\times$ 1000 MNIST images): (a) posterior distributions of informative LDs, (b) posterior distributions of non-informative LDs, (c) histogram of posterior means, (d) histogram of posterior variances (among the 15,000 LDs, exactly 7,000 display near-zero variances in their distribution and correspondingly exhibit large KL divergence).

## 5 Conclusion and Discussion

We proposed bfVAE as a unified framework for disentangling VAE latent space and offered an information-bottleneck perspective on its formulation. We introduced two ground-truth-free evaluation techniques, FVH-LT and DBSR-LS, to support interpreting and quantifying LD-feature associations, along with GAS to address label switching in latent space from multiple VAE runs. We further developed LSSI, a scalar metric derived from the LD-feature association matrices produced by FVH-LT and DBSR-LS to summarize overall latent structural separation. Importantly, FVH-LT, DBSR-LS, GAS or LSSI do not require knowledge of ground-truth generative factor. Across extensive experiments, bfVAE provides an effective framework for promoting latent-space disentanglement and generally achieves more favorable reconstruction-disentanglement trade-off relative to benchmark VAEs, as assessed by FVH-LT, DBSR-LS, LSSI, and reconstruction error.

While both FVH-LT and DBSR-LS effectively uncover can semantically meaningful, domain-aligned latent structures, we recommend FVH-LT as the default technique from both computational and methodological perspectives. DBSR-LS can be computational intensive given its sparse regression framework especially when $p$ is large, and methodological, it relies on a linear surrogate to characterize the LD-feature relation for straightforward interpretability. Because DBSR-LS estimates LD-feature associations directly from learned latent space and is not decoder-mediated like FVH-LT, it can serve as a useful alternative when decoder-based LT is unstable, especially when $p$ is not large.

Future directions include applying our procedures to other data types (e.g., graphs), adapting the procedures to other deep learning models that may benefit from interpretable latent spaces disentanglement, and extending DBSR-LS to non-linear regression settings. A focused assessment of alternative alignment strategies to the greedy strategy proposed in this paper, such as Hungarian matching and optimal transport, would make another interesting direction for future work.

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

## Supplementary Materials to "A Unified Latent Space disentangled VAE Framework with Robust Disentanglement Effectiveness Evaluation"

## A  DBSR-LS Flowchart

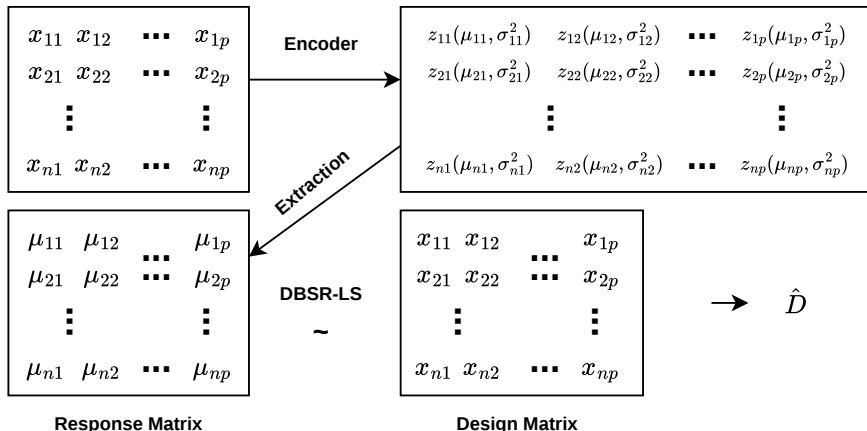

Figure 9: Conceptual illustration of DBSR-LS. Posterior means $\boldsymbol{\mu}_Z$ in the latent space are multi-task regression responses; $\mathbf{x}$ are predictors; sparse regression coefficient matrix $\hat{D}$ summarizes latent-feature association.

## B  Simulation Models for Synthetic Data Generation and Experimental Settings

### B.1  Simulated tabular data with ground-truth knowledge

For the synthetic tabular experiments, we used a fully connected VAE architecture with a symmetric encoder-decoder design, trained using the loss in Eq. (1). The encoder consists of three hidden layers with dimensions 256, 128, and 64, each followed by a ReLU activation and dropout with a rate of 0.2. The encoder outputs are passed through two parallel linear layers to produce the mean and log-variance for the latent space. The decoder mirrors the encoder, with hidden layers of size 64, 128, and 256, followed by ReLU and dropout activations, and concludes with a linear output layer mapping to the input dimension. Samples from the latent space were obtained using the reparameterization trick (Kingma & Welling, 2013). A discriminator network is used to estimate the Total Correlation (TC) of the aggregated posterior (Kim & Mnih, 2018). The discriminator consists of five fully connected layers with 1000 units each, interleaved with LeakyReLU activations (negative slope = 0.2), and ends with a linear layer outputting two logits that distinguish between real and permuted latent.

For data FA15, we used $\beta = 0.05$, $\gamma = 5$, and $C = 15$ for both FVH-LT and DBSR-LS. For DBSR-LS, the regularization coefficients were set to $\lambda_B = 0.1$ and $\lambda_S = 0.1$.

For data FA24, we used $\beta = 5 \times 10^{-2}$ and $\gamma = 6$ for both FVH-LT and DBSR-LS. The regularization parameters for DBSR-LS were set to $\lambda_B = 0.2$ and $\lambda_S = 0.2$. Due to similarity with FA15, the FA24 results are provided in the Supplementary Materials only.

For data FA100, we used $\beta = 0.3$, $\gamma = 3$, and $C = 20$ for both FVH-LT and DBSR-LS. For DBSR-LS, the regularization coefficients were set to $\lambda_B = 0.3$ and $\lambda_S = 0.3$.

For all datasets, models were trained with a batch size of 16 using the Adam optimizer with a learning rate of $1 \times 10^{-4}$. The correlation threshold for alignment was set to $\rho_{\min} = 0.5$. Latent traversal was centered at the posterior mean and used a half-width of 15 posterior standard deviations, and each experiment was repeated $R = 10$ times with identical configurations to ensure stability and enable consistent latent alignment.

### B.2   Real-world tabular data without ground-truth knowledge

For the real-world data experiments, we used a fully connected VAE architecture with a symmetric encoder-decoder design, trained using the objective in Eq.(1). For RNA-seq, the encoder consists of three hidden layers with dimensions 512, 256, and 128, each followed by a ReLU activation and dropout with rate 0.2. For the White wine and 2018 FIFA statistics datasets, the encoder uses hidden layers with dimensions 256, 128, and 64, each followed by a ReLU activation and dropout with rate 0.1. The encoder outputs are passed through two parallel linear layers to produce the mean and log-variance of a 6-dimensional latent space. The decoder mirrors the encoder, followed by ReLU and dropout activations, and concludes with a linear output layer mapping to the input dimension. Samples from the latent space were obtained using the reparameterization trick. A discriminator network is used to estimate the Total Correlation (TC) of the aggregated posterior. The discriminator consists of five fully connected layers with 1000 units each, interleaved with LeakyReLU activations (negative slope = 0.2), and ends with a linear layer outputting two logits that distinguish between real and permuted latent.

For RNA-seq data, we used $\beta = 15$, $\gamma = 10$, and $C = 20$ for FVH-LT. The model was trained with a batch size of 16 using the Adam optimizer with a learning rate of $1 \times 10^{-4}$. The correlation threshold for alignment was set to $\rho_{\min} = 0.5$.

For white wine data, we used $\beta = 0.7$, $\gamma = 1$, and $C = 2$ for both FVH-LT and DBSR-LS. The DBSR-LS regularization coefficients were set to $\lambda_B = 1 \times 10^{-2}$ and $\lambda_S = 1 \times 10^{-2}$. Models was trained with a batch size of 32 using the Adam optimizer with a learning rate of $1 \times 10^{-4}$. The correlation threshold for alignment was set to $\rho_{\min} = 0.5$.

For 2018 FIFA statistics data, we used $\beta = 0.07$, $\gamma = 2$, and $C = 10$ for both FVH-LT and DBSR-LS. The regularization coefficients for DBSR-LS were also set to $\lambda_B = 1 \times 10^{-2}$ and $\lambda_S = 1 \times 10^{-2}$. Model was trained with a batch size of 8 using the Adam optimizer with a learning rate of $1 \times 10^{-4}$. The correlation threshold for alignment was set to $\rho_{\min} = 0.5$.

LT was centered at the posterior mean and performed over $[\mu - 5, \mu + 5]$ for each latent dimension, and each experiment was repeated $R = 10$ times with identical configurations to ensure stability and enable consistent latent alignment.

For the CVAE applications, we used a fully connected VAE architecture with a mirrored encoder-decoder design and objective function in Eq.(1). The encoder consists of three hidden layers with dimensions 256, 128, and 64, each followed by a ReLU activation and dropout with a rate of 0.2. The encoder outputs are passed through two parallel linear layers to produce the mean and log-variance of a 6-dimensional latent space. The decoder contains three hidden layers of size 64, 128, and 256, followed by ReLU and dropout activations, and concludes with a linear output layer mapping to the input dimension. Samples from the latent space were obtained using the reparameterization trick.

We used $\beta = 0.1$ and $\gamma = 3$. Model was trained with a batch size of 16 using the Adam optimizer with a learning rate of $1 \times 10^{-4}$. LT was conducted over the quality latent using a data-driven range of $[-3.2, 3.5]$, while the remaining latent dimensions were traversed over a fixed interval of $[-15, 15]$. To ensure stability and enable consistent alignment of latent dimensions, each experiment was repeated $R = 10$ times under identical configurations.

### B.3   MNIST

We employed VAE architecture in the Pythae (Chadebec et al., 2022) with a disentangled $\beta$-VAE loss function. Specifically, we used a ResNet-based encoder-decoder design tailored for image inputs. The encoder consists of four sequential blocks: (1) a convolutional layer with 64 output channels, 4×4 kernel, stride 2, and padding 1; (2) a convolutional layer with 128 output channels, 4×4 kernel, stride 2, padding 1; (3) a convolutional layer with 128 output channels, 3×3 kernel, stride 2, padding 1; (4) two residual blocks, each composed of ReLU → Conv2d(128, 32, 3×3) → ReLU → Conv2d(32, 128, 1×1). The output is flattened and passed through two fully connected layers (2048 → 10) to generate the latent mean and log-variance vectors for a 10-dimensional latent space.

The decoder mirrors the encoder structure in reverse. It begins with a fully connected layer ($10 \rightarrow 2048$), followed by a transposed convolution: ConvTranspose2d(128, 128, 3×3, stride 2, padding 1). This is followed by two residual blocks identical in structure to those in the encoder, and then two upsampling layers: (1) ConvTranspose2d(128, 64, 3×3, stride 2, padding 1, output padding 1) with ReLU activation; (2) ConvTranspose2d(64, 1, 3×3, stride 2, padding 1, output padding 1) with Sigmoid activation to reconstruct the 28×28 grayscale image.

The model was trained for 35 epochs with a MSE reconstruction loss. For MNIST, we used $\beta = 30$, $\gamma = 5$, and target capacity $C = 30$ for both FVH-LT and DBSR-LS. Latent traversals were performed within the fix range $[-2, 2]$. Each experiment was repeated with $R = 10$ independent runs to ensure stability and enable consistent latent alignment. The regularization coefficients were set to $\lambda_B = 1 \times 10^{-3}$ and $\lambda_S = 1 \times 10^{-3}$. The correlation threshold for alignment was set to $\rho_{\min} = 0.3$.

### B.4 CelebA

We employed the disentangled $\beta$-VAE architecture implemented in the Pythae library. Specifically, we used a ResNet-based encoder-decoder design tailored for CelebA images. The encoder consists of four sequential convolutional blocks: (1) a convolutional layer with 64 output channels, 4×4 kernel, stride 2, and padding 1; (2) a convolutional layer with 128 output channels, 4×4 kernel, stride 2, and padding 1; (3) a convolutional layer with 128 output channels, 3×3 kernel, stride 2, and padding 1; (4) a convolutional layer with 128 output channels, 3×3 kernel, stride 2, and padding 1.

This is followed by two residual blocks, each composed of ReLU $\rightarrow$ Conv2d(128, 32, 3×3) $\rightarrow$ ReLU $\rightarrow$ Conv2d(32, 128, 1×1). The output is flattened and passed through two fully connected layers ($2048 \rightarrow 20$) to generate the latent mean and log-variance vectors for a 16-dimensional latent space.

The decoder mirrors the encoder structure in reverse. It begins with a fully connected layer ($20 \rightarrow 2048$), followed by a transposed convolution: ConvTranspose2d(128, 128, 3×3, stride 2, padding 1). This is followed by two residual blocks identical in structure to those in the encoder, and then three upsampling layers: (1) ConvTranspose2d(128, 128, 5×5, stride 2, padding 1) with Sigmoid activation; (2) ConvTranspose2d(128, 64, 5×5, stride 2, padding 1, output padding 1); (3) ConvTranspose2d(64, 3, 4×4, stride 2, padding 1) with Sigmoid activation to reconstruct the 64×64 RGB image.

The model was trained for 64 epochs with a MSE reconstruction loss, $\beta = 100$, $\gamma = 10$, and target capacity $C = 125$ (Eq. (1)) for the FVH-LT method. Latent traversals were performed using a data-dependent range of $[\mu - 1, \mu + 1]$, where $\mu$ denotes the posterior mean of each latent dimension. Each experiment was repeated with $R = 5$ independent runs to ensure stability and enable consistent latent alignment. For visualization, FVH-LT was applied separately to each color channel, and the resulting variance heatmaps were globally normalized and combined into a single composite heatmap. DBSR-LS was not applied to CelebA due to the high dimensionality of image features.

### B.5 FactorVAE as a benchmark method

For FA15, FA100, RNA-seq, White wine, 2018 FIFA statistics, MNIST, and CelebA, the benchmark VAE methods used the same dataset-specific architecture, optimizer settings, train–test split, latent dimensionality, and alignment procedure as the corresponding bfVAE experiments described above.

For FA15, we used $\gamma = 3$ for both FVH-LT and DBSR-LS. For FA100, we used $\gamma = 10$ for both FVH-LT and DBSR-LS. For RNA-seq, we used $\gamma = 10$ for FVH-LT. For White wine, we used $\gamma = 0.7$ for both FVH-LT and DBSR-LS. For 2018 FIFA statistics, we used $\gamma = 5$ for both FVH-LT and DBSR-LS. For MNIST, we used $\gamma = 5$. For CelebA, we used $\gamma = 10$ for FVH-LT.

### B.6 $\beta$-VAE as a benchmark method

For FA15, FA100, RNA-seq, White wine, 2018 FIFA statistics, MNIST, and CelebA, the benchmark VAE methods used the same dataset-specific architecture, optimizer settings, train–test split, latent dimensionality, and alignment procedure as the corresponding bfVAE experiments described above.

For $\beta$-VAE, we used: $\beta = 0.03$ for FA15, $\beta = 0.05$ for FA100, $\beta = 3$ for RNA-seq, $\beta = 0.3$ for White wine, $\beta = 0.03$ for 2018 FIFA statistics, $\beta = 20$ for MNIST, and $\beta = 150$ for CelebA. The same $\beta$ was used for both FVH-LT and DBSR-LS when both methods were evaluated; RNA-seq and CelebA were evaluated using FVH-LT only.

### B.7 Vanilla VAE as a benchmark method

For FA15, FA100, RNA-seq, White wine, 2018 FIFA statistics, MNIST, and CelebA, the benchmark VAE methods used the same dataset-specific architecture, optimizer settings, train–test split, latent dimensionality, and alignment procedure as the corresponding bfVAE experiments described above.

For the Vanilla VAE, no additional hyperparameters were used; equivalently, we set $\beta = 1$, $\gamma = 0$, and $C = 0$ in the VAE objective.

### B.8 DIP-VAE-I as a benchmark method

For the DIP-VAE-I experiments, we used a fully connected VAE architecture with a symmetric encoder-decoder design, trained using the loss function

$$\mathcal{L}(\boldsymbol{\phi}, \boldsymbol{\omega}) = \frac{1}{n} \sum_{i=1}^{n} \left[ -\mathbb{E}_{q(\mathbf{z}_i|\mathbf{x}_i,\boldsymbol{\phi})} \log p(\mathbf{x}_i|\mathbf{z}_i, \boldsymbol{\omega}) + \mathrm{KL}(q(\mathbf{z}_i|\mathbf{x}_i, \boldsymbol{\phi}) \| p(\mathbf{z}_i)) \right]$$
$$+ \lambda_{\mathrm{od}} \sum_{j \neq j'=1}^{K} \left[ \mathrm{Cov}_{p(\mathbf{x})}[\boldsymbol{\mu}_\phi(\mathbf{x})] \right]_{jj'}^2 + \lambda_d \sum_{j=1}^{K} \left( \left[ \mathrm{Cov}_{p(\mathbf{x})}[\boldsymbol{\mu}_\phi(\mathbf{x})] \right]_{jj} - 1 \right)^2.$$

For DIP-VAE-I, we used dataset-specific DIP regularization weights: $(\lambda_{od}, \lambda_D) = (1, 1)$ for FA15, $(0.1, 1)$ for FA100, $(10, 10)$ for RNA-seq, $(0.1, 0.5)$ for White wine, $(0.1, 0.5)$ for 2018 FIFA statistics, $(1, 10)$ for MNIST, and $(1, 10)$ for CelebA. The same DIP-VAE-I hyperparameters were used for both FVH-LT and DBSR-LS when both methods were evaluated. For DBSR-LS, the regularization coefficients followed the corresponding dataset-specific DBSR-LS settings described above.

### B.9 DIP-VAE-II as a benchmark method

For the DIP-VAE-II experiments, we used a fully connected VAE architecture with a symmetric encoder-decoder design, trained using the loss function

$$\mathcal{L}(\boldsymbol{\phi}, \boldsymbol{\omega}) = \frac{1}{n} \sum_{i=1}^{n} \left[ -\mathbb{E}_{q(\mathbf{z}_i|\mathbf{x}_i,\boldsymbol{\phi})} \log p(\mathbf{x}_i|\mathbf{z}_i, \boldsymbol{\omega}) + \mathrm{KL}(q(\mathbf{z}_i|\mathbf{x}_i, \boldsymbol{\phi}) \| p(\mathbf{z}_i)) \right]$$
$$+ \lambda_{\mathrm{od}} \sum_{j \neq j'=1}^{K} \left[ \mathrm{Cov}_{q_\phi(\mathbf{z})}[\mathbf{z}] \right]_{jj'}^2 + \lambda_d \sum_{j=1}^{K} \left( \left[ \mathrm{Cov}_{q_\phi(\mathbf{z})}[\mathbf{z}] \right]_{jj} - 1 \right)^2.$$

For DIP-VAE-II, we used dataset-specific DIP regularization weights: $(\lambda_{od}, \lambda_D) = (0.1, 1)$ for FA15, $(0.01, 1)$ for FA100, $(5, 10)$ for RNA-seq, $(0.05, 0.5)$ for White wine, $(0.01, 0.5)$ for 2018 FIFA statistics, $(1, 1)$ for MNIST, and $(1, 1)$ for CelebA. The same DIP-VAE-II hyperparameters were used for both FVH-LT and DBSR-LS when both methods were evaluated. For DBSR-LS, the regularization coefficients followed the corresponding dataset-specific DBSR-LS settings described above.

## C Domain Knowledge in the Wine Quality Experiment

VC is a key indicator of fermentation quality and excessive levels often lead to sensory defects. C contributes to the perceived wine saltiness and may reflect suboptimal production conditions. D captures the balance between residual sugar and A, influencing wine texture. A strongly affects mouthfeel and overall quality perception (higher levels are generally associated with higher quality).

# D    Benchmark Results

## D.1    FA15 : FVH-LT comparision

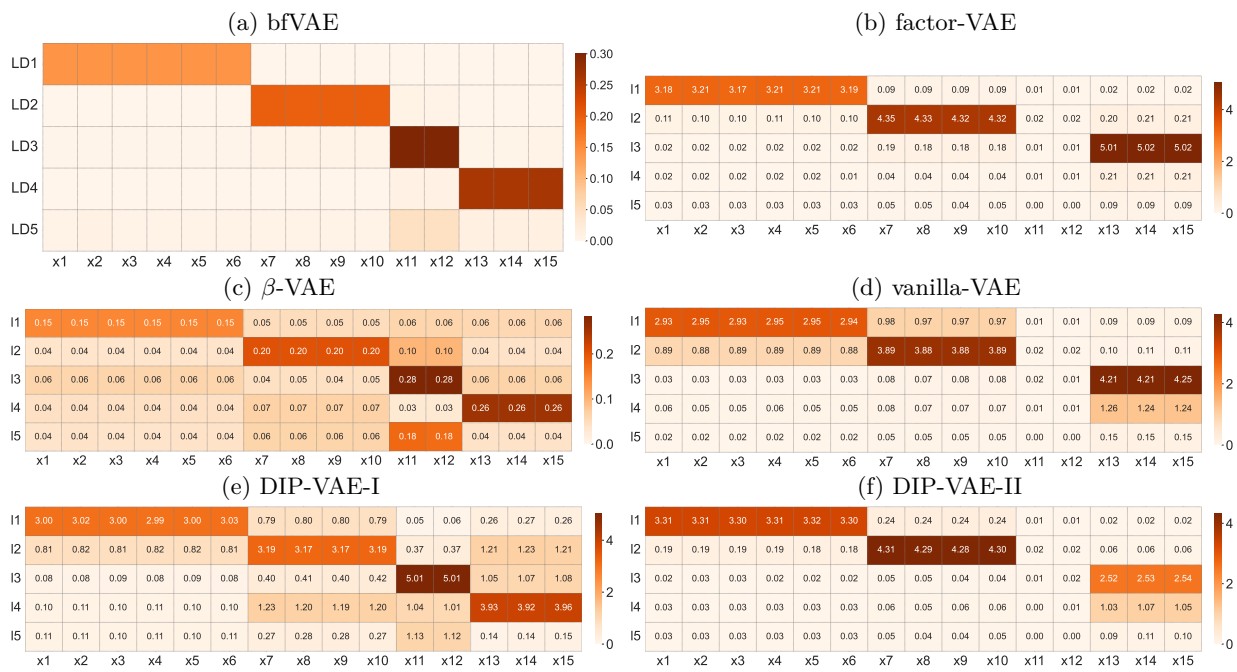

## D.2    FA15 : DBSR-LS comparision

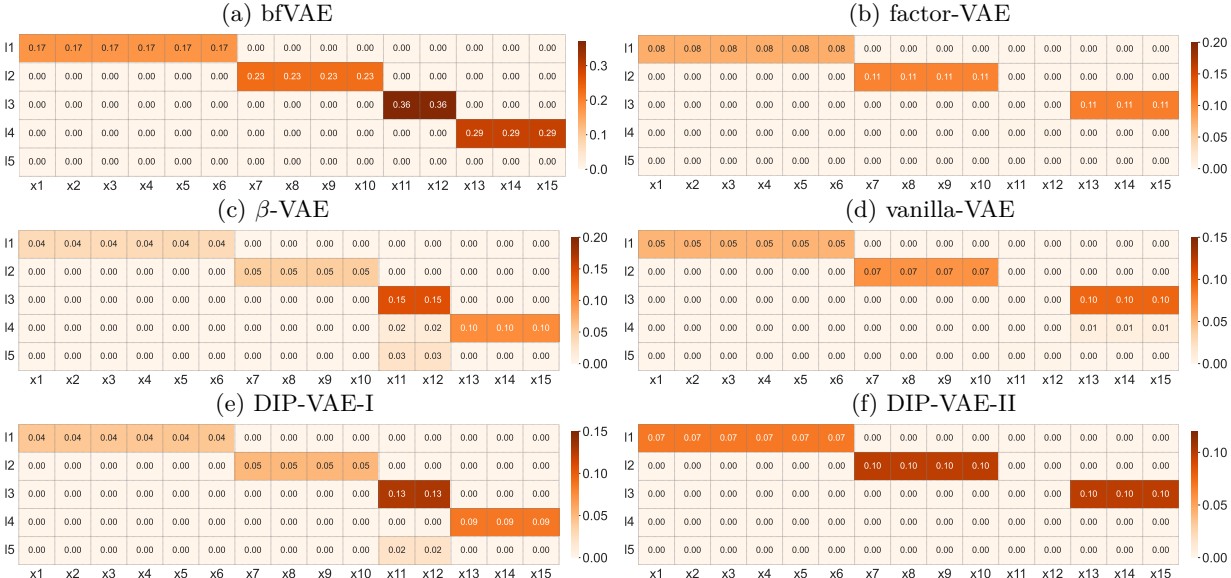

## D.3 FA100 : FVH-LT comparision

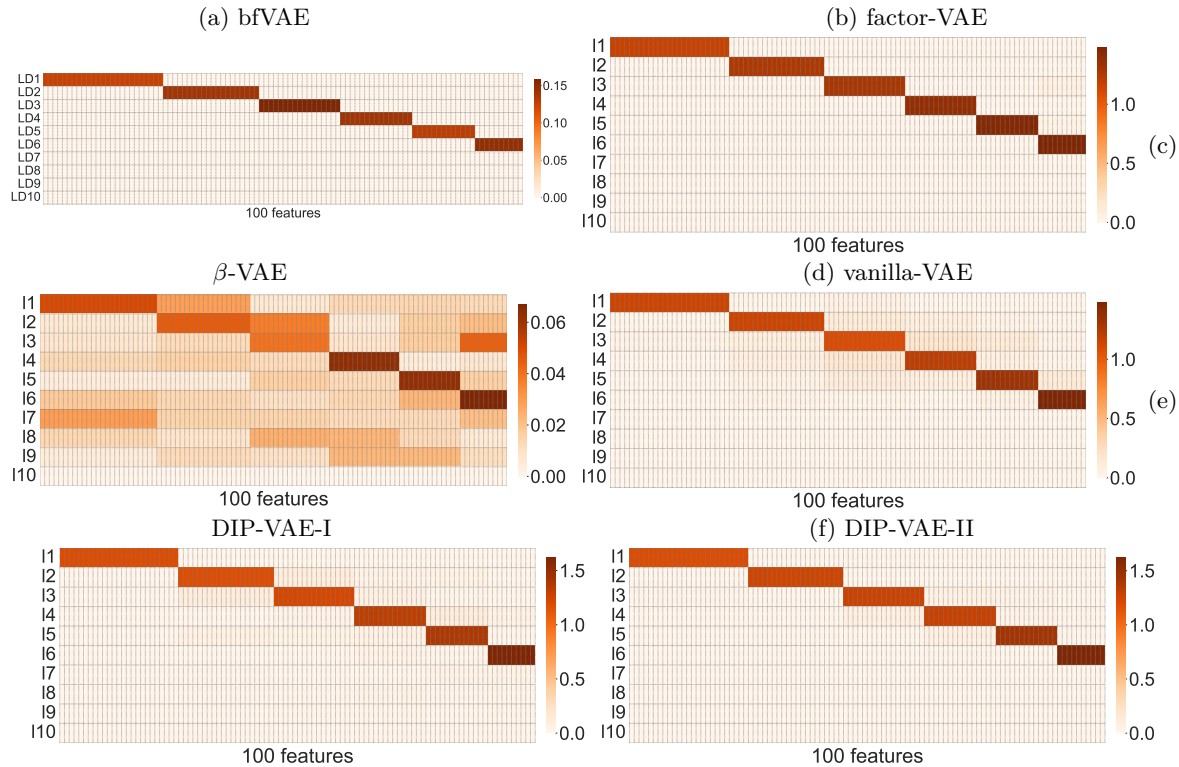

## D.4 FA100 : DBSR-LS comparision

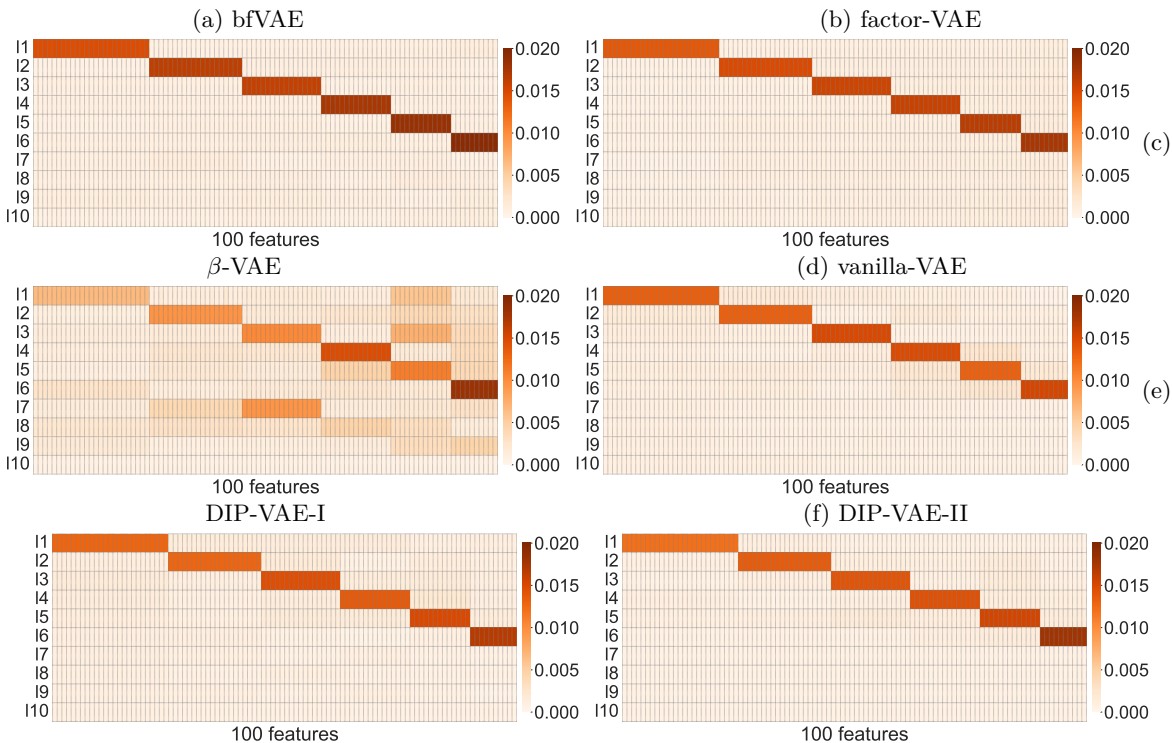

## D.5 RNA-seq: FVH-LT comparison

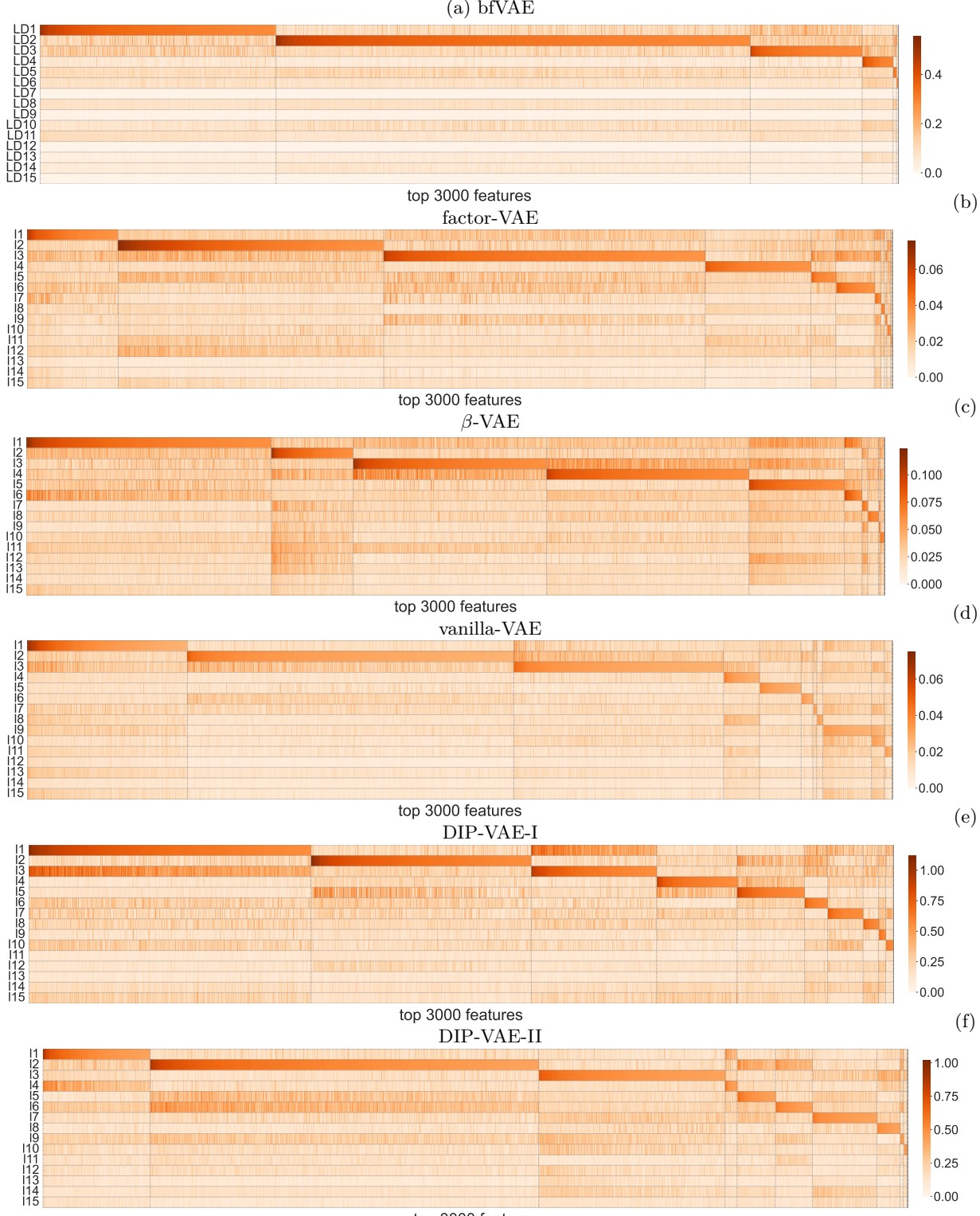

## D.6   White Wine: FVH-LT comparison

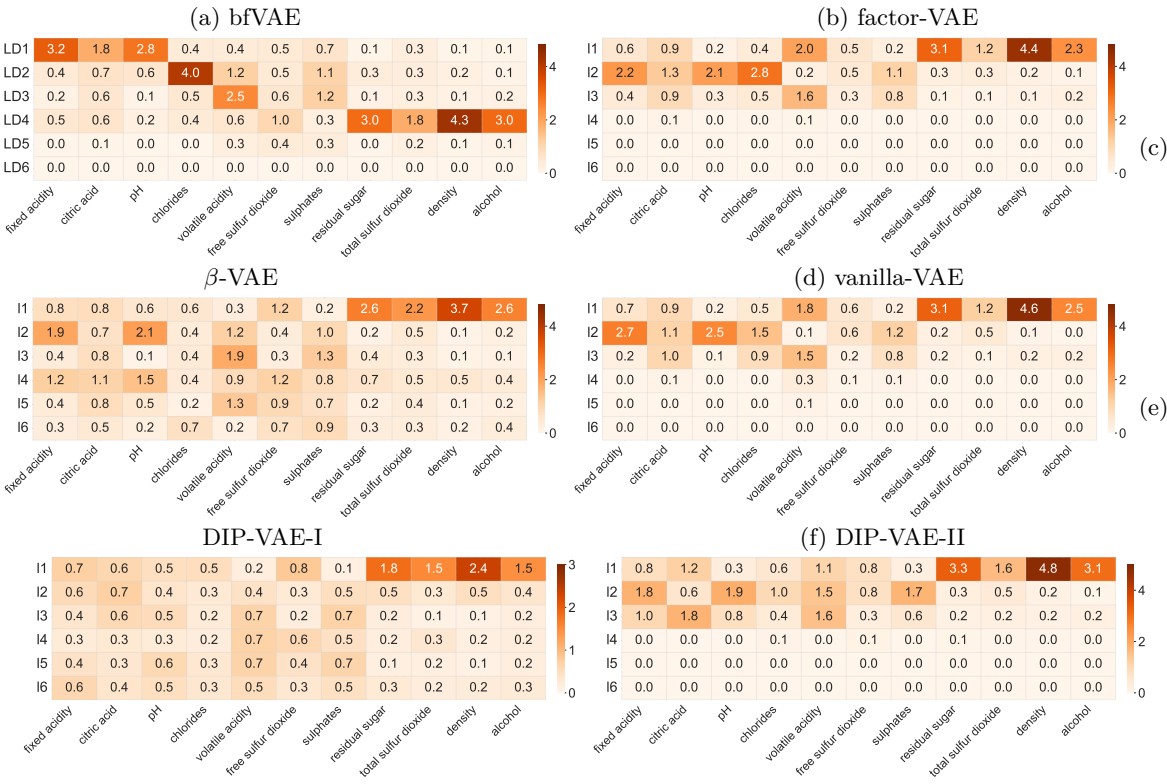

## D.7   White Wine: DBSR-LS comparison

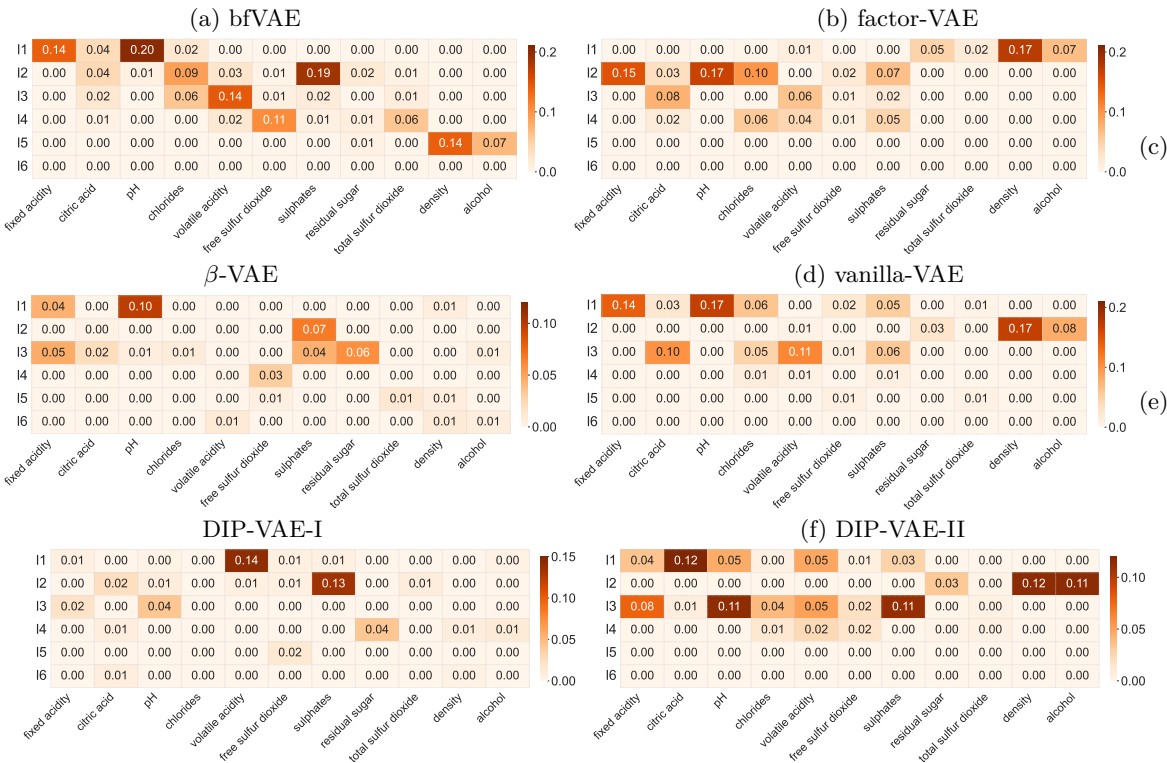

## D.8    2018 FIFA: FVH-LT comparison

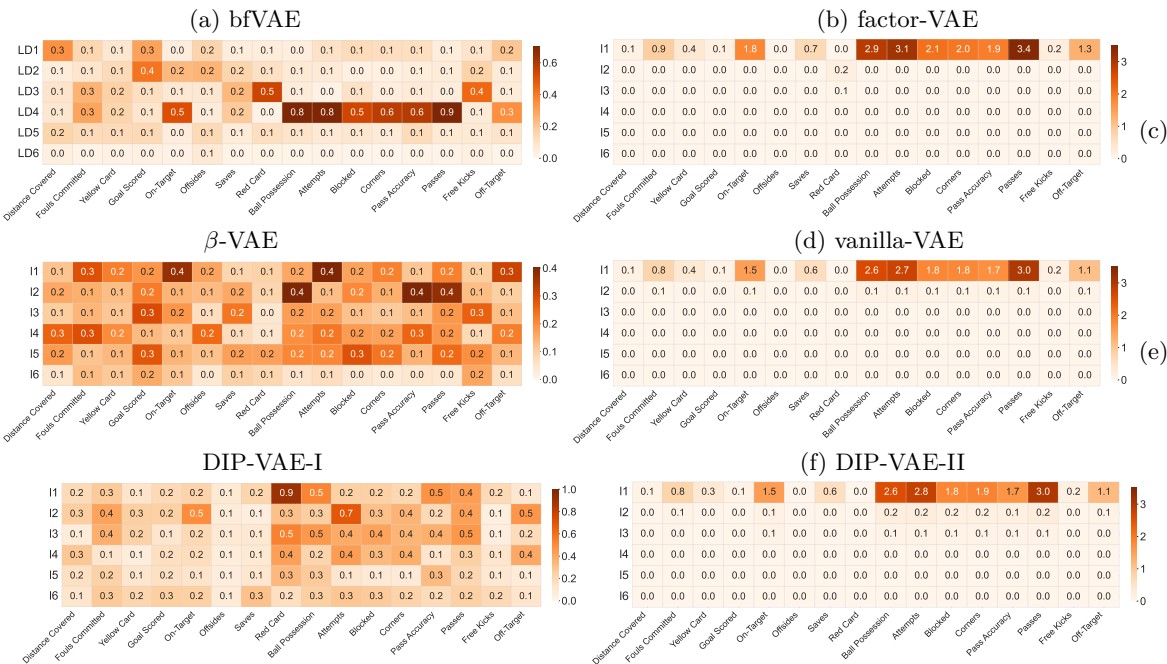

## D.9    2018 FIFA: DBSR-LS comparison

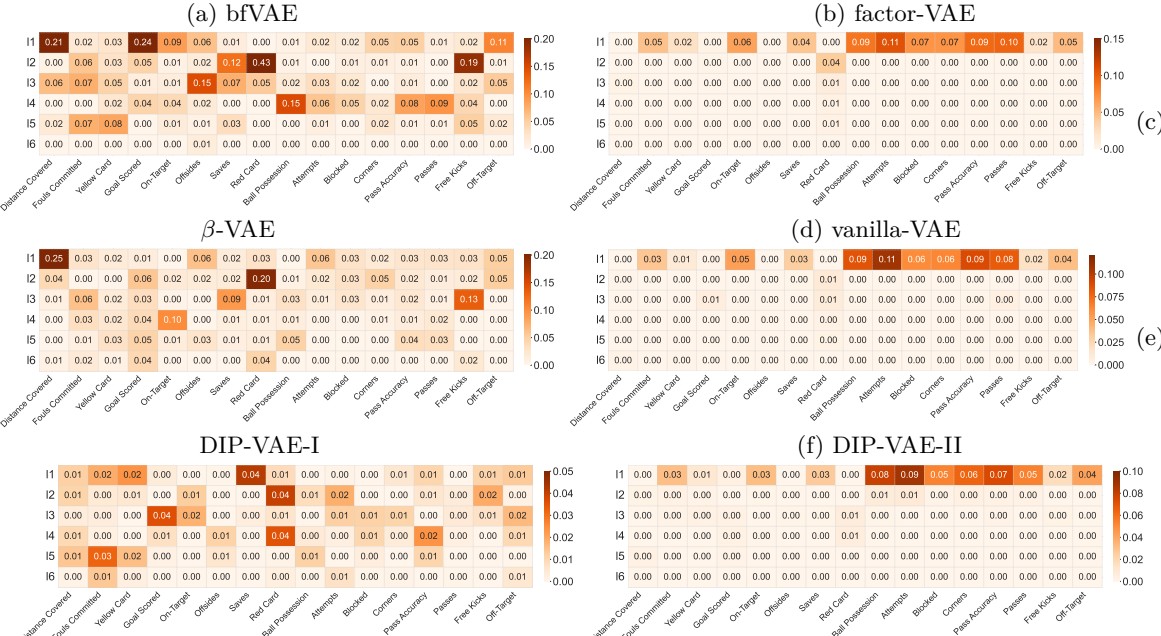

## D.10  MNIST: FVH-LT comparison

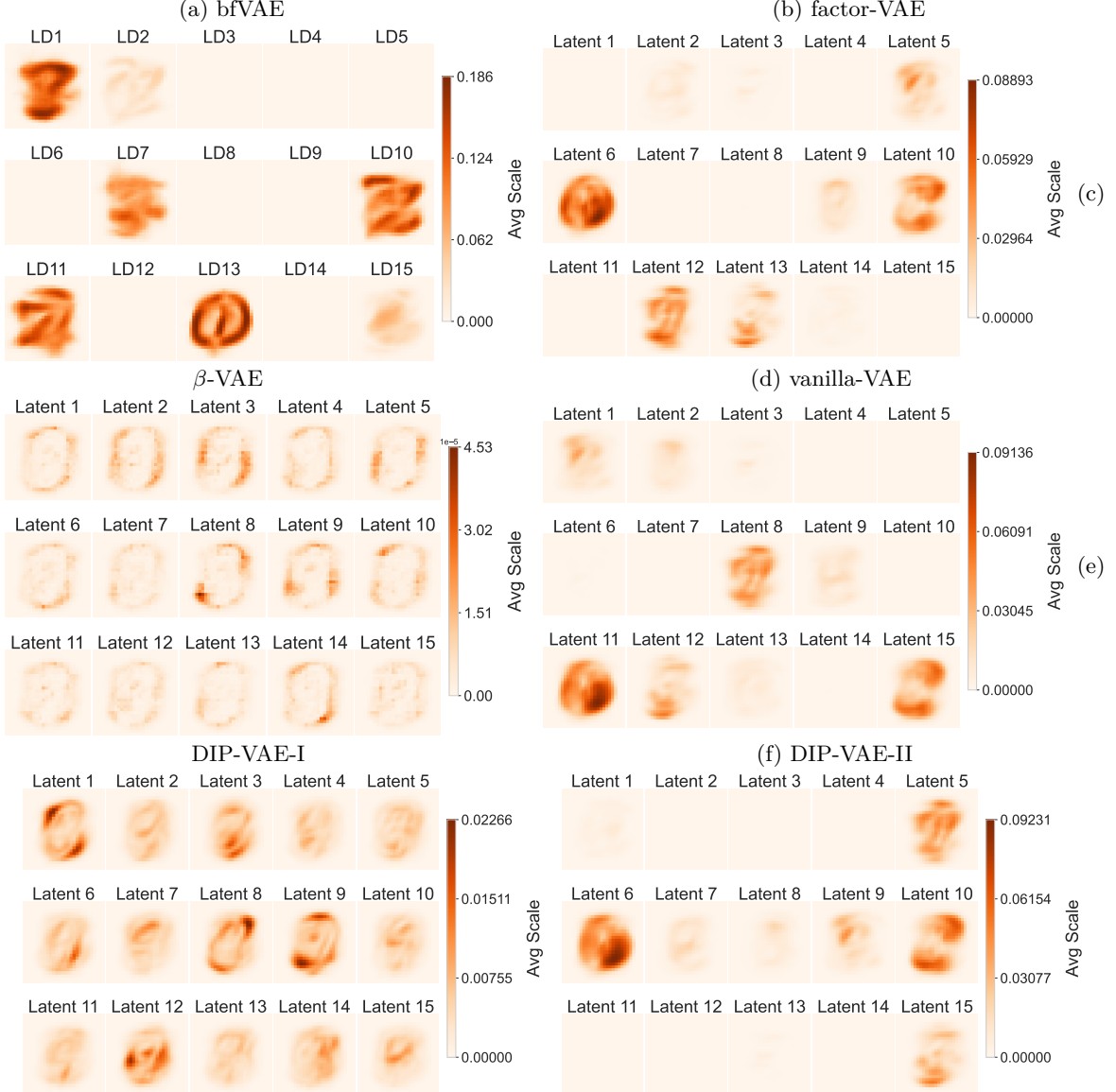

## D.11 CelebA: FVH-LT comparison

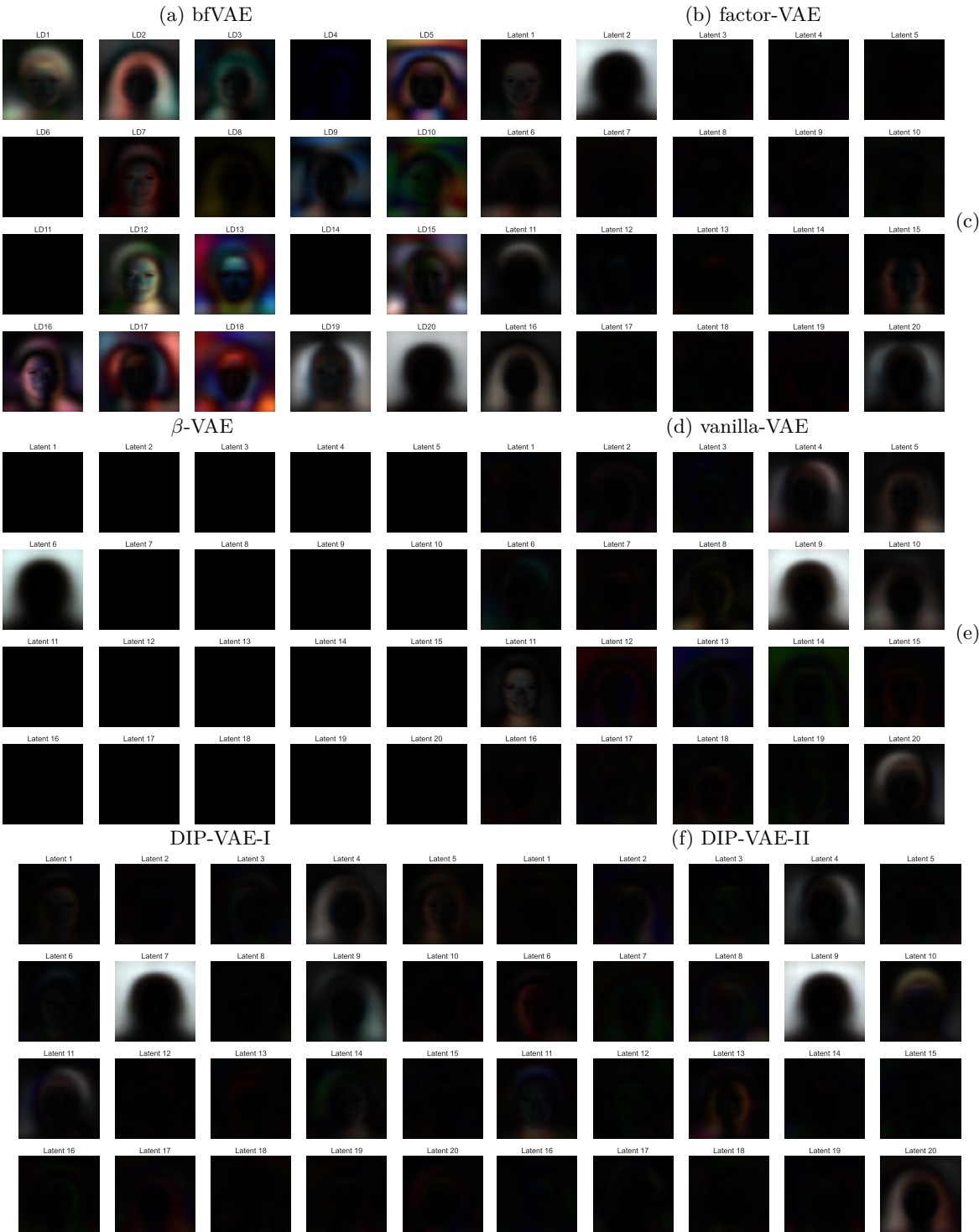

# E  LSSI on All LDs and Informative LDs

## E.1  FA15: LSSI on all LD and informative LDs

| Data | Method | Model | $K_{\max}$ | $\tau$ | $c_1$ | $c_{1,\text{info}}$ | $c_2$ | $c_{2,\text{info}}$ | LSSI | $\text{LSSI}_{\text{info}}$ |
|---|---|---|---|---|---|---|---|---|---|---|
| FA15 | FVH-LT | bfVAE | 4 | 0.142 | 1.000 | 1.000 | 0.957 | 0.957 | **0.891** | **0.922** |
| | | Factor VAE | 3 | 0.213 | 0.867 | 0.867 | 0.998 | 0.986 | *0.807* | *0.805* |
| | | $\beta$-VAE | 4 | 0.149 | 0.867 | 0.867 | 1.000 | 0.969 | 0.366 | 0.393 |
| | | Vanilla VAE | 4 | 1.244 | 0.867 | 0.867 | 0.750 | 0.750 | 0.544 | 0.527 |
| | | DIP-VAE-I | 4 | 2.995 | 1.000 | 1.000 | 0.935 | 0.935 | 0.633 | 0.618 |
| | | DIP-VAE-II | 4 | 1.048 | 0.867 | 0.867 | 0.635 | 0.635 | 0.498 | 0.496 |
| | DBSR-LS | bfVAE | 4 | 0.173 | 1.000 | 1.000 | 0.979 | 0.979 | *0.979* | *0.979* |
| | | Factor VAE | 3 | 0.003 | 1.000 | 0.867 | 0.988 | 0.975 | **0.987** | 0.845 |
| | | $\beta$-VAE | 4 | 0.037 | 1.000 | 1.000 | 0.966 | 0.966 | 0.907 | 0.940 |
| | | Vanilla VAE | 4 | 0.008 | 0.867 | 0.867 | 0.699 | 0.699 | 0.597 | 0.595 |
| | | DIP-VAE-I | 4 | 0.042 | 1.000 | 1.000 | 0.989 | 0.989 | 0.962 | **0.984** |
| | | DIP-VAE-II | 3 | 0.000 | 0.867 | 0.867 | 0.971 | 0.971 | 0.841 | 0.841 |

**Bold** denotes the highest score within each method and LSSI column; ***underlining*** denotes the second highest.

## E.2  FA100: LSSI on all LD and informative LDs

| Data | Method | Model | $K_{\max}$ | $\tau$ | $c_1$ | $c_{1,\text{info}}$ | $c_2$ | $c_{2,\text{info}}$ | LSSI | $\text{LSSI}_{\text{info}}$ |
|---|---|---|---|---|---|---|---|---|---|---|
| FA100 | FVH-LT | bfVAE | 6 | 0.124 | 1.000 | 1.000 | 0.924 | 0.924 | 0.913 | 0.915 |
| | | Factor VAE | 6 | 1.171 | 1.000 | 1.000 | 0.949 | 0.949 | **0.940** | **0.935** |
| | | $\beta$-VAE | 9 | 0.040 | 0.900 | 0.900 | 0.591 | 0.591 | 0.252 | 0.218 |
| | | Vanilla VAE | 6 | 1.087 | 1.000 | 1.000 | 0.941 | 0.941 | 0.866 | 0.809 |
| | | DIP-VAE-I | 6 | 1.177 | 1.000 | 1.000 | 0.952 | 0.952 | 0.874 | 0.887 |
| | | DIP-VAE-II | 6 | 1.183 | 1.000 | 1.000 | 0.951 | 0.951 | *0.929* | *0.915* |
| | DBSR-LS | bfVAE | 6 | 0.015 | 1.000 | 1.000 | 0.952 | 0.952 | *0.636* | **0.728** |
| | | Factor VAE | 6 | 0.014 | 1.000 | 1.000 | 0.945 | 0.945 | 0.602 | 0.702 |
| | | $\beta$-VAE | 9 | 0.007 | 0.750 | 0.750 | 0.605 | 0.605 | 0.217 | 0.204 |
| | | Vanilla VAE | 6 | 0.013 | 1.000 | 1.000 | 0.925 | 0.925 | 0.591 | 0.649 |
| | | DIP-VAE-I | 6 | 0.013 | 1.000 | 1.000 | 0.946 | 0.946 | 0.592 | 0.644 |
| | | DIP-VAE-II | 6 | 0.012 | 1.000 | 1.000 | 0.963 | 0.963 | **0.660** | *0.710* |

**Bold** denotes the highest score within each method and LSSI column; ***underlining*** denotes the second highest.

## E.3  RNA-seq: LSSI on all LD and informative LDs

| Data | Method | Model | $K_{\max}$ | $\tau$ | $c_1$ | $c_{1,\text{info}}$ | $c_2$ | $c_{2,\text{info}}$ | LSSI | $\text{LSSI}_{\text{info}}$ |
|---|---|---|---|---|---|---|---|---|---|---|
| RNA-seq | FVH-LT | bfVAE | 11 | 0.145 | 0.637 | 0.637 | 0.285 | 0.285 | **0.081** | 0.054 |
| | | Factor VAE | 14 | 0.024 | 0.542 | 0.542 | 0.455 | 0.454 | 0.057 | 0.054 |
| | | $\beta$-VAE | 12 | 0.040 | 0.498 | 0.498 | 0.574 | 0.566 | 0.056 | 0.053 |
| | | Vanilla VAE | 12 | 0.018 | 0.493 | 0.493 | 0.559 | 0.541 | 0.052 | 0.045 |
| | | DIP-VAE-I | 13 | 0.343 | 0.502 | 0.502 | 0.590 | 0.589 | *0.072* | **0.071** |
| | | DIP-VAE-II | 14 | 0.278 | 0.477 | 0.477 | 0.560 | 0.560 | 0.058 | *0.057* |

**Bold** denotes the highest score within each method and LSSI column; ***underlining*** denotes the second highest.

### E.4 White wine: LSSI on all LD and informative LDs

| Data | Method | Model | $K_{\max}$ | $\tau$ | $c_1$ | $c_{1,\text{info}}$ | $c_2$ | $c_{2,\text{info}}$ | LSSI | $\text{LSSI}_{\text{info}}$ |
|---|---|---|---|---|---|---|---|---|---|---|
| White wine | FVH-LT | bfVAE | 5 | 1.176 | 0.909 | 0.909 | 0.586 | 0.586 | **0.390** | **0.355** |
| | | Factor VAE | 4 | 1.132 | 0.909 | 0.909 | 0.407 | 0.407 | *0.308* | *0.270* |
| | | $\beta$-VAE | 5 | 1.206 | 0.727 | 0.727 | 0.534 | 0.534 | 0.175 | 0.176 |
| | | Vanilla VAE | 5 | 1.140 | 0.909 | 0.909 | 0.296 | 0.296 | 0.220 | 0.208 |
| | | DIP-VAE-I | 4 | 0.684 | 0.818 | 0.727 | 0.295 | 0.112 | 0.090 | 0.033 |
| | | DIP-VAE-II | 4 | 1.250 | 0.818 | 0.818 | 0.434 | 0.434 | 0.280 | 0.236 |
| | DBSR-LS | bfVAE | 5 | 0.041 | 0.909 | 0.909 | 0.910 | 0.910 | **0.736** | **0.714** |
| | | Factor VAE | 4 | 0.046 | 0.818 | 0.818 | 0.657 | 0.657 | 0.478 | *0.440* |
| | | $\beta$-VAE | 5 | 0.011 | 0.818 | 0.636 | 0.650 | 0.473 | 0.466 | 0.281 |
| | | Vanilla VAE | 5 | 0.028 | 0.818 | 0.818 | 0.475 | 0.475 | 0.356 | 0.348 |
| | | DIP-VAE-I | 4 | 0.010 | 0.909 | 0.545 | 0.855 | 0.479 | *0.719* | 0.245 |
| | | DIP-VAE-II | 4 | 0.036 | 0.727 | 0.727 | 0.609 | 0.609 | 0.396 | 0.365 |

**Bold** denotes the highest score within each method and LSSI column; ***underlining*** denotes the second highest.

### E.5 2018 FIFA: LSSI on all LD and informative LDs

| Data | Method | Model | $K_{\max}$ | $\tau$ | $c_1$ | $c_{1,\text{info}}$ | $c_2$ | $c_{2,\text{info}}$ | LSSI | $\text{LSSI}_{\text{info}}$ |
|---|---|---|---|---|---|---|---|---|---|---|
| 2018 FIFA | FVH-LT | bfVAE | 5 | 0.248 | 0.875 | 0.875 | 0.252 | 0.252 | *0.132* | *0.120* |
| | | Factor VAE | 2 | 0.052 | 0.938 | 0.938 | 0.022 | 0.017 | 0.020 | 0.015 |
| | | $\beta$-VAE | 5 | 0.249 | 0.750 | 0.750 | 0.903 | 0.903 | **0.197** | **0.168** |
| | | Vanilla VAE | 2 | 0.116 | 0.812 | 0.812 | 0.040 | 0.040 | 0.031 | 0.029 |
| | | DIP-VAE-I | 5 | 0.392 | 0.562 | 0.562 | 0.695 | 0.695 | 0.104 | 0.098 |
| | | DIP-VAE-II | 3 | 0.159 | 0.000 | 0.000 | 0.000 | 0.000 | 0.000 | 0.000 |
| | DBSR-LS | bfVAE | 5 | 0.064 | 0.812 | 0.812 | 0.764 | 0.764 | **0.455** | **0.421** |
| | | Factor VAE | 2 | 0.006 | 0.812 | 0.812 | 0.161 | 0.102 | 0.127 | 0.082 |
| | | $\beta$-VAE | 5 | 0.041 | 0.750 | 0.750 | 0.846 | 0.728 | 0.404 | *0.340* |
| | | Vanilla VAE | 2 | 0.005 | 0.875 | 0.875 | 0.096 | 0.063 | 0.081 | 0.053 |
| | | DIP-VAE-I | 5 | 0.014 | 0.688 | 0.625 | 1.000 | 0.799 | *0.450* | 0.323 |
| | | DIP-VAE-II | 3 | 0.006 | 0.812 | 0.812 | 0.052 | 0.041 | 0.040 | 0.030 |

**Bold** denotes the highest score within each method and LSSI column; ***underlining*** denotes the second highest.

### E.6 Image data: LSSI on all LD and informative LDs

| Data | Method | Model | $K_{\max}$ | $\tau$ | $c_1$ | $c_{1,\text{info}}$ | $c_2$ | $c_{2,\text{info}}$ | LSSI | $\text{LSSI}_{\text{info}}$ |
|---|---|---|---|---|---|---|---|---|---|---|
| MNIST | FVH-LT | bfVAE | 7 | 0.097 | 0.453 | 0.453 | 0.555 | 0.555 | **0.184** | **0.096** |
| | | Factor VAE | 9 | 0.024 | 0.406 | 0.406 | 0.292 | 0.292 | 0.089 | 0.068 |
| | | $\beta$-VAE | 14 | 0.000 | 0.258 | 0.258 | 0.724 | 0.685 | 0.053 | 0.050 |
| | | Vanilla VAE | 10 | 0.024 | 0.411 | 0.411 | 0.260 | 0.260 | 0.078 | 0.062 |
| | | DIP-VAE-I | 8 | 0.007 | 0.365 | 0.364 | 0.982 | 0.802 | 0.088 | *0.074* |
| | | DIP-VAE-II | 9 | 0.024 | 0.415 | 0.415 | 0.302 | 0.302 | *0.095* | 0.072 |
| CelebA | FVH-LT | bfVAE | 14 | 0.021 | 0.678 | 0.678 | 0.282 | 0.282 | **0.116** | **0.090** |
| | | Factor VAE | 8 | 0.005 | 0.667 | 0.667 | 0.105 | 0.105 | *0.052* | *0.049* |
| | | $\beta$-VAE | 1 | 0.000 | 0.000 | 0.000 | 0.000 | 0.000 | 0.000 | 0.000 |
| | | Vanilla VAE | 10 | 0.003 | 0.678 | 0.678 | 0.066 | 0.066 | 0.029 | 0.030 |
| | | DIP-VAE-I | 10 | 0.003 | 0.702 | 0.702 | 0.064 | 0.064 | 0.031 | 0.031 |
| | | DIP-VAE-II | 10 | 0.004 | 0.672 | 0.672 | 0.062 | 0.062 | 0.029 | 0.029 |

**Bold** denotes the highest score within each dataset and LSSI column; ***underlining*** denotes the second highest.

## F    FA24 Results

### F.1    FVH-LT result

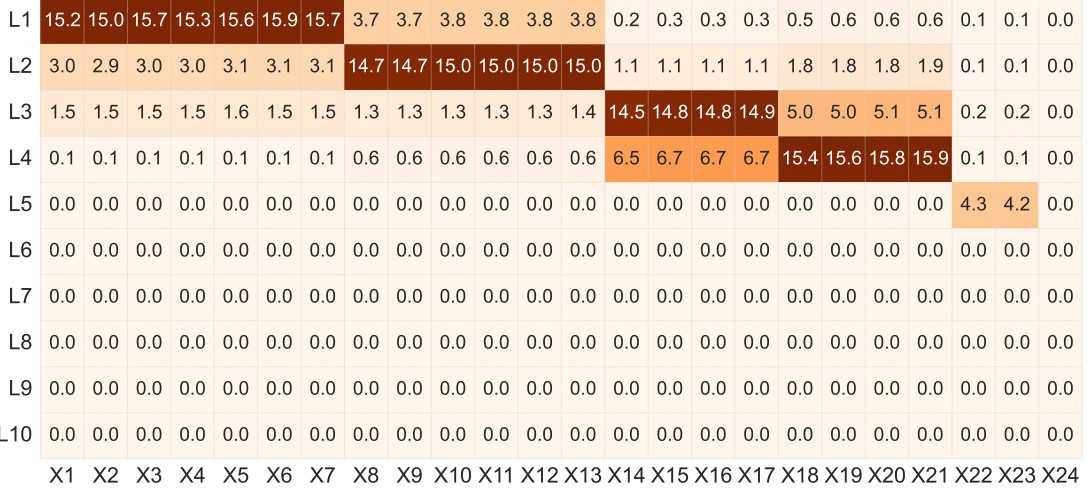

### F.2    DBSR-LS results

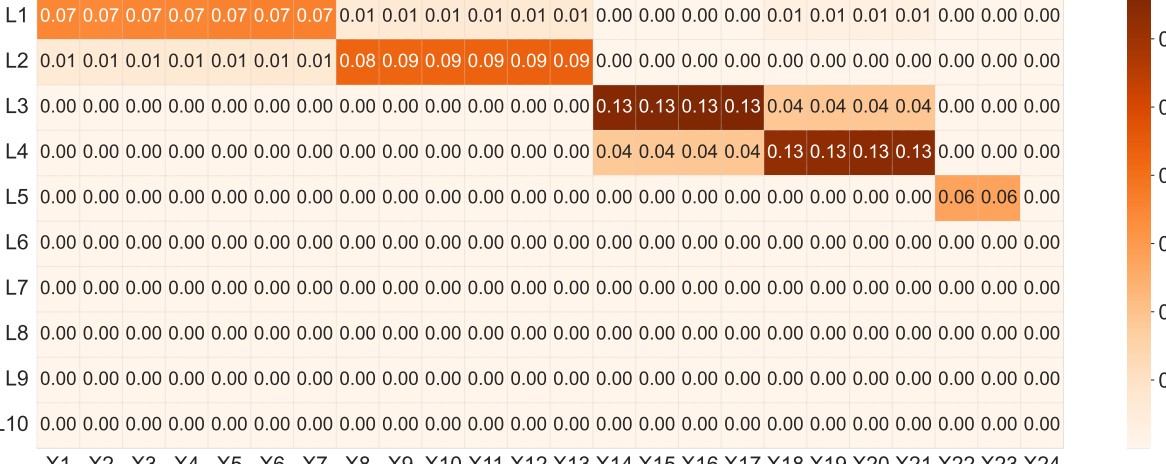

# G  Single-Run Results Corresponding to the bfVAE Results in Main Text

## G.1  FA15: FVH-LT result with bfVAE

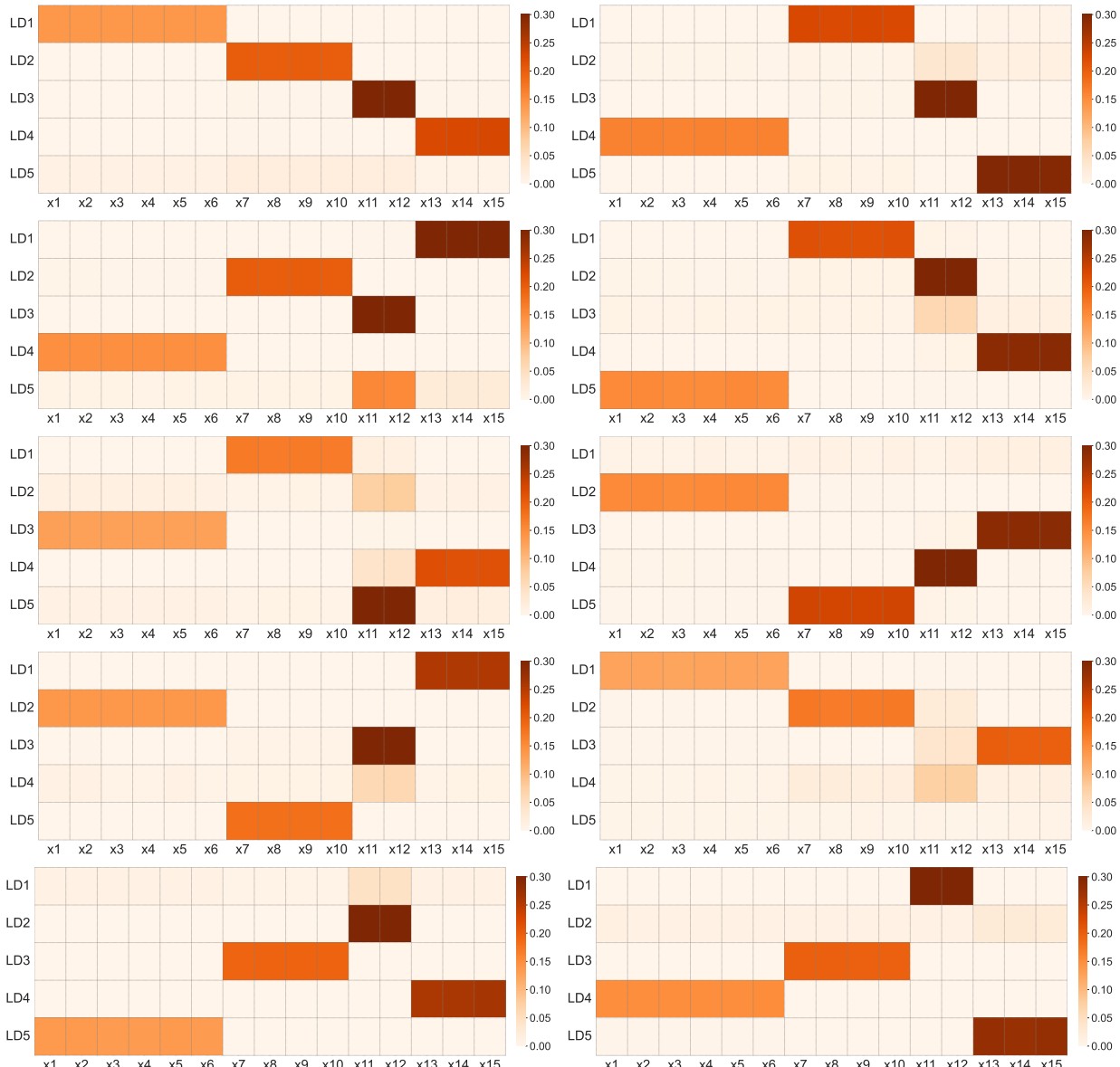

## G.2  FA15: DBSR-LS result with bfVAE

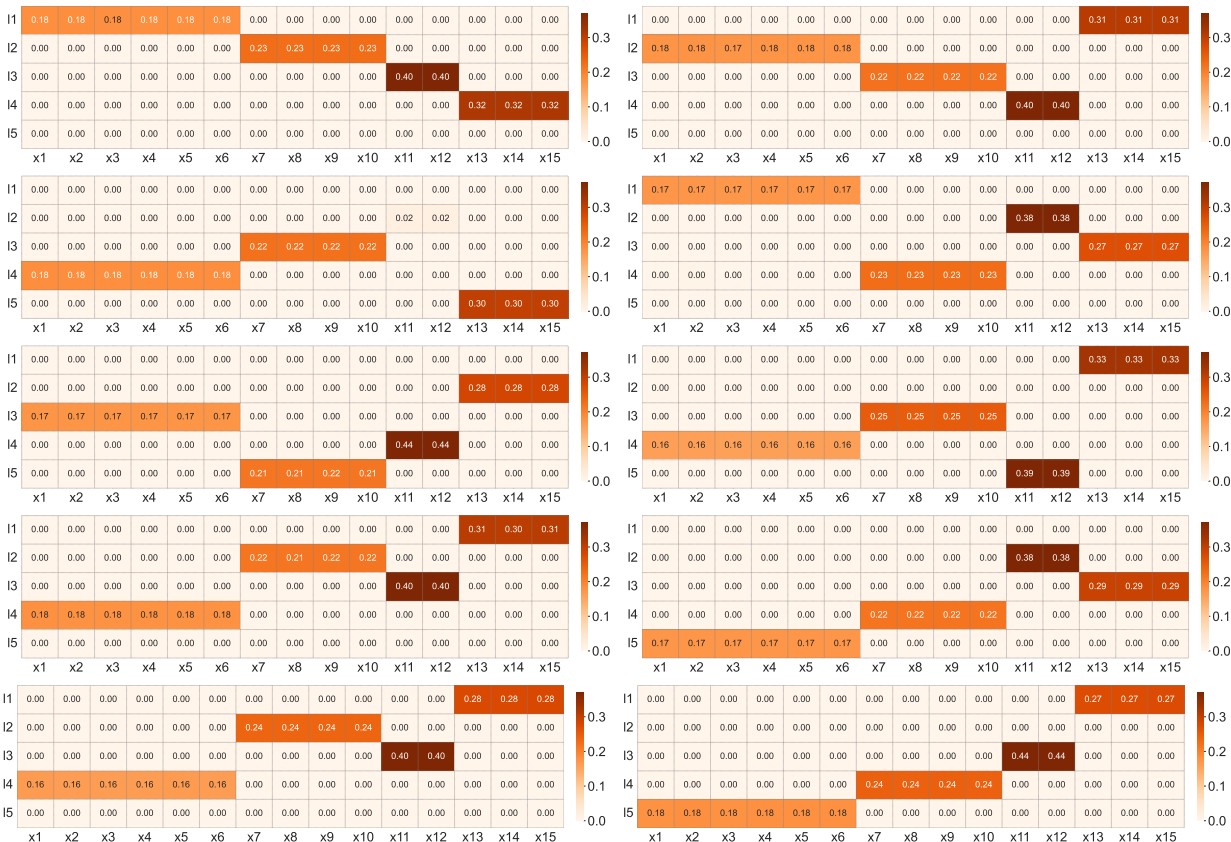

## G.3    FA100: FVH-LT result with bfVAE

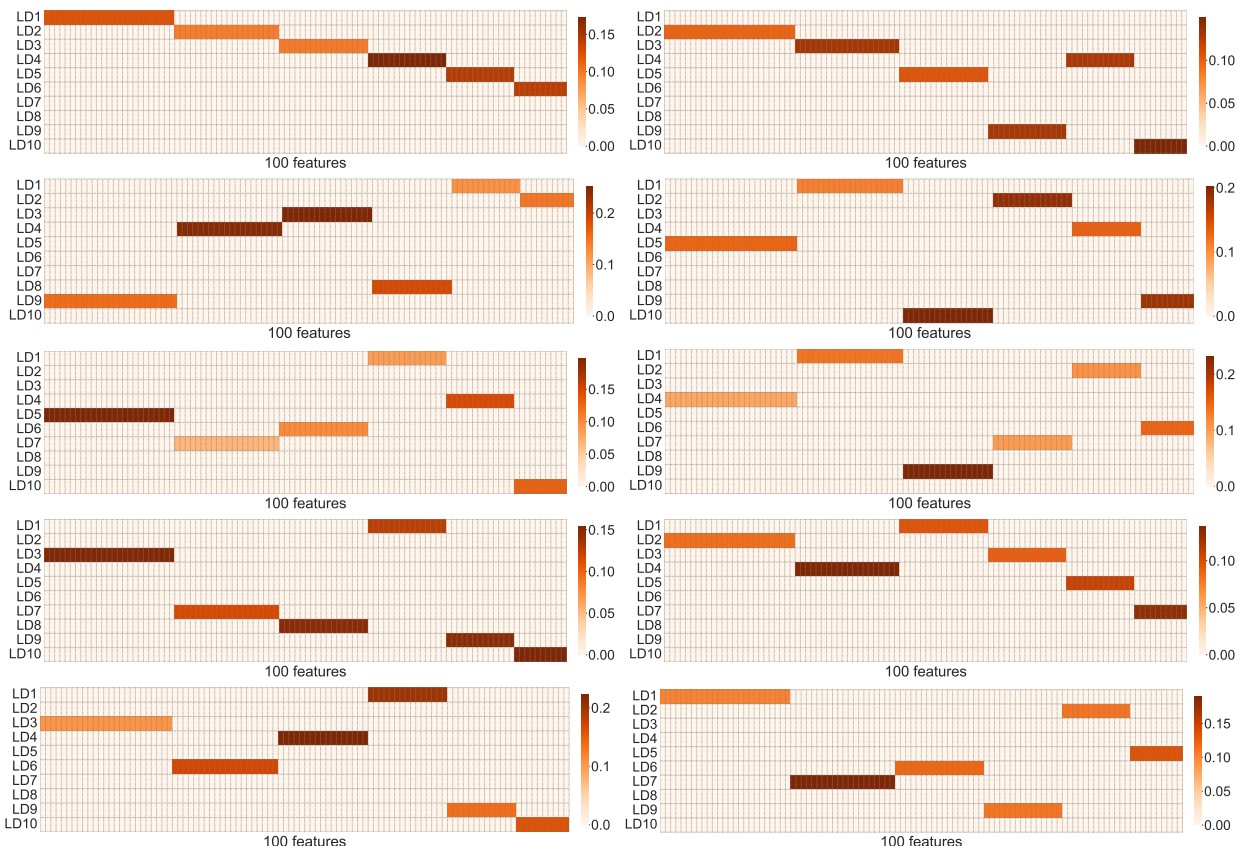

## G.4   FA100: DBSR-LS result with bfVAE

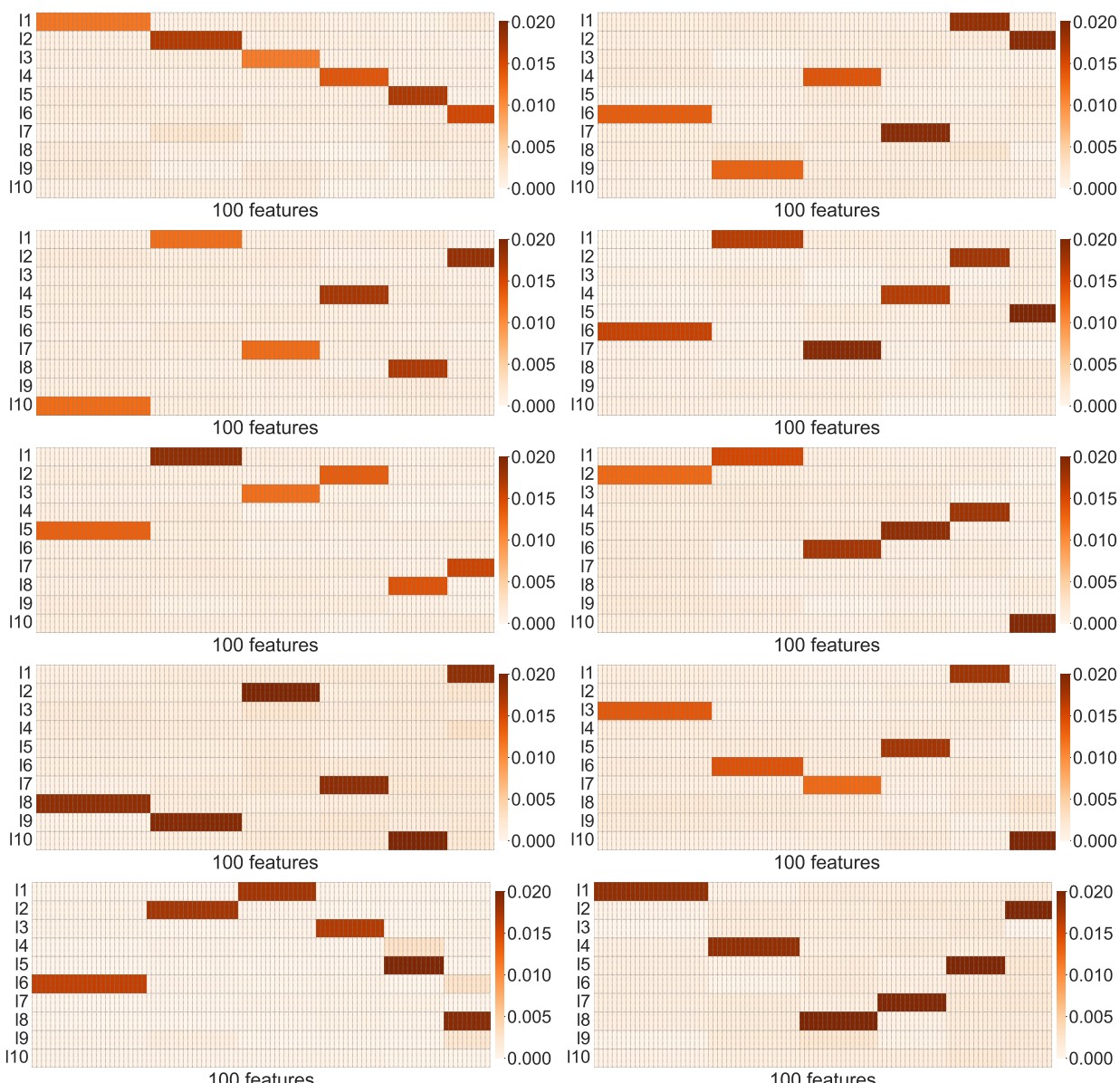

## G.5    RNA-seq: FVH-LT result with bfVAE

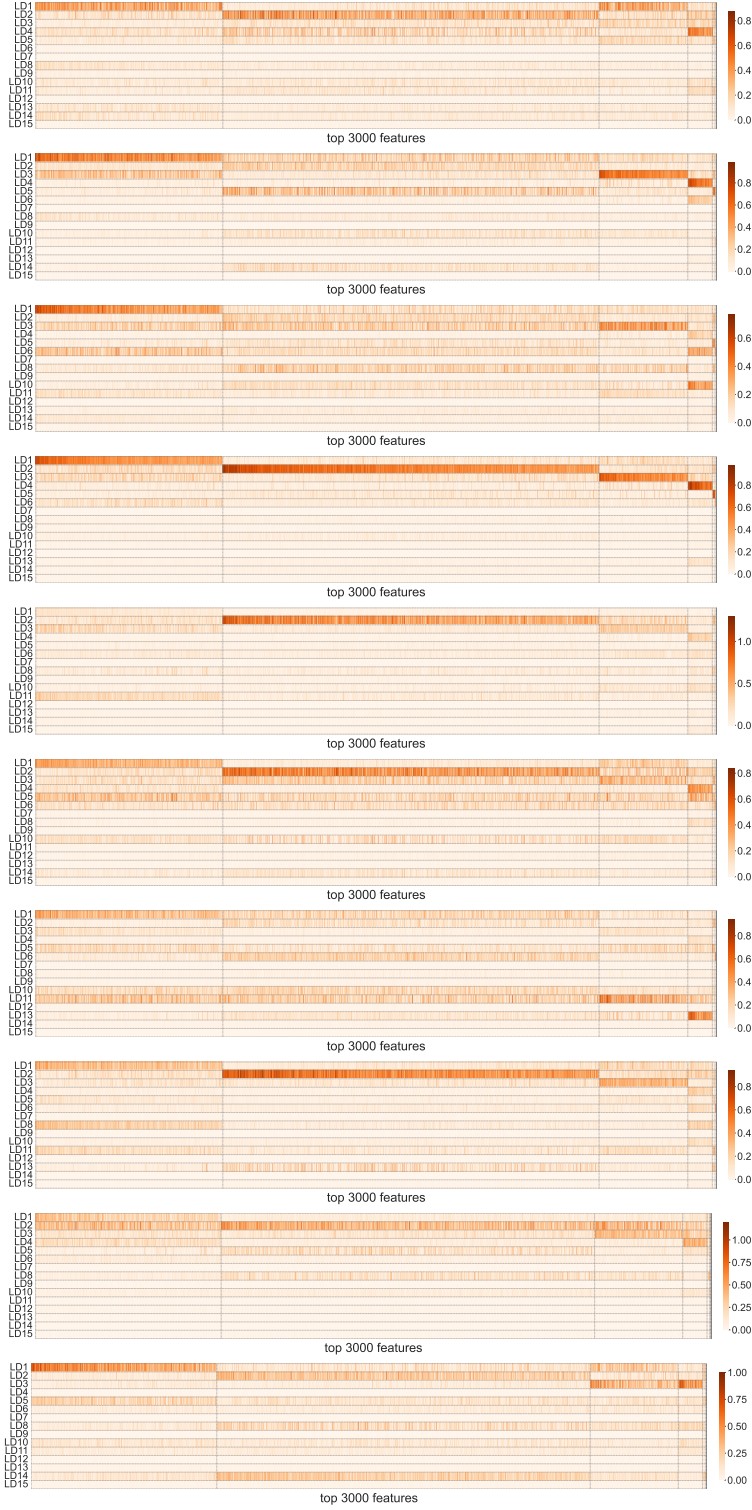

## G.6   White wine: FVH-LT result with bfVAE

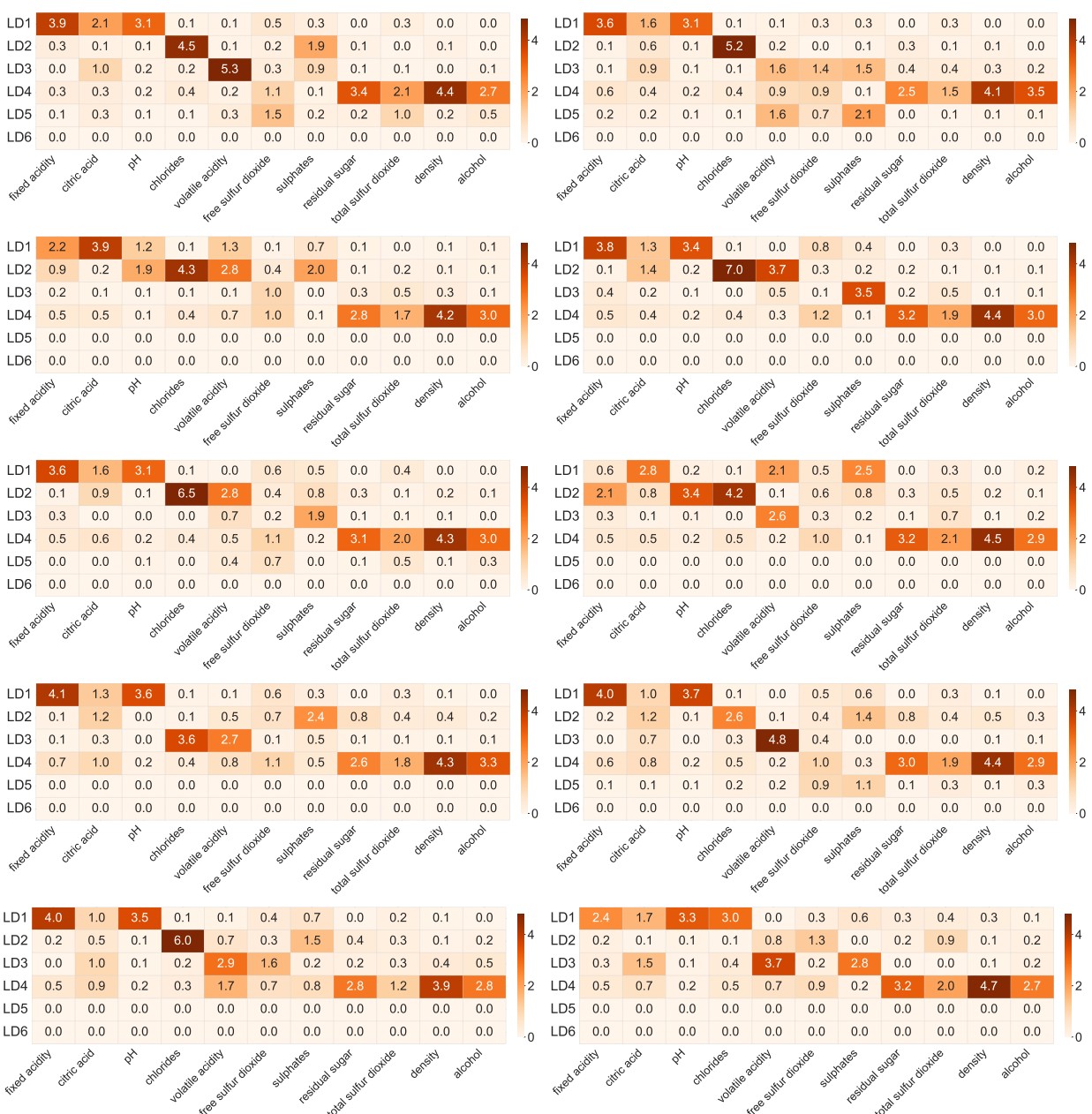

## G.7   White wine: DBSR-LS result with bfVAE

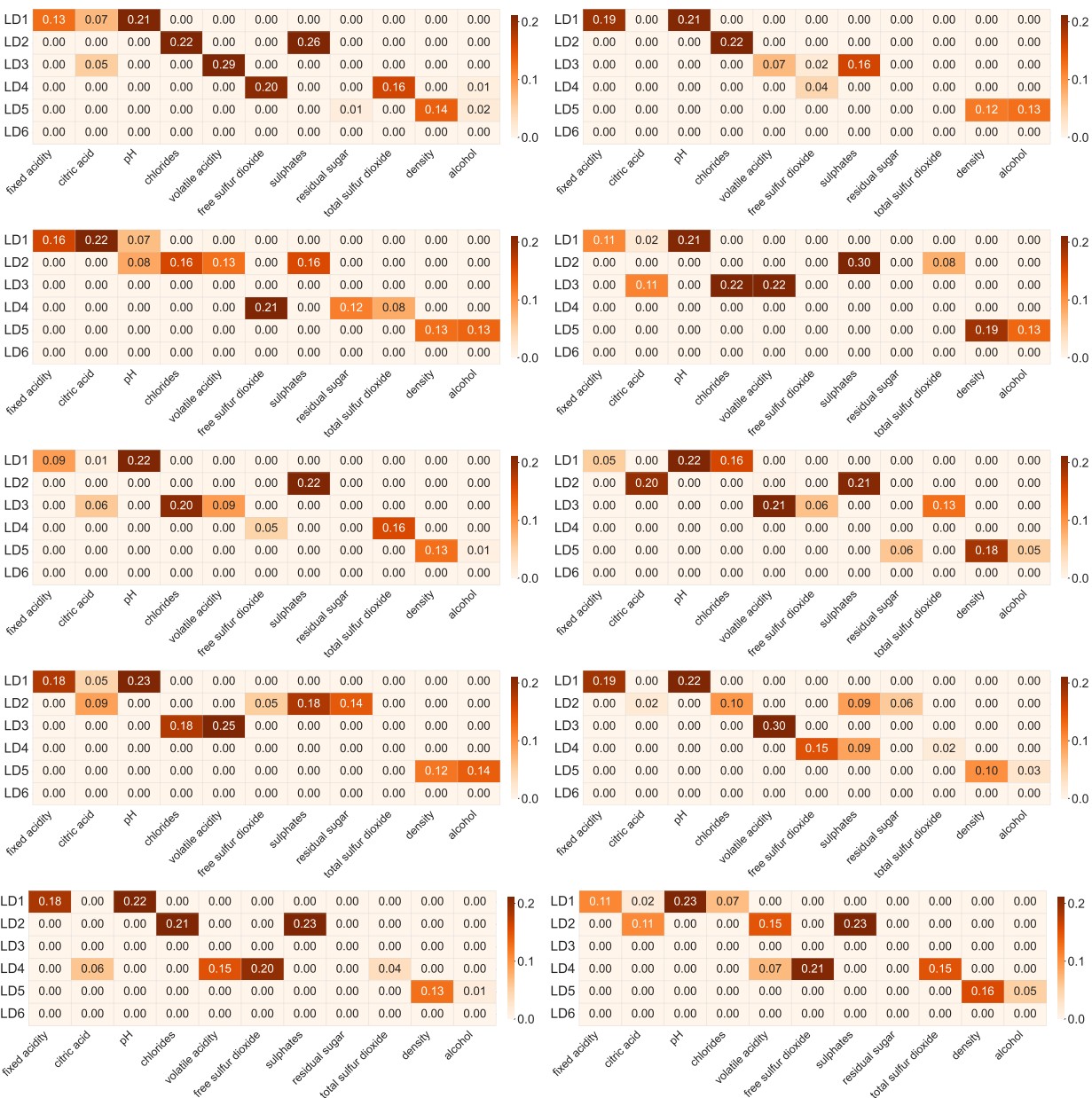

## G.8    2018 FIFA statistics data: FVH-LT result with bfVAE

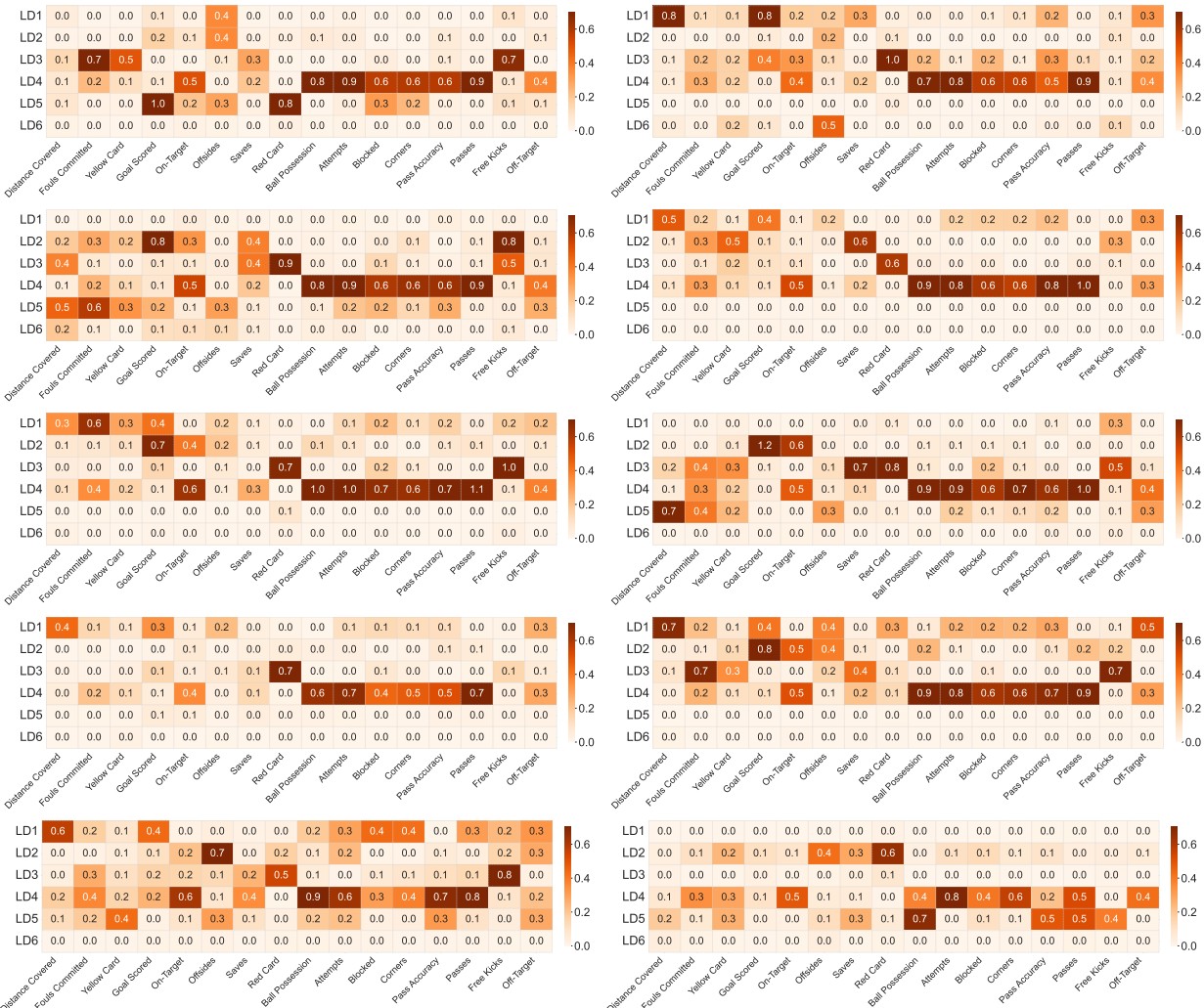

## G.9 2018 FIFA statistics data: DBSR-LS result with bfVAE

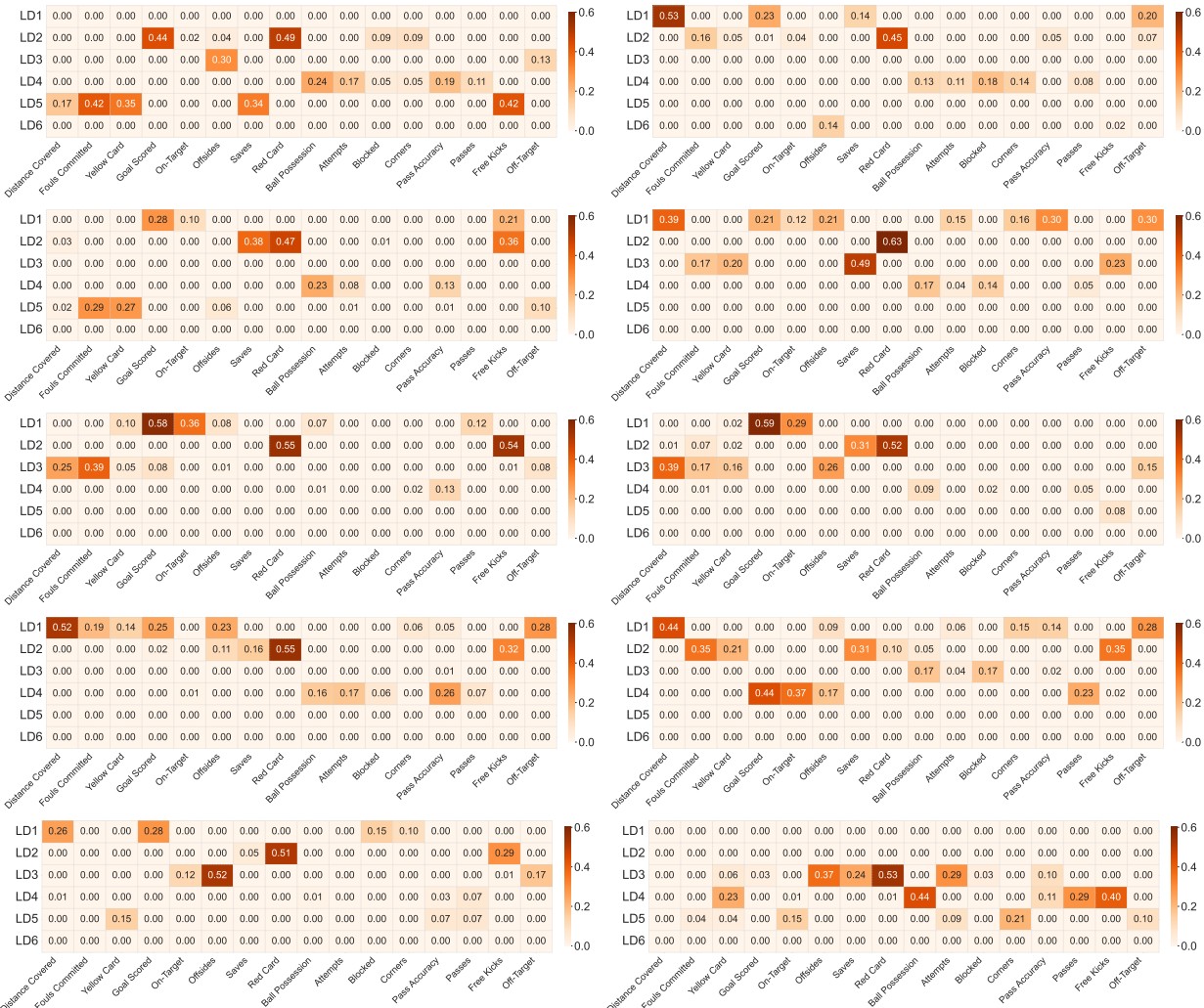

## G.10    MNIST: FVH-LT result with bfVAE

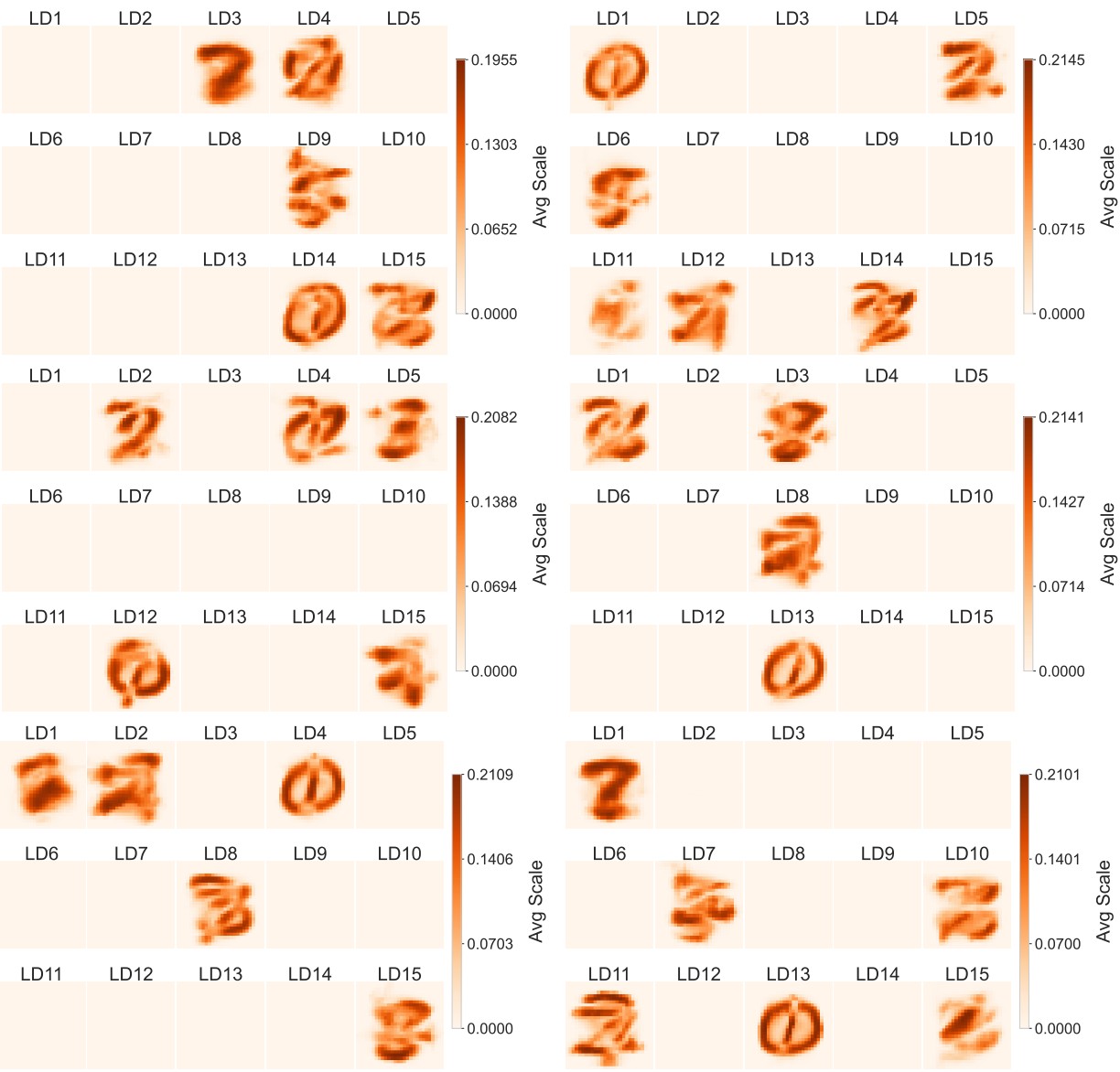

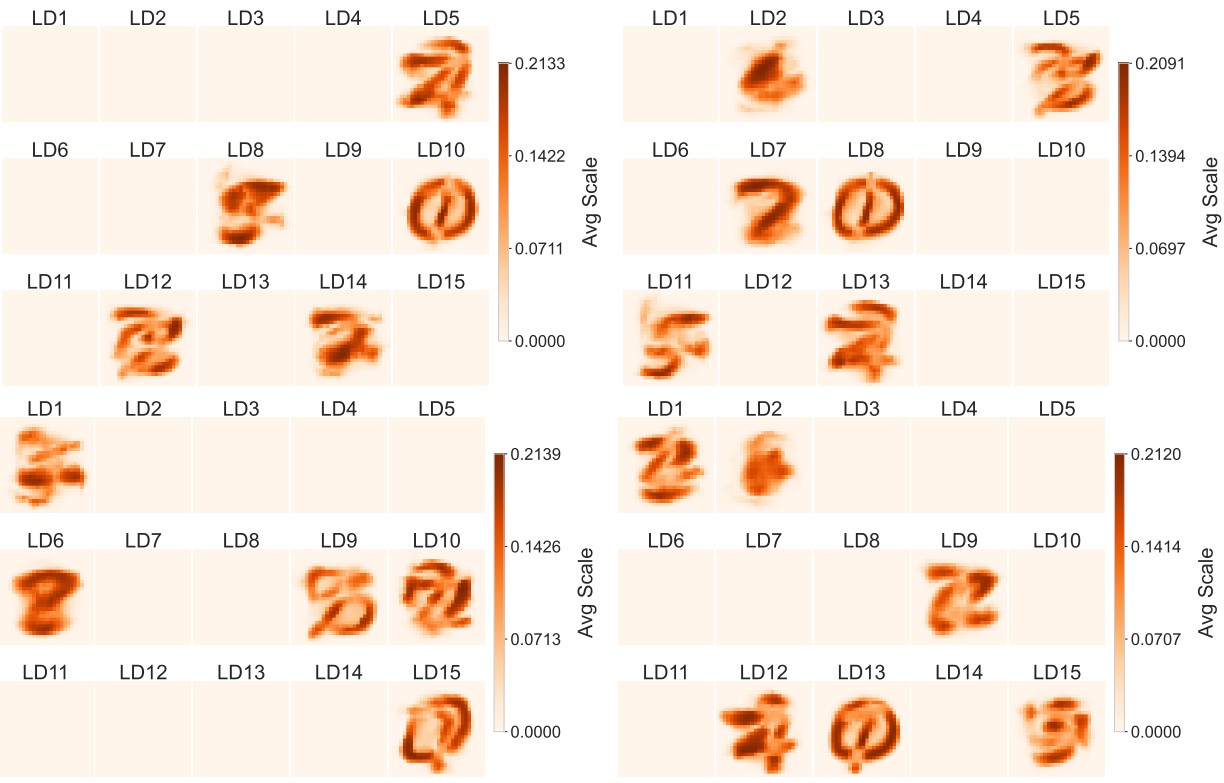

## G.11 CelebA: FVH-LT result with bfVAE

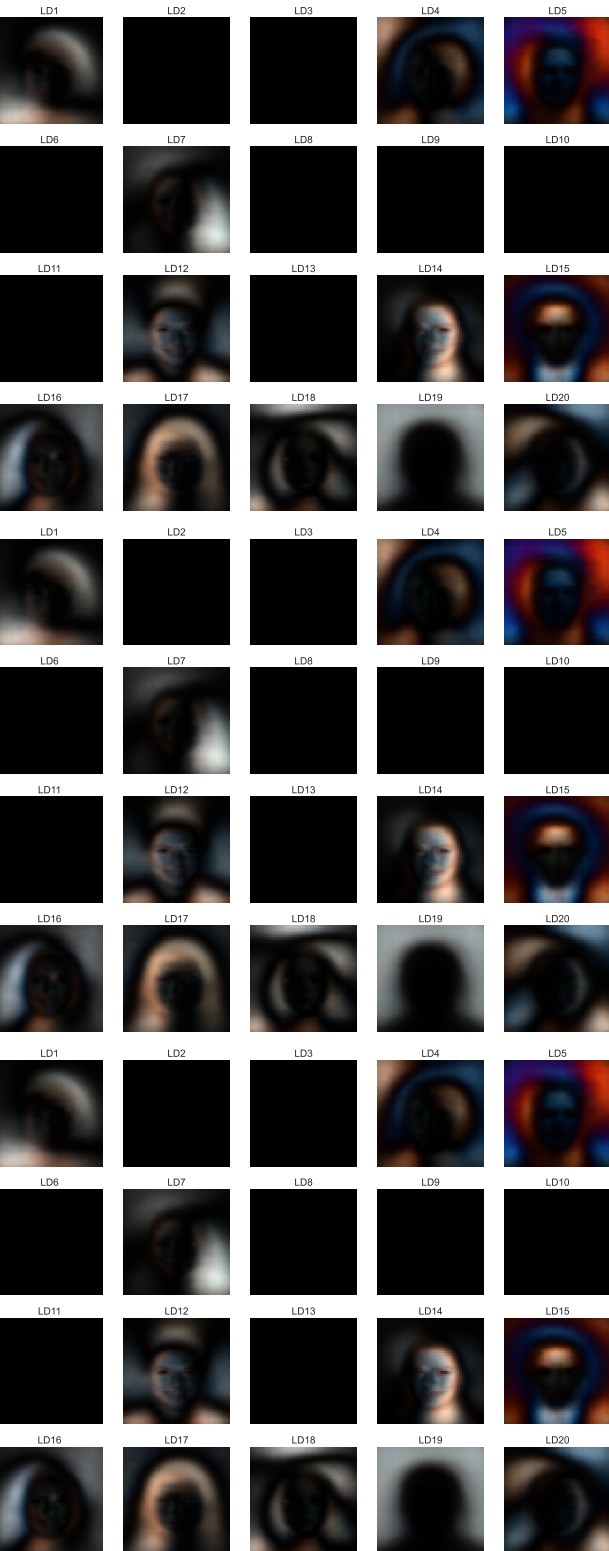

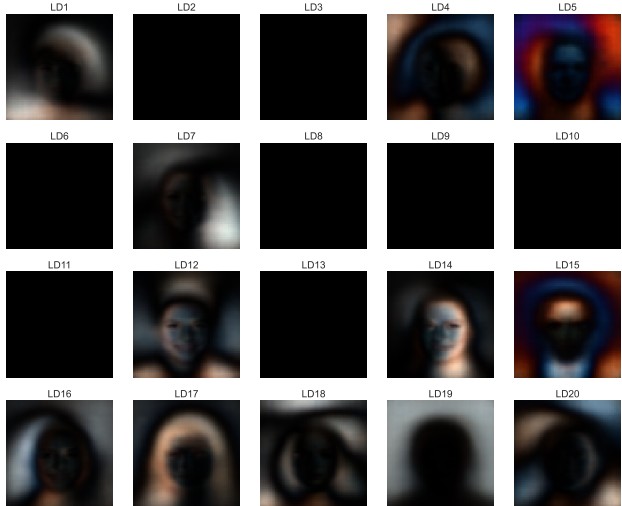

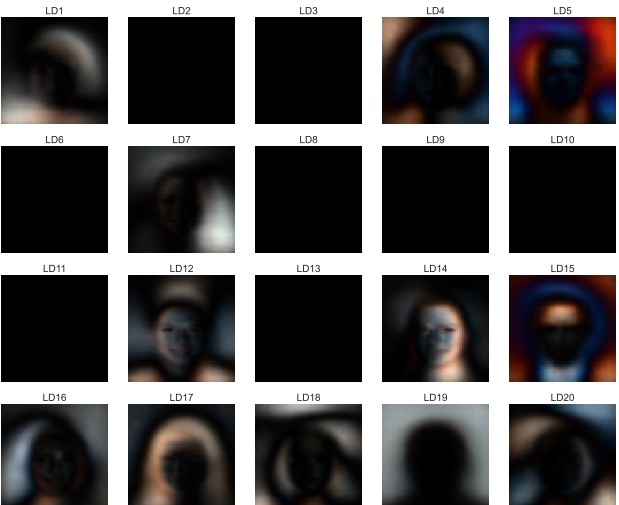

## H   bf-CVAE Implementation and FVH-LT Results

### H.1   bf-CVAE Implementation

We slightly modified the bfVAE structure and the FVH-LT procedure by incorporating the wine quality score $y$ as part of the input, i.e., $\mathbf{x}' \leftarrow (\mathbf{x}, y)$, to the encoder. We then appended $y$ to the learned latent space $\mathbf{z}$ i.e., $\mathbf{z}' \leftarrow (\mathbf{z}, y)$, before being passed to the decoder. We performed LT on $y$ with its range in the observed data while fixing the latent $\mathbf{z}$ at a randomly drawn sample from its posterior distribution. Variances of the generated $\hat{\mathbf{x}}$ were computed.

### H.2   FVH-LT Results across 10 runs

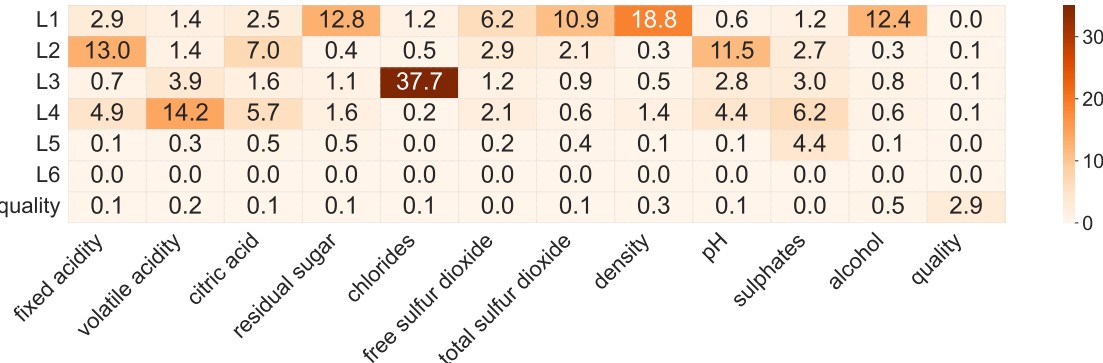

### H.3   Single run

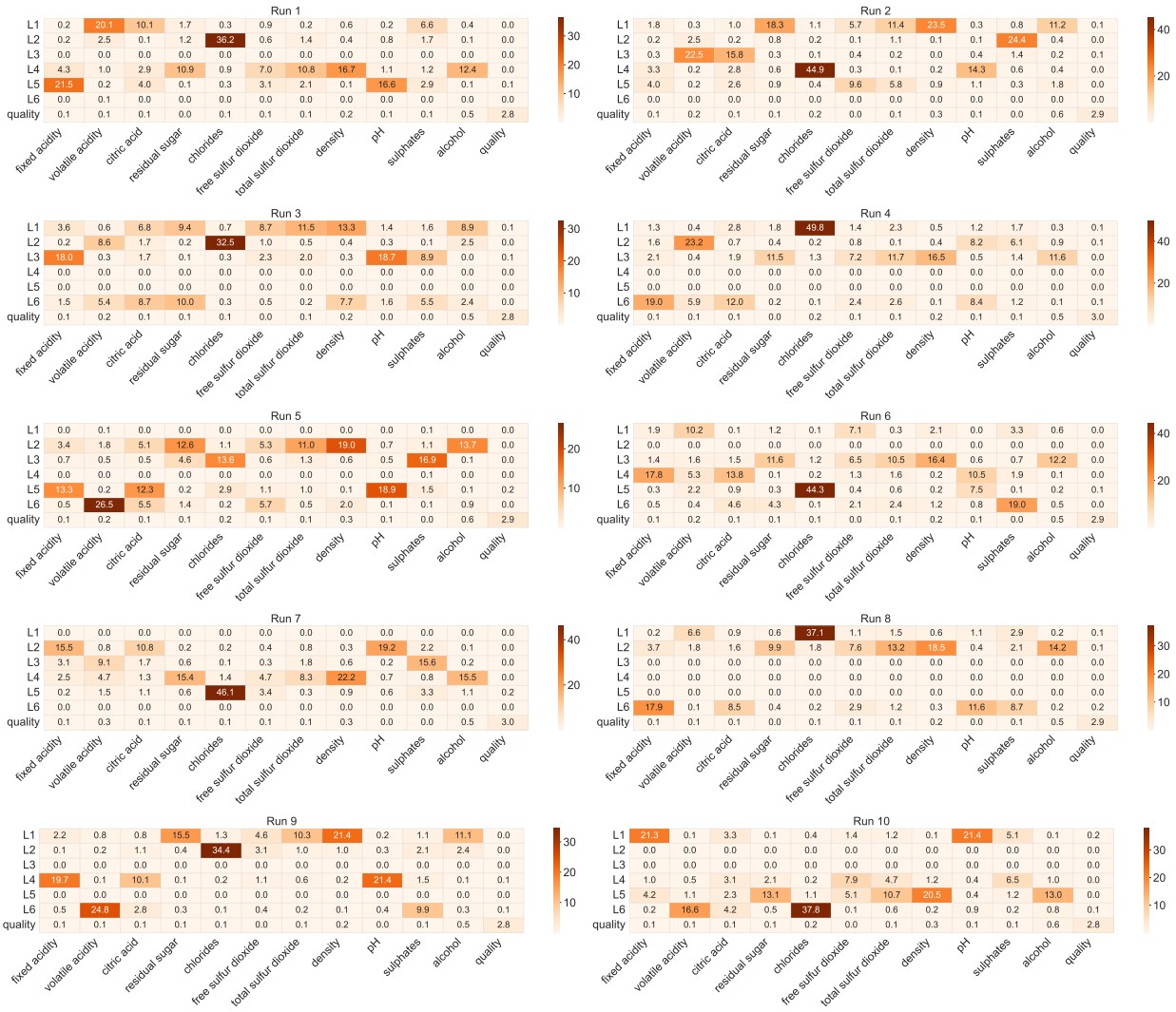

