# OpenReview forum: "A Unified Latent Space Disentanglement VAE Framework with Robust Disentanglement Effectiveness Evaluation"
_TMLR — Under review for TMLR_

### Review · Reviewer_zUe4 · 2026-05-17

**Summary Of Contributions:**

The authors present bfVAE, which applies the relaxation from beta VAE to the factor VAE (fVAE), and introduces a capacity target term on the latent prior KL. Additionally, the authors present two methods (FVH-LT and DBSR-LS) for quantifying how well a latent from a VAE can be disentangled across its components, aggregated into a "latent space disentanglement index" (LSDI) metric.

Overall, the writing needs improvement. The first 5 pages are highly repetitive for just describing their contributions. For example, Section 1.2 is unnecessarily verbose for introducing their methods, and Section 2 could be condensed into stating that there is a tradeoff between enforcing independent latent factors, latent regularization, and reconstruction performance. The notation in Section 3 is also confusing. In section 3.2, the authors use superscripts with parentheses, terms like "D[,k]" to index a matrix, and the dimensions of some terms are denoted with subscripts. The abundance of superscripts and subscripts (some of which have commas and others that do not), ambiguous unbolded/bolded vectors, and unconventional notation makes it difficult to follow.

Regarding the FVH-LT and DBSR-LS: The methods are presented algorithmically; however, the design choices are not well motivated. The algorithms are heuristic in nature, and without a better discussion of the design choices (e.g., being influenced by causality or compression literature), they are not well-motivated mathematically. Regarding DBSR-LS: using linear regression to approximate the mean of the latent posterior could be problematic. Can the authors discuss these potential implications? The encoder network is highly nonlinear, and it's unlikely that the latent relationship can be summarized linearly, especially in higher dimensions. On a positive note, their proposed FVH-LT method does not rely on such a linear simplification, and I believe applying GAS to FVH-LT (and techniques in this direction) offers a much more compelling case (the authors note that FVH-LT performed slightly better than DBSR-LS -- so perhaps this is related).

A recurring issue is that the paper treats $\beta$=1 as a neutral baseline (e.g., when comparing against Factor VAE). But the reconstruction term (T1) scales with data dimensionality and the KL term (T2) scales with bottleneck size, so $\beta$=1 implies a different tradeoff for every dataset and architecture. In practice, a simple change in loss reduction for T1 (e.g., using mean instead of sum over pixels) implicitly rescales $\beta$. Claims about "posterior collapse at $\beta$=1" are therefore misleading without considering the choice of dataset and bottleneck size. Relatedly, it's unclear whether the introduction of the capacity target C is meaningful. The proposed objective $\beta$ *|KL - C| with annealed C appears closely related to simply annealing $\beta$ against the standard $\beta$ * KL penalty: in both cases, the KL follows a controlled trajectory. The authors should clarify what this formulation buys beyond a reparameterization of the $\beta$ schedule, and ideally show that the two are not equivalent. Furthermore, since "bfVAE reduces to the fVAE formulation when $\beta$=1 and C=0" and "for tabular studies, empirical studies suggest C=0 works well", it is unclear if bfVAE provides anything new to the tabular setting (considering tabular data is one of the authors' primary motivations) beyond the fVAE. It feels overstated to introduce this as a new method, since using fVAE with a $\beta$ different from 1 is a relatively straightforward modification.

**Additional Comments:**

Overall, I cannot recommend this paper for acceptance in its current form. There needs to be substantial improvements to the writing and presentation to be reconsidered.

**Audience:**

Yes

**Audience Explanation:**

I think there is an audience for interpretable latents. However, this paper needs to largely improve its presentation, writing, and mathematical motivation.

**Claims And Evidence:**

No

**Claims Explanation:**

I ask that the authors please address my concerns about the chosen $\beta$ for fVAE, as it has a significant effect on the baseline results for fVAE. Furthermore, claiming to be better than all methods on the FA15 dataset is a bit misleading. For one, the vanilla VAE's objective is not to disentangle the latent representation, so it is an ill-posed baseline for this task. Second, DIP-VAE-1 assigns the same components as bfVAE (if you take the argmax over the assignments), so both methods reach the same conclusion effectively.

**Requested Changes:**

1) The figures for the disentanglement of image datasets (e.g., Celeb A and MNIST) do not add much value. What defines a good latent in this case? It is hard to tell visually, and it's unclear what is considered "better" than a standard VAE.
2) Better mathematical motivation for the LSDI metric, which seems to be a heuristic / ensemble of choices.
3) The presentation can be much more concise. Writing and figures can be greatly condensed and simplified.
4) Mathematical notation is unconventional and difficult to follow, especially for FVH-LT and DBSR-LS

---

> ### Author Response · Authors · 2026-06-07
>
> THANK YOU for the careful and constructive comments. We revised the manuscript accordingly. Due to the 5K character limit, we briefly summarize our responses below. Please refer to the list of revisions and the revised manuscript for details.
>
> **C below stands for Comment, R for Response**
>
> **C1: Writing needs improvement**
>
> **R1:** We condensed the manuscript, corrected typos, and made the notation more consistent and clear.
>
> **C2: Motivation for FVH-LT and DBSR-LS; limitation of DBSR-LS**
>
> **R2:** The motivation is clarified at the beginning of Sec 3.  A more technical motivation for FVH-LT is added at the end of Sec 3.1;  the rationale behind DBSR-LS is added at the end of Sec 3.2 with its computational complexity provided in the newly added Sec 3.4 and its limitation further discussed in Sec 5. We also added the computational time in the experiments in the newly added Sec 4.4.
>
> Briefly, FVH-LT is motivated by LT but makes it quantitative rather than purely visual. It provides a feature-level sensitivity summary from coordinate-wise latent perturbation. For a trained decoder $g(z)$, FVH-LT measures how much reconstructed feature $\\hat x_j=g_j(z)$ changes when only latent dimension $z_k$ is varied. Specifically, if $g_j(z)$ is locally approximately linear over the LT grid, then
> $V(g_j(z_{-k},z^{\\ast}_k))\\approx(\\partial g_j(z)/\\partial z_k)^2 V(z^{\\ast}_k)$,  which is closely related to an average squared decoder sensitivity of feature $j$ to LD $k$.
>
> DBSR-LS is not intended to replace the nonlinear encoder or provide a predictive model of $q_{\\phi}(z|x)$ (the encoder itself would be sufficient for that reason, no need for a linear proxy), but rather an interpretable linear surrogate for 1st-order latent-feature associations. Its dirty-model structure separates shared associations from LD-specific ones: if LDs are separated, their sparse feature-association profiles should differ; if LDs rely on the same features, DBSR-LS reveals this overlap.
>
> **C3: $\\beta=1$, capacity target $C$, and claims about bfVAE superiority**
>
> **R3:** We agree that posterior-collapse claims must be tied to dataset, architecture, loss scaling, and hyperparameter choices. Our use of $\\beta=1$ was meant as a reference to the standard VAE/FactorVAE objectives both of which do fix $\\beta=1$, not as a claim that $\\beta=1$ is neutral or optimal. We removed the text where collapse was claimed without context provided. Our ablation/benchmark studies were correctly run as they tuned each VAE type within its own loss formulation rather than fixing hyperparameters at bfVAE-optimal values.
>
> We have streamlined Sec 2.2 to clarify the role of $C$, removing the redundant text that previously obscured this explanation. Briefly, $C$ specifies the target latent information level while $\\beta$ controls how strongly $C$ is enforced: the standard KL-penalized loss $\\beta \\mathrm{KL}(q_{\\phi}(z\\mid x)|p(z)) $ penalizes larger KL values whereas $\\beta|\\mathrm{KL}(q_{\\phi}(z\\mid x)|p(z))-C|$  penalizes deviation from $C$.  When $C=0$, bfVAE still differs from FactorVAE with the additional $\\beta$.
>
> We softened claims of bfVAE superiority and revised the contribution to focus on bfVAE unifying disentanglement VAE formulations and achieving stronger disentanglement-reconstruction tradeoffs than evaluated benchmarks in the studied settings, which is supported by the SIGNIFICANTLY expanded the experiment Sec, with VAE benchmarks  running  in all datasets.
>
> **C4: Image-dataset figures add limited value**
>
> **R4:**  While we believe the image experiments strengthen our work, we recognize that the original draft did not clearly convey their value. We have revised the figure captions to include key interpretations and takeaways. We also added benchmark VAE results in in suppl. materials as baselines to bfVAE.
>
> **C5: Better mathematical motivation for LSDI**
>
> **R5:** We added the practical motivation and an axiomatic interpretation at the start of Sec 3.5, we revised LSDI (now named LSSI) with two correction factors and provided the rational for each term in two paragraphs after Def 2. In brief, a well-separated representation should have minimally overlapping feature-association profiles, assign low scores to redundant LDs, be scale-invariant, and ``ignores'' inactive LDs.
> The raw LSSI score is
> $\\frac{\\sum{k<j} w_{kj}d(k,j)}{\\sum_{k<j} w_{kj}},$  a mass-weighted average of normalized non-overlap between the LD-feature association profiles, where  $ w{kj}=|A_{[k,]}|1+|A{[j,]}|1$  and $d(k,j)=\\frac{||A_{[k,]}|-|A_{[j,]}||1}{|A{[k,]}|1+|A{[j,]}|1}$, $c_1$ measures feature coverage and $c_2$ measures LD utilization and penalizes signal concentration in only a few LDs.
>
> **C6: Writing and figures should be condensed and simplified**
>
> **R6:** We revised the manuscript accordingly and simplified some figures and moved some to the supplementary materials.

---

### Review · Reviewer_xBHV · 2026-05-19

**Summary Of Contributions:**

Thi manuscript propose two things regarding disentangled representation learning: The first is bfVAE, a VAE objective intended to unify and combine ideas from β-VAE and FactorVAE. The loss combines reconstruction, a capacity-constrained KL term, and a total-correlation penalty on the aggregated posterior. The authors then propose two post-hoc interpretability/evaluation procedures: FVH-LT, which quantifies feature sensitivity under latent traversal, and DBSR-LS, which regresses learned latent posterior means on input features using a multi-task sparse model. These matrices are aligned across repeated runs using GAS, a greedy latent-dimension matching procedure, and summarized by a scalar LSDI score meant to quantify structural separation without access to ground-truth generative factors. The paper emphasizes tabular data, where standard image-oriented disentanglement tools are less useful, and evaluates on synthetic factor-analysis datasets, wine-quality data, FIFA 2018 data, but also on image benchmarks MNIST and CelebA.

**Audience:**

Yes

**Audience Explanation:**

Disentangled representation learning is deeply related with unsupervised learning and generative modeling, both key topics for the TMLR audience

**Claims And Evidence:**

Yes

**Claims Explanation:**

The core practical motivation is strong: disentanglement metrics based on known generative factors are often unusable in real data, and qualitative latent traversals are not informative for tabular data. The authors correctly identify a real gap, i.e., the need for model-agnostic, feature-level, repeatable latent-space diagnostics that can work when ground-truth factors are absent. Their FVH-LT heatmaps and DBSR-LS coefficient matrices are potentially useful tools for practical interpretation, especially for tabular applications. The usefuleness of LSDI can be seen in multiple cases, especially in Tab. 2 where it is clear to see the separation between bf-VAE and DIP-VAE-I in compared to the supervised Disentanglement Score. GAS addresses the important issue of latent dimensions possibly switching order across random initializations. The emphasis on repeated runs, aggregation, and avoiding cherry-picked latent traversals is valuable. Finally, the tabular focus is both timely and refreshing, since currently this is a trending data type to work and most applications demand interpretability.

**Requested Changes:**

The paper has several weaknesses in its current state. In the following I will mix the weaknesses with some questions, which I believe would be beneficial for the authors in terms of what to modify:

- The novelty of bfVAE is overstated. The objective is largely a combination of known TC-penalized VAE and capacity-controlled β-VAE ideas [1,2]. The paper should more directly position itself especailly relative to β-TCVAE, in my understanding there is in fact nothing different between the two beyond the capacity term C. This deserves crucial attention.

- Related to the above, Section 2.2 gives an information-bottleneck interpretation: reconstruction is the relevance term, the KL/capacity term controls transmitted information, and the TC term shapes dependence across latent dimensions. This interpretation is reasonable, but much of it restates known facts about VAEs, β-VAEs, and the KL decomposition. The claim that KL controls the information rate and that TC penalizes dependence among latent coordinates is standard in the disentanglement literature [2], and is of course even more standard in Information Theory. Restating these facts might not deserve a complete and elongated section.

- Shouldn't the sample estimated posterior for a latent variable be marginalized over x (before the loss in Eq. (1))? Otherwise how is this effectively marginalizing over data samples?

- LSDI measures structural separation of an association matrix, not necessarily disentanglement. It may be better framed as a latent-feature separation score which is great for interpretability, but knowing whether or not there is a causal association to a generating factor remains unclear. Of course, this issue can be raised for essentially all evaluation metrics in this research field, which is why the authors should consider to comment more in depth.

- The associated computational cost with computing the proposed evaluation metrics may limit scalability, especially since one has to run GAS over repeated runs for FVH-LT/DBSR-LS. This can become expensive, particularly for high-dimensional scientific data, therefore the authors should comment on this aspect and also present some results regarding the complexity of evaluation.

- Connecting to the previous, suppose for example we wish to apply these disentanglement metrics on domains like omics or gene expression, where disentangled VAE representations have found great usage.  How sensitive are LSDI and FVH-LT to the number of inactive latent dimensions? Furthermore, given feature importance may be extremely sparse and biologically redundant, should one rely on the aggregate LSDI score or inspect the full feature-latent matrix? In general, considering how to use the score and the interpretability analysis on large-scale data is quite important, since for many applications VAEs are precisely used to compress very high-dimensional inputs. In the paper we see this on the image datasets, where unfortunately the proposed evaluation metrics do not elucidate much.

[1] Chen, Ricky TQ, et al. "Isolating sources of disentanglement in variational autoencoders." Advances in neural information processing systems 31 (2018).

[2] Burgess, Christopher P., et al. "Understanding disentangling in $\beta $-VAE." arXiv preprint arXiv:1804.03599 (2018).

---

> ### Author Response · Authors · 2026-06-07
>
> THANK YOU for the careful and constructive comments. We revised the manuscript accordingly. Due to the 5K character limit, we briefly summarize our responses below. Please refer to the list of revisions and the revised manuscript for details.
>
> **C below stands for Comment, R for Response**
>
> **C1: Novelty of bfVAE**
>
> **R1:** We removed text where novelty of bfVAE was claimed. The revision positions bfVAE as an framework unifying the existing disentanglement VAE models, together with ground-truth-free interpretation and evaluation tools. The SIGNIFICANTLY expanded experiment supports the unification,  showing improved disentanglement-reconstruction trade-offs than baseline VAE models under the studied settings.
>
> **C2: Information-bottleneck interpretation**
>
> **R2:** We agree that some points in Sec 2.2 exist in the  literature. We SIGNIFICANTLY condensed this Sec and clarified that it motivates our bfVAE formulation rather than claiming new theoretical insights. The revision explains how reconstruction corresponds to relevance, the KL/capacity term regulates latent information, and the TC term discourages dependence among latent coordinates.
>
> **C3: Aggregated posterior notation**
>
> **R3:** We agree that our previous wording was misleading. The distribution is NOT marginalized over $x$ as our previously described. We removed that as well as the phrase “marginalized posterior distribution of $Z$,” and made sure the same notations as in Chen et al. (2018) and Makhzani et al. (2015) are used.
>
> **C4: LSDI measures structural separation, not causal disentanglement**
>
> **R4:** We agree. In the strongest sense, disentanglement means recovering independent causally generative factors, which generally requires ground truth, interventions, or strong domain assumptions. Our metric is ground-truth-free and cannot by itself prove causal recovery. Following your suggestion, we renamed LSDI to Latent Space Separation Index (LSSI) and revised its definition and the surrounding narrative. LSSI is framed as a measure of structural separation in the latent-feature association matrix, useful for interpretation rather than proof of causal disentanglement. We also added two correction factors to better reflect its meaning, and a LSSI variant for the purposes of benchmarking on synthetic data. when  ground-truth factors are  known
>
> **C5: Computational cost and scalability**
>
> **R5:** The computational cost was discussed in the previous version, but it was embedded in a paragraph in Sec 3. Given your comment, we have expanded it to a newly added Sec 3.4 to include the added cost of GAS. We also added Sec 4.4 that shows the run time for each procedure in the experiments. GAS is run after each run produces an association matrix and is not the computational bottleneck; rather, repeated VAE training, FVH-LT traversal, or DBSR-LS regression are notably more costly. We also added computational recommendations: parallelize runs, batch LT over samples/LDs/grid points, vectorize GAS, and use DBSR-LS selectively for very high-dimensional data.
>
> **C6: Inactive LDs and high-dimensional scientific data**
>
> **R6:** The proposed interpretation tools - FVH-LT, DBSR-LS, GAS, LSSI - are designed to accommodate over-specified LD $K$ and the presence of non-informative LDs. This robustness is supported by the informative-LD identification step and the alignment of informative LDs through GAS, and is consistent with the experimental results. In the 1st version, we evaluated only image data (Tab 1), which are high-dimensional with substantial feature redundancy. Motivated by the your comment, we added a real high-dimensional tabular RNA-seq dataset ($n=640, p=20,264$) to the experiments. The results further demonstrate that the proposed pipeline remains relatively insensitive to over-specification of $K$. For LSSI, we further improved the metric (see R4 above).
>
> In summary, for high-dimensional tabular data such as RNA-seq datasets, all proposed tools can still be applied. Furthermore, in many scientific applications, domain knowledge may be available to further interpret the learned latent space. For example, feature-LD associations identified by FVH-LT or DBSR-LS can be summarized at the pathway, gene-set, or module level when such annotations are available. We have added this discussion to Sec 4.2.

---

### Review · Reviewer_QUpR · 2026-05-24

**Summary Of Contributions:**

This work proposes a new Variational Auto-Encoder (VAE) framework, termed the beta-factor VAE (bfVAE). This work also introduces two VAE "evaluation toolkits"- Feature Variance Heterogeneity via Latent Traversal (FVH-LT) and Dirty Block Sparse Regression in Latent Space (DBSR-LS). The contributions are as follows:

1) The bfVAE framework aims to compress the input data whilst minimizing reconstruction error and encouraging statistical independence (disentanglement) amongst the Latent Dimensions (LD). It is a generalized disentanglement framework applicable on multiple data modalities (tabular, image etc).
2) Two methods - Feature Variance Heterogeneity via Latent Traversal (FVH-LT) and Dirty Block Sparse Regression in Latent Space (DBSR-LS) - are proposed. FVH-LT and DBSR-LS identify and quantify "latent-feature associations" (associations between latent variables and input features) in cases where knowledge of the ground-truth generative factors is unknown.
3) A Greedy Alignment Strategy (GAS) is proposed in order to address the problem of label switching/label misalignment.
4) The above methods are evaluated through experiments on various data types. Specifically, FVH-LT and DBSR-LS are applied to various VAEs.


Strengths:

1) The work is well presented and the overall quality of the writing in the paper is noteworthy.

2) I think bfVAE approach is novel and deserves attention.


Weaknesses:

1) The experimental section could benefit from being broader and more rigorous.

**Audience:**

Yes

**Audience Explanation:**

Researchers in the field of unsupervised representation learning and information-theoretic machine learning would be interested in this work.

**Broader Impact Concerns:**

No Broader Impact Concerns.

**Claims And Evidence:**

No

**Claims Explanation:**

The authors make the following claims:

>1) The framework, bfVAE, is general disentanglement framework that can support data of multi modalities.

This claim is supported by Eq. $1$, which formulates the loss function of bfVAE from an "information bottleneck perspective" to minimize the reconstruction error, compress the data, and encourage disentanglement between the LDs. Additionally, in Section $4$: Experiments, Table $1$ shows the use of both tabular and image data, and Table $2$ shows bfVAE exhibits a high disentanglement score on the synthetic tabular dataset FA $15$.

>2) The evaluation tools FVH-LT and DBSR-LS measure the latent–feature associations that can be conveniently visualized via a
heatmap, providing an effective interpretable summary of the individual LDs, and do not require access to ground truth generative features.

I believe this claim is supported by evidence. Algorithm $1$ and $2$ show how FVH-LT and DBSR-LS work. Results on synthetic tabular data (such as those in Fig. $5$ and $8$), and image data (such as Fig. $10$) show the heat map output of FVH-LT on datasets with/without knowledge on generative factors.

>3) bfVAE is superior to other VAE methods.

I believe this claim is not fully supported by evidence. bfVAE has not been independently evaluated using established, disentanglement metrics in the current literature. Additionally, for multiple datasets considered in this work (FA24, FA100, White Wine, FIFA2018...) the performance of bfVAE has not been compared to other existing VAE methods. In summary, it is difficult to fully appreciate the novelty of bfVAE as well as FVH-LT without a more comprehensive comparison of bfVAE to other VAEs, and FVH-LT to other established metrics. (I expand on this in the "Requested Changes" Section)

**Requested Changes:**

> 1) The work is missing a more rigorous evaluation of bfVAE and FVH-LT

I believe this to be my primary concern. The paper introduces a VAE model-agnostic evaluation protocol, FVH-LT, which measures latent–feature associations by sweeping through each LD and measuring the changes in reconstructed outputs via the decoder. While this is a creative and practical approach to interpreting VAE behavior, one may pose the following questions:

a) Does FVH-LT conflate latent space disentanglement with decoder reconstruction fidelity? Because FVH-LT relies on the decoder to measure feature changes, could a baseline model (such as FactorVAE) still receive a poor FVH-LT association matrix simply because its decoder yields softer or less sensitive reconstructions?
b) Conversely, bfVAE might score higher purely due to superior decoder performance rather than better structural separation in the latent space itself. To clarify this relationship and validate the metric, can the trained models been evaluated using established disentanglement metrics (e.g., FactorVAE metric, "Kim, H., & Mnih, A. (2018). Disentangling by factorising. Proceedings of the 35th International Conference on Machine Learning (ICML), PMLR")? Showing a strong correlation between LSDI and other metrics would prove that FVH-LT is genuinely capturing latent structure rather than decoder sensitivity.

> 2) I would appreciate the authors' insights on the following regarding GAS:

a) How sensitive is the final LSDI score to the accuracy of GAS? For instance, if GAS misaligns a latent dimension, does the resulting 'blur' drastically penalize the score?

b) Did the authors consider an ablation study comparing GAS to standard post-hoc alignment baselines (e.g., Hungarian matching)? Understanding why a custom procedure may be necessary here would be highly informative.

> 3) It is unclear why VAEs are deployed on low-dimensional tabular data with low data redundancy.

a) I acknowledge that the work successfully maps latent-feature associations for such types of datasets. However, I am left wondering about the fundamental computational costs. VAEs are commonly deployed on images and other high-dimensional data with only few non-redundant features. Such data types take advantage of the VAEs compression capability. What is the targeted real-world use case for disentangling low-dimensional tabular data and are there no stand-alone disentangling methods that are more computationally efficient than a VAE?

b) It would be helpful to understand the broader utility of bfVAE. Does bfVAE maintain high-quality data reconstruction alongside this high disentanglement? It would be highly reassuring to see the reconstruction error metrics reported alongside the disentanglement scores to ensure the model isn't sacrificing data fidelity for latent separation.

> 4) Minor comments:

a) What are the specific edge cases or regimes where DBSR-LS is preferred over FVH-LT?

b) Algorithm $1$, line $1$ is missing a comma. It should be $$
(\mathbf{z}_1, ..., \mathbf{z}_K)
$$

c) inconsistent symbols. Fig. $1$ uses $\hat{x}$, but its caption says $\tilde{x}$

---

> ### Author Response · Authors · 2026-06-07
>
> THANK YOU for the careful and constructive comments. We revised the manuscript accordingly. Due to the 5K character limit, we briefly summarize our responses below. Please refer to the list of revisions and the revised manuscript for details.
>
> **C below stands for Comment, R for Response**
>
> **C1/C2**: Experimental sec should be broader and more rigorous. Claim that bfVAE is superior to other VAEs is not fully supported.
>
> **R1/R2**: We totally agree. To address the comment, 1) we SIGNIFICANTLY expanded the experiment by adding a high-dimensional RNA-seq real tabular data experiment, running all VAE baselines in all datasets, adding two existing disentanglement metric baselines (Kim and Mnih'18; Higgins et al.'17)  in synthetic datasets with known ground-truth factors; 2) we revised the paper to avoid a general superiority claim for bfVAE and  now state that bfVAE achieves better disentanglement-reconstruction trade-off than the evaluated baselines in our experimental settings.
>
> **C3**: Decoder-mediated FVH-LT; possible confounding with reconstruction fidelity; LSSI validation against established disentanglement metrics
>
> **R3**: We agree that decoder strength and reconstruction quality can affect FVH-LT, since it is  post-hoc, decoder-mediated via LT. A weak decoder may produce a less informative FVH-LT association matrix while better reconstruction may strengthen the FVH-LT signal. One way to mitigate this concern is to  report reconstruction performance, which we have added to all the experiments, in addition to reporting FVH-LT, DBSR-LS, and LSSI. Second, DBSR-LS provides a complementary assessment by estimating feature-LD associations directly from  latent space, rather than through decoder-mediated LT. The results show agreement between FVH-LT and DBSR-LS, suggesting that the identified associations are not solely artifacts of decoder's quality. Third, as suggested, we  added comparisons to established  disentanglement metrics in the synthetic data experiments where we know ground-truth factors since all existing metrics require ground-truth to our knowledge. The trend of LSSI across different VAE models agrees with these the two representative existing  metrics (it should  be noted that the two existing metrics also rely on decoder-mediated LT).
>
> **C4**: Sensitivity of LSDI to GAS and comparison with Hungarian matching.
>
> **R4**: GAS is a post-hoc alignment step for repeated VAE runs to solve label switching in the latent space .  It aligns LDs by thresholded correlations of FVH-LT or DBSR-LS vectors between a run and a reference run on identified informative LDs. Since it stops when no pair exceeds the threshold, GAS avoids forcing weak matches. Our experiment results suggests GAS generates robust alignment and significantly improved the results based on a single run or random alignment.
>
> Hungarian matching (HM) could solve the assignment SUB-problem within GAS, but a na\\"ive application would not work; it would still need informative-LD selection, reference-run selection, and weak-match thresholds in GAS in addition to matching.  While comparing post-hoc alignment algorithms (e.g  optimal transport is another potential choice) for stochastic VAE latent spaces is an interesting direction, a full alignment benchmark is beyond the scope of this work and would distract from our main focus on bfVAE and the proposed ground-truth-free evaluation tools. We have noted it as future work in the Discussion. We have added this discussion in the last paragraphs in Sec 3.3 and Sec 5.
>
> **C5**: Why use VAEs for low-dimensional tabular data given may existing dimensionality reductions? do they preserve reconstruction fidelity?
>
> **R5**: While PCA, ICA, or sparse factor models may be preferable for purely linear factor discovery in tabular data, our motivation of applying disentangled VAEs is more than just data compression, but synthetic data generation, stochastic generative modeling, conditional sampling (eg. the bf-cVAE in the white wine experiment) via VAE with structured interpretable non-linear latent representations. We agree tabular data may be less redundant than images, we thus apply weak bottlenecks, motivating bfVAE’s capacity control. Furthermore, low-dimensional tabular data can arise from even fewer latent factors; in our examples, $p=11-15$  with $K=3-5$.
>
> We also added a new high-dimensional RNA-seq tabular data experiment with $p>20,000$.
>
> We have reported reconstruction errors (Tabs 2, 4 and 6) along with disentanglement results to demonstrate their trade-off. bfVAE yields the best trade-offs vs. examined VAE benchmarks in all the examined settings.
>
> **C6**: When is DBSR-LS preferred over FVH-LT?
>
> **R6**: FVH-LT is decoder-mediated whereas DBSR-LS estimates feature-LD associations directly from learned latent space, DBSR-LS may  serve as a useful complementary approach when decoder-based LT is unstable, especially when $p$ is not very large. We have added a similar narrative to Sec 3.2 and Sec 5.

---

> > ### Comment · Reviewer_QUpR · 2026-06-18
> > **Response to Authors**
> >
> > I would like to thank the Authors for their response.
> >
> > I particularly appreciate the author's expanded discussion on GAS and motivation for using VAEs on tabular data. I also think that adding reconstruction scores greatly improves the authors' argument and quality of the paper.
> >
> > Regrading the comparison with existing disentanglement metric baselines, I agree that bf-VAE has a more favorable disentanglement-reconstruction tradeoff. Comparing bf-VAE to $\beta$-VAE highlights this particularly well. It also seems that the general behavior of the error metric LSSI follows that of other error metrics (i.e., a higher LSSI generally corresponds to a higher "Kim and Mnih'18" and "Higgins et al.'17" score). However, in various datasets, when LSSI determines bf-VAE to be superior, the baseline metric scores show that the performance of bf-VAE also matches a few other methods. So saying that "bf-VAE consistently achieves *highest* scores" is slightly misleading. A more precise assessment would be: "If LSSI determines that bf-VAE outperforms other methods, existing benchmark scores would confirm bf-VAE's high performance, but would not necessarily single out bf-VAE as the *only* highest scorer.". This behavior of LSSI can indicate that LSSI favors/rewards bf-VAE more than other methods. That being said, I believe this to be a minor comment as long as the "disentanglement-reconstruction tradeoff" is discussed and highlighted well enough.

---

> > > ### Author Response · Authors · 2026-06-18
> > > **Response to Reviewer QUpR**
> > >
> > > Thank you for the thoughtful follow-up. We are glad that your comments were addressed and the revised manuscript significantly improved. The constructive comments from the first round were genuinely helpful in strengthening the paper! We also appreciate your additional suggestion on how to best phrase the key takeaways. We will revise the manuscript to emphasize that bf-VAE shows a favorable disentanglement-reconstruction tradeoff, while avoiding overstating its performance relative to the benchmark methods.